# Prognostic genome and transcriptome signatures in colorectal cancers

Luís Nunes[1,10,11], Fuqiang Li[2,3,4,11], Meizhen Wu[2,3,4,11], Tian Luo[2,3,4,11], Klara Hammarström[1], Emma Torell[1], Ingrid Ljuslinder[5], Artur Mezheyeuski[1], Per-Henrik Edqvist[1], Anna Löfgren-Burström[6], Carl Zingmark[6], Sofia Edin[6], Chatarina Larsson[1], Lucy Mathot[1], Erik Osterman[1], Emerik Osterlund[1], Viktor Ljungström[1], Inês Neves[1], Nicole Yacoub[1], Unnur Guðnadóttir[1], Helgi Birgisson[7], Malin Enblad[7], Fredrik Ponten[1], Richard Palmqvist[6], Xun Xu[2], Mathias Uhlén[8,9], Kui Wu[2,3,4,12]✉, Bengt Glimelius[1,12]✉, Cong Lin[2,3,4,12]✉ & Tobias Sjöblom[1,12]✉

Colorectal cancer is caused by a sequence of somatic genomic alterations affecting driver genes in core cancer pathways[1]. Here, to understand the functional and prognostic impact of cancer-causing somatic mutations, we analysed the whole genomes and transcriptomes of 1,063 primary colorectal cancers in a population-based cohort with long-term follow-up. From the 96 mutated driver genes, 9 were not previously implicated in colorectal cancer and 24 had not been linked to any cancer. Two distinct patterns of pathway co-mutations were observed, timing analyses identified nine early and three late driver gene mutations, and several signatures of colorectal-cancer-specific mutational processes were identified. Mutations in WNT, EGFR and TGFβ pathway genes, the mitochondrial CYB gene and 3 regulatory elements along with 21 copy-number variations and the COSMIC SBS44 signature correlated with survival. Gene expression classification yielded five prognostic subtypes with distinct molecular features, in part explained by underlying genomic alterations. Microsatellite-instable tumours divided into two classes with different levels of hypoxia and infiltration of immune and stromal cells. To our knowledge, this study constitutes the largest integrated genome and transcriptome analysis of colorectal cancer, and interlinks mutations, gene expression and patient outcomes. The identification of prognostic mutations and expression subtypes can guide future efforts to individualize colorectal cancer therapy.

Colorectal cancer (CRC) is the third most common and the second deadliest tumour type in both sexes, with 1,900,000 new cases and 900,000 deaths annually. About 20% of patients have metastatic disease already at diagnosis, and another 20% will develop metastases later[2]. From exome[3–7] and whole-genome[8–10] sequencing, the mutational landscape of CRC is best characterized in coding regions, whereas non-coding regions remain understudied. Approximately 80–85% of CRCs are classified as copy-number altered microsatellite stable (MSS), 10–16% as highly mutated with microsatellite instability (MSI) and 1–2% as ultramutated resulting from somatic *POLE* mutations[4,11]. The MSI status predicts response to checkpoint inhibitors[12], whereas *KRAS*, *NRAS* and *BRAF* mutations predict poor response to EGFR-targeted therapies[13]. The WNT, EGFR–KRAS–BRAF, PIK3CA, TGFβ and p53 pathways are regulated by mutations in CRC[1], and several driver gene mutations in these pathways have been linked to prognosis. To advance the understanding of CRC pathogenesis, identify driver events and find prognostic features, we analysed whole genomes along with tumour transcriptomes in a large, population-based CRC cohort with clinical outcomes.

## Mutational landscape

We obtained high-quality whole-genome sequences (average 53-fold coverage) from patient-matched tumour and unaffected control samples along with tumour transcriptome sequence (average 30 million paired reads; Supplementary Table 1) from 1,063 out of 1,126 CRC cases. Of the 1,063 CRCs, 943 were primary tumour surgical specimens and 120 were primary tumour biopsies. Control samples were taken from peripheral blood for 522 cases and from adjacent tissue for 541 cases, which did not introduce batch effects (Supplementary Table 2).

[1]Department of Immunology, Genetics and Pathology, Science for Life Laboratory, Uppsala University, Uppsala, Sweden. [2]HIM-BGI Omics Center, Hangzhou Institute of Medicine (HIM), Chinese Academy of Sciences (CAS), BGI Research, Hangzhou, China. [3]Guangdong Provincial Key Laboratory of Human Disease Genomics, Shenzhen Key Laboratory of Genomics, BGI Research, Shenzhen, China. [4]Institute of Intelligent Medical Research (IIMR), BGI Genomics, Shenzhen, China. [5]Department of Radiation Sciences, Oncology, Umeå University, Umeå, Sweden. [6]Department of Medical Biosciences, Pathology, Umeå University, Umeå, Sweden. [7]Department of Surgical Sciences, Uppsala University, Akademiska sjukhuset, Uppsala, Sweden. [8]Department of Neuroscience, Karolinska Institutet, Stockholm, Sweden. [9]Science for Life Laboratory, Department of Protein Science, KTH-Royal Institute of Technology, Stockholm, Sweden. [10]Present address: Department of Molecular Oncology, Institute for Cancer Research, Oslo University Hospital, Oslo, Norway. [11]These authors contributed equally: Luís Nunes, Fuqiang Li, Meizhen Wu, Tian Luo. [12]These authors jointly supervised this work: Kui Wu, Bengt Glimelius, Cong Lin, Tobias Sjöblom. ✉e-mail: wukui@genomics.cn; bengt.glimelius@igp.uu.se; lincong@genomics.cn; tobias.sjoblom@igp.uu.se

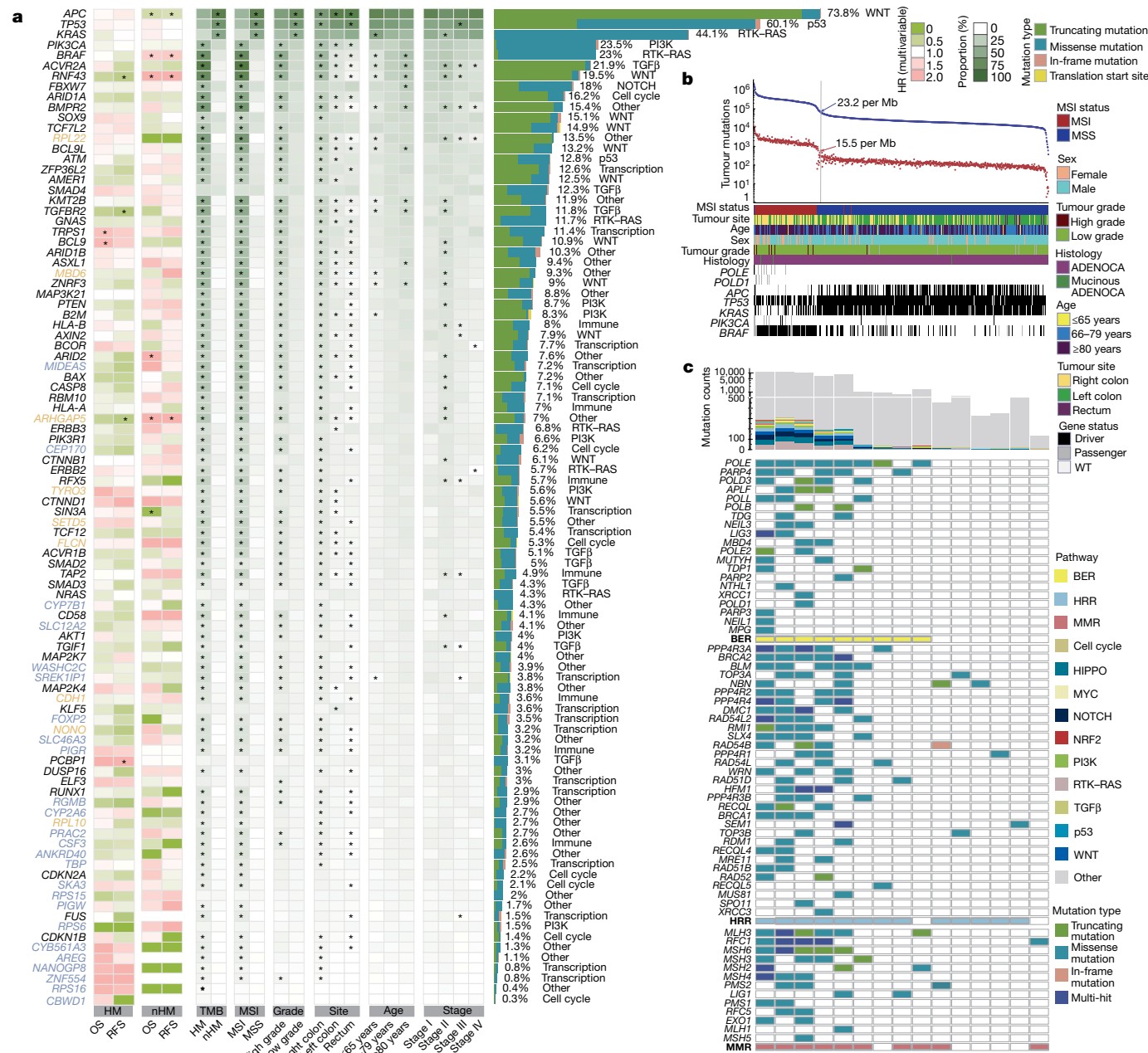

**Fig. 1 | Somatic mutation analysis of 1,063 CRC genomes identifies 96 driver genes.** Somatic mutations were called (Methods) and significantly mutated genes were identified using dNdScv. **a**, The 96 genes mutated at a significant level in this cohort. The association of driver genes with survival (HR) is shown for HM and nHM tumours (multivariable Cox regression). The association of driver genes with clinical and genomic features is shown by the proportion of tumours affected (Fisher's exact test). *FDR-adjusted *P* < 0.05. The mutation type and prevalence is indicated on the right, including a description of the affected pathway. Colour keys for HR for OS and RFS, and for genomic feature proportions are shown on the far right. Genes that were not previously designated as drivers in CRC (orange) or in any cancer type (blue) are indicated.

**b**, The prevalence of total (blue) and non-synonymous (red) mutations in each tumour. Cut-offs for HM and nHM are indicated (grey line). The clinical features and mutation status for selected genes are shown at the bottom. Mutations that are considered to be drivers are either probably oncogenic mutations annotated by OncoKB or hotspots catalogued by Cancer Hotspots. **c**, DNA damage response (DDR) gene mutations in the 15 out of 21 HM tumour cases that were MSS. Not all DNA damage response genes included here can be interpreted as the direct cause of the high TMB in these MSS samples. Top, the total non-synonymous mutation counts for each sample are coloured by the affected oncogenic pathways. ADENOCA, adenocarcinoma; BER, base excision repair; HRR, homologous recombination repair; MMR, mismatch repair.

Of all of the patients, 126 (12%) had been pretreated before the tumour specimens were obtained at surgery, and 92 of these samples were rectal cancers treated with either chemoradiotherapy or radiotherapy before surgery.

In total, 96 mutated driver genes were identified, and 1,056 (99%) of the tumours had a somatic mutation in at least one of these (Fig. 1a and Supplementary Table 3). On the basis of the total mutation count,

242 (23%) tumours were hypermutated (HM) with >23.16 mutations per megabase (Fig. 1b and Supplementary Table 4). Compared with non-hypermutated (nHM) cases, the HM cases were older (median age, 76 versus 71 years), female and had right-sided, mucinous, high-grade (poorly differentiated or undifferentiated) and stage II tumours more often[14] (Supplementary Table 5). The MSI criteria (MSIsensor score ≥ 3.5; Methods) were fulfilled in 223 (21%) patients, of which only

2 were nHM. This population-based cohort, representative of Swedish patients with CRC (Supplementary Table 6), contrasts with cohorts with younger, fitter patients in whom MSI prevalence is lower[15]. In total, 15 HM tumours were MSS with *POLE* or other DNA-damage-repair gene mutations (Fig. 1c) and 6 were MSS with high non-coding tumour mutation burden (TMB) but no repair gene mutations. Not all repair gene mutations in the HM MSS tumours were drivers; thus, it may not fully explain their high TMB. Analyses in three groups (HM, nHM and the entire cohort of tumours) identified 96 unique driver genes, along with 13 additional subtype-specific drivers (Supplementary Table 3). In the HM tumours, genes mutated in more than 20% of the cases belonged to the TGFβ–BMP, WNT, RTK–RAS, ribosomal proteins, epigenetic regulation, PI3K, SCF complex, p53 and immune system pathways, and the most frequently recurring mutations were in *ACVR2A* (p.K437Rfs*5; 78%), *BRAF* (p.V600E; 65%) and *RNF43* (p.G659Vfs*41; 57%). Correspondingly, the WNT, p53, RTK–RAS, PI3K, SCF complex and TGFβ pathways had genes mutated in more than 10% of nHM tumours (Fig. 1a and Supplementary Table 3). The most common hotspot mutations in nHM tumours were *KRAS* p.G12D (15%) and p.G12V (11%).

Of the 96 driver genes, the 24 that had not previously been designated as drivers in any cancer[16,17] were linked to BMP (*RGMB*) and EGFR (*AREG*) signalling, cell cycle (*CEP170* and *SKA3*), immune system (*PIGR* and *CSF3*), ion transport (*SLC12A2* and *CYB561A3*), metabolism (*PIGW*, *CYP2A6* and *CYP7B1*), mRNA splicing (*SREK1IP1*), protein transport (*WASHC2C* and *SLC46A3*), transcriptional regulation (*FOXP2*, *NANOGP8*, *TBP* and *ZNF554*), ribosomal proteins (*RPS15*, *RPS16* and *RPS6*) and other pathways (*CBWD1*, *PRAC2* and *ANKRD40*). Nine drivers that had not previously been observed in CRC[3–5,8,10,11,18–21] were linked to the immune system (*CDH1*), histone modification (*SETD5*), transcription regulators (*MIDEAS* and *NONO*), PI3K signalling (*TYRO3*), cellular response (*FLCN*), ribosomal proteins (*RPL10* and *RPL22*) and UCH proteinase (*MBD6*).

Two distinct patterns of RTK–WNT pathway co-mutations—(1) *KRAS*, *APC* and *AMER1*, and (2) *BRAF* and *RNF43*—were identified (Extended Data Fig. 1a and Supplementary Table 7). For the *KRAS*, *APC* and *AMER1* group, the nHM tumours had co-occurring *PIK3CA* (FDR-adjusted $P = 1.98 \times 10^{-5}$) and *TCF7L2* (FDR-adjusted $P = 4.57 \times 10^{-4}$) and mutually exclusive *TP53* (FDR-adjusted $P = 1.06 \times 10^{-7}$) and *NRAS* (FDR-adjusted $P = 1.76 \times 10^{-6}$) mutations. In the *BRAF* and *RNF43* tumours, co-occurring mutations were observed in *ACVR2A* (FDR-adjusted $P = 0.06$) in HM tumours, and *AKT1* (FDR-adjusted $P = 0.03$) and *TYRO3* (FDR-adjusted $P = 0.08$) in the nHM tumours (Extended Data Fig. 1b). In the TGFβ pathway, co-occurring mutations were found in *SMAD2* and *SMAD3* (FDR-adjusted $P = 1.03 \times 10^{-10}$) in nHM tumours, whereas *TGIF1* co-occurred with *PIK3CA* (FDR-adjusted $P = 0.09$) in the HM cases. The HM tumours had mutually exclusive mutations in *B2M* and *HLA-A* (FDR-adjusted $P = 0.07$)[22], and co-occurring mutations in *KMT2B* and *CD58* (FDR-adjusted $P = 0.01$) and *ERBB3* (FDR-adjusted $P = 0.09$). In all, we identified 33 additional CRC drivers along with previously unidentified co-mutation patterns within and across CRC pathways.

## SVs and timing analyses

To encompass all types of genomic events in the progression of CRC, we compiled copy-number variants (CNVs) and structural variants (SVs)[23]. The most common chromosome arm aberrations were gains of 7p and 20q in around 50% of tumours, and loss of heterozygosity (LOH) of 17p, 18p and 18q in more than 40% (Fig. 2a). Novel focal CNVs identified in nHM tumours included deletions of 15q24.3 (25%) containing *MIR3713*, 22q12.3 (24%) and 8p11.22 (19%) containing ADAM-family protease genes. The frequency of gene CNVs was higher in nHM compared with in HM tumours, affecting an average of 55% versus 11% of the driver genes. The drivers most frequently affected by CNVs were *GNAS* and *ASXL1*, for which 82% and 81% of nHM tumours had gains and/ or amplifications, whereas *SMAD4* (79%), *SMAD2* (77%) and *TP53* (76%)

had more deletions and LOH. In HM tumours, the antigen-presenting genes *HLA-B* (26%), *HLA-A* (25%) and *TAP2* (24%) had the highest LOH frequency, whereas *TRPS1* (26%), *ACVR1B* (22%), *CYP7B1* (22%), *MBD6* (22%) and *ERBB3* (22%) had the highest frequency of gains (Extended Data Fig. 2a and Supplementary Table 8). Deletions were the most common SV, primarily in high-grade (FDR = 0.011) and less in stage I (FDR = 0.021) tumours. Translocations were more common in older patients (FDR-adjusted $P = 0.004$), and in HM (FDR-adjusted $P = 0.014$), MSI (FDR-adjusted $P = 0.003$) and high-grade (FDR-adjusted $P = 0.003$) tumours. By contrast, inversions and tandem duplications were less common in HM (FDR-adjusted $P = 4.69 \times 10^{-17}$ and 0.0091), MSI (FDR-adjusted $P = 2.75 \times 10^{-16}$ and 0.029) and right-sided (FDR-adjusted $P = 6.45 \times 10^{-8}$ and 0.005; Fig. 2b) tumours. Half of driver gene SVs were deletions, most frequently affecting tumour suppressor genes including *RUNX1* ($n = 38$), *PTEN* ($n = 33$) and *SMAD3* ($n = 30$). The most frequently affected DNA repair gene was *RAD51B* ($n = 33$; Extended Data Fig. 2b). Extrachromosomal DNA (ecDNA) was observed in 250 tumours (24%), of which 91% were nHM ($P = 2.9 \times 10^{-9}$). Circular amplicons were found in 87 (35%) of ecDNA+ cases, and the oncogenes most frequently contained were *ERBB2* ($n = 9$; 10%), *FLT3* ($n = 7$; 8%), *CDX2* ($n = 7$; 8%), *CDK12* ($n = 5$; 6%) and *MYC* ($n = 5$; 6%; Supplementary Fig. 1a). Tumour ecDNA correlated with shorter survival in a pan-cancer study[24], but no ecDNA-type-dependent differences in overall survival (OS) or recurrence-free survival (RFS) were observed here (Supplementary Fig. 1b).

The sequence of genomic events during CRC evolution has not previously been determined in a large set of nHM tumours[25]. Here the earliest events were somatic mutations in *APC*, *TP53*, *KRAS*, *BRAF* and *ZFP36L2*, followed by *TCF7L2*, *FBXW7*, *BCL9L* and *SOX9* and loss of chromosomes 17p and 18. Among them, (1) *TP53* and 17p loss are known early mutations that are frequently found in multiple cancers; (2) *APC*, *KRAS*, *BRAF* and *TP53* mutations drive CRC development; and (3) *ZFP36L2*, *TCF7L2*, *BCL9L* and *SOX9* are previously unknown early events in cancer[26]. Late or subclonal events included whole-genome duplication, gains of 1q, 6p, 9p, 12, 16p, 17q and 19q, and mutations in *TRPS1*, *GNAS* and *CEP170* (Fig. 2c and Supplementary Table 9). These findings inform strategies for early detection and events of potential relevance for CRC invasion and metastasis.

## Mutational signatures

Mutational signatures in CRC have been linked to ageing, mismatch repair (MMR) deficiency, polymerase proofreading, colibactin exposure and unknown aetiologies[27]. Here a de novo analysis identified 27 single-base substitution (SBS; Supplementary Fig. 2), 8 doublet-base substitution (DBS; Supplementary Fig. 3) and 11 small insertion and deletion (ID; Supplementary Fig. 4) signatures (Extended Data Fig. 3a–c and Supplementary Tables 10 and 11). Of the 27 SBS signatures, 25 decomposed to 32 COSMIC SBS signatures[27] (Extended Data Fig. 3a and Supplementary Tables 12 and 13). A new signature, termed SBS-CRC1, was found in 17 tumours, all MSI or *POLE* mutant, and correlated with the defective DNA MMR SBS15 signature ($r = 0.40$, FDR-adjusted $P = 7.82 \times 10^{-40}$; Supplementary Table 14). Another new signature, SBS-CRC2, was observed in 17 cases with low-grade and MSS tumours, and correlated with the signature of unknown aetiology SBS128[28] (cosine similarity = 0.94; Supplementary Table 15). Notably, HM tumours with the DNA MMR SBS44 signature were primarily right colon (85% versus 70%, $P = 0.0064$), *BRAF* V600E mutated (70% versus 45%, $P = 0.0015$), less frequently stage IV (4% versus 15%, $P = 0.0386$) and had longer OS (multivariable hazard ratio (HR) = 0.558, 95% confidence interval (CI) = 0.319–0.974; Extended Data Fig. 3d). Of the 8 DBS signatures, 3 could be decomposed to 6 COSMIC DBS signatures (Extended Data Fig. 3b and Supplementary Tables 12 and 13). The new DBS-CRC3 signature had the highest somatic mutation density and occurred in 98% of MSI cases. The defective DNA MMR signatures SBS15

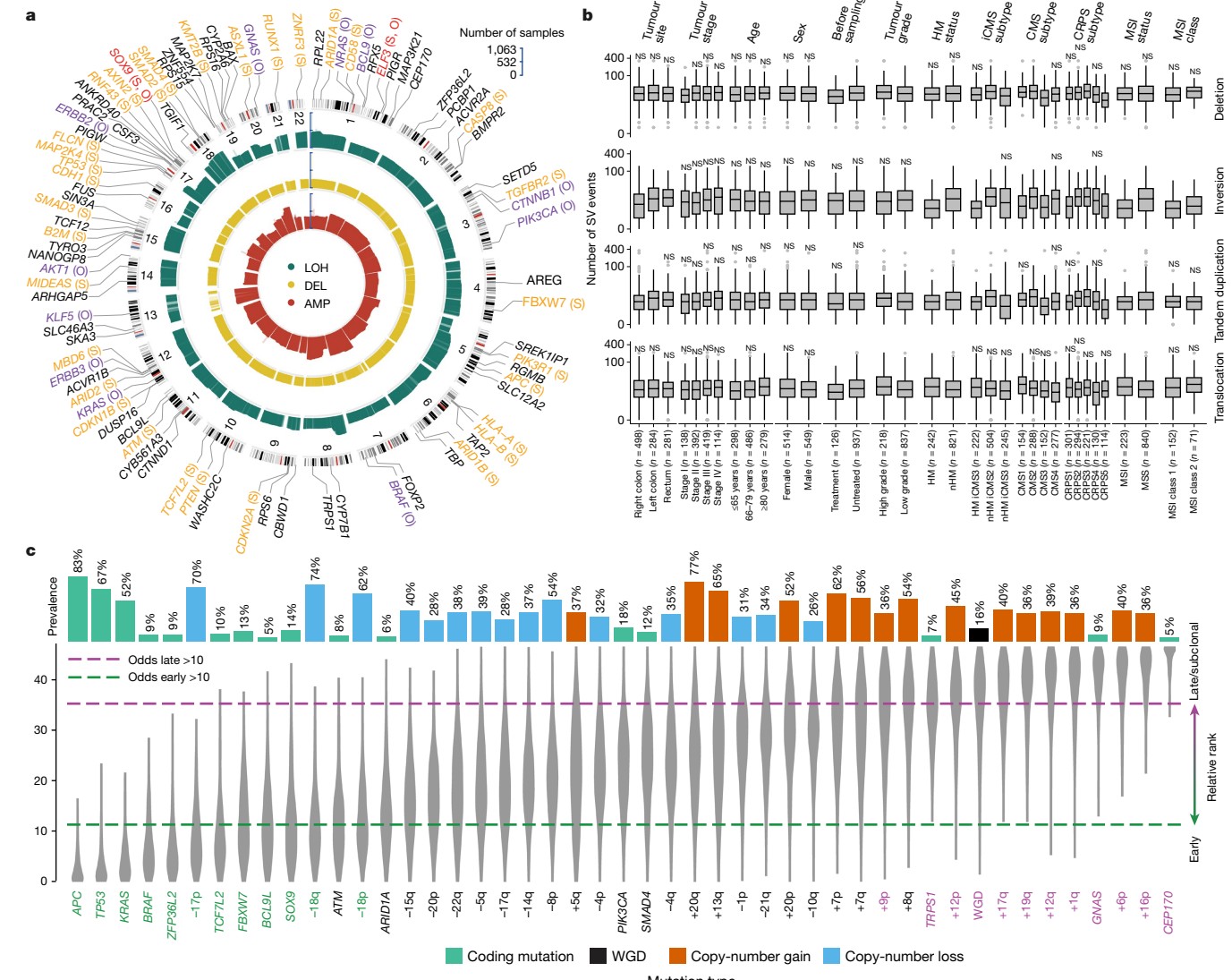

**Fig. 2 | Structural variation and relative timing of somatic events in CRC.**
**a**, Gene CNVs in driver genes displayed by type: LOH (green), deletion (yellow) and amplification (red). The bar height is proportional to the fraction of tumours with respective alteration. The 91 autosomal driver genes are indicated as oncogenes (O; purple), tumour suppressor genes (S; orange), both (S, O; red) or genes with an unknown role (black), and are displayed by genomic location. **b**, The SV landscape for deletions, inversions, tandem duplications and translocations displayed by clinical, genomic and transcriptomic features. The boxes represent the interquartile ranges (IQRs) between the first and third

quartiles, the centre line represents the median, and the whiskers extend to 1.5× the IQR from the top and bottom of the box. Statistical analysis was performed using two-sided Wilcoxon rank-sum tests; *FDR-adjusted $P < 0.05$, **FDR-adjusted $P < 0.01$, ***FDR-adjusted $P < 0.001$, ****FDR-adjusted $P < 0.0001$. **c**, The prevalence and relative timing of driver gene mutations and SVs in 801 nHM CRC tumours by PhylogicNDT. Early/clonal (green), intermediate (black) and late/subclonal (purple) alterations are indicated. WGD, whole-genome duplication.

and SBS44 strongly correlated with DBS-CRC3 ($r = 0.61$ and $r = 0.84$, FDR-adjusted $P = 3.88 \times 10^{-108}$ and $P = 8.11 \times 10^{-28}$), which was similar to the MMR deficiency signature DBS19 described previously[28] (cosine similarity = 0.87; Supplementary Table 15). The signatures SBS10a and SBS10b, associated with MSS *POLE* mutated tumours, co-occurred with SBS28[29] ($r = 0.48$ and $0.61$, FDR-adjusted $P = 5.13 \times 10^{-62}$ and $5.04 \times 10^{-107}$) and the new DBS-CRC5 signature ($r = 0.43$, FDR-adjusted $P = 8.21 \times 10^{-47}$; Supplementary Table 14). Lastly, from the 11 ID signatures, 9 decomposed to 9 COSMIC ID signatures, of which the most frequent, ID1 (in 87%) and ID2 (in 98%), are related to DNA slippage during replication[27] (Extended Data Fig. 3c and Supplementary Tables 12 and 13). Notably, the new ID-CRC1 signature had the highest somatic mutation density (>10 mutations per Mb), and 89% of cases with ID-CRC1 also had the defective DNA MMR signature SBS44. Together, 47 known and

9 previously unknown (Extended Data Fig. 3e) mutational signatures were identified, of which SBS28 and DBS-CRC5 were associated with *POLE* mutant MSS CRC, the SBS-CRC1, DBS-CRC3 and ID-CRC1 signatures with MMR, and the DNA MMR SBS44 signature in HM tumours with longer OS.

## Mitochondrial genomes

High median copy numbers and enrichment of truncating mutations characterize CRC mitochondrial DNA (mtDNA)[30]. We identified 3,982 single-nucleotide variants (SNVs) and 949 indel mutations in mtDNA in 1,027 (97%) tumours (Supplementary Table 16). The mtDNA mutations were most frequent in the non-coding promoter D-loop (48%) and in the complex I genes *ND5* (41%) and *ND4* (30%; Extended Data

Fig. 4a). Truncating mutations were enriched in *ND5* and *ND4*, representing 35% and 29% of mutations. Like in other cancer types, missense mtDNA mutations were more frequently near-homoplasmic (variant allele frequency (VAF) > 60%) compared with silent and truncating mutations, and their overall dN/dS ratio was close to 1 at different VAFs (Extended Data Fig. 4b,c). Truncating mutations with VAF > 60% occurred in 6.6% of tumours, compared with <3% in other cancers[30], suggesting that mitochondrial dysregulation is important for CRC tumorigenesis. While HM status did not correlate with mtDNA mutation counts, age at diagnosis did (Extended Data Fig. 4d). Co-occurrence but not mutual exclusivity was observed between mtDNA mutations (Extended Data Fig. 4e). Mitochondrial genome copy number (mtDNA-CN) was lower in right colon, high-grade and HM tumours (Supplementary Fig. 5a). When divided into low ($n$ = 127) and high ($n$ = 912) tumour mtDNA-CN groups, there was a trend toward longer OS in high-mtDNA-CN cases (Supplementary Fig. 5b–d). The mtDNA-CN correlated positively with clock-like (SBS1 and SBS5) and ROS (SBS18) SBS signatures but negatively with most MMR signatures (Supplementary Fig. 5e–g).

## Prognostic alterations

Compared with cohorts from clinical trials, referral hospitals or actively treated patients, patient age was higher (median age, 72 versus 54–68 years), right-sided tumours were more common (47% versus 30–39%) and the fraction of MSI cases was higher (21% versus 8–12%) in this cohort, leading to different prognostic cohort features[4,8,11]. When compared with all surgically resected CRCs in Sweden, rectal, stage I and stage IV tumours were slightly under-represented, but OS was similar (Supplementary Table 6). In all, the cohort is representative of the resected Swedish CRC patient population and, as such, of Western real-life populations[21,31]. The MSS cases had shorter RFS ($P$ = 0.048) compared with MSI cases, while non-pretreated stage IV MSI cases had shorter OS ($P$ = 0.004) compared with their MSS counterparts. The worst OS and RFS were observed in patients aged >80 years, and for high-grade and more-advanced-stage tumours. For non-pretreated stage I–III cases, tumour location in the rectum or left colon was correlated with longer OS ($P$ = 0.032) but not RFS ($P$ = 0.365). Right-sided colon tumours correlated with shorter OS only for stage IV cases receiving first-line chemotherapy without metastasectomy ($P$ = 0.0003; Supplementary Table 17).

Based on 994 (94%) patients that had 5-year survival data, of which 219 (22%) had HM CRC, we identified mutated driver genes associated with OS or RFS (Supplementary Table 18). In the nHM group, *APC* mutations correlated with longer OS (HR = 0.61, 95% CI = 0.46–0.82) and RFS (HR = 0.68, 95% CI = 0.49–0.94)[32], *MT-CYB* mutations with longer OS (HR = 0.59, 95% CI = 0.43–0.82) and RFS (HR = 0.67, 95% CI = 0.47–0.94; Supplementary Table 19), and *SIN3A* mutations with longer OS (HR = 0.11, 95% CI = 0.02–0.79). Mutations in *ARHGAP5*, *BRAF*[33] and *RNF43*[34] correlated with shorter OS (HR = 4.09, 1.58 and 2.30; 95% CI = 1.79–9.33, 1.10–2.27 and 1.37–3.84) and RFS (HR = 2.68, 1.65 and 2.03; 95% CI = 1.09–6.59, 1.08–2.53 and 1.07–3.85), and *ARID2* mutations with shorter OS (HR = 2.07, 95% CI = 1.17–3.65). By contrast, HM cases with *ARHGAP5*, *RNF43* and *TGFBR2* mutations had longer RFS (HR = 0.40, 0.43 and 0.34; 95% CI = 0.19–0.85, 0.23–0.78 and 0.19–0.62), whereas *BCL9* and *TRPS1* correlated with shorter OS (HR = 1.89 and 1.77; 95% CI = 1.11–3.20 and 1.03–3.03), and *PCBP1* with shorter RFS (HR = 4.22, 95% CI = 1.40–12.74).

We analysed candidate *cis*-regulatory elements (cCREs) in nHM tumours and identified 7 proximal enhancer-like, 1 promoter-like, 7 DNase-only and 11 CTCF-only elements along with 34 differentially expressed linked genes (Supplementary Table 20). Of the genes with deregulated expression, *ID2* and *HS3ST1* had regulatory element mutations linked to shorter OS (HR = 3.49 and 2.94, 95% CI = 1.62–7.53 and 1.29–6.70; Supplementary Table 21) and *DAPK1* with shorter RFS

(HR = 2.68, 95% CI = 1.25–5.74). Targeting *ID2* reduced CRC growth in vivo and its expression was increased by WNT signalling under hypoxia[35,36], and *DAPK1* loss was linked to invasiveness of CRC cells[37], supporting their roles in CRC pathogenesis.

Prognostic CNVs in nHM tumours included known events such as amplification of 20q11.1, 20q11.21 and 20q13.33, and loss of 16p13.3, along with unknown events in which amplifications correlated with longer survival, and losses with shorter survival. For the HM tumours, we identified five prognostic CNVs linked to shorter survival including the known −4q22.1 event (Supplementary Table 22). Patients with nHM tumours with *SMAD4* deletion had shorter RFS (HR = 2.13, 95% CI = 1.04–4.34), and those with *TCF7L2* translocation had shorter OS (HR = 4.82, 95% CI = 2.10–11.03) and RFS (HR = 7.50, 95% CI = 2.72–20.69; Supplementary Table 23). Together, we observed associations with prognosis for mutations in 12 known cancer genes, 21 CNVs, 1 mitochondrial gene and 3 regulatory elements (Table 1).

## Expression of drivers and fusion genes

High-quality genome and transcriptome sequences from the same large set of tumours enable integrated analyses of gene mutations and gene expression levels. Tumour mutations in RTK–RAS, PI3K, p53 and TGFβ pathway genes *EGFR*, *KRAS*, *PIK3CA*, *CDKN2A*, *TGFBR1* and *ACVR2A* were associated with increased gene expression, while mutations in *APC*, *PTEN* and *TP53* all had decreased expression in nHM and HM tumours (Fig. 3 and Supplementary Table 24). Although nonsense-mediated decay can complicate interpretation of differential gene expression levels[38], among the WNT pathway driver genes, *RNF43*, *AXIN2*, *SOX9*, *ZNRF3*, *CTNNB1* and *AMER1* had 55–334% higher expression in tumours while *TCF7L2*, *APC* and *CTNND1* had 15–24% higher expression in unaffected control colorectal tissue (Supplementary Table 25). Tumours with *SOX9* or *TCF7L2* mutations had increased expression of the respective genes, while other mutant WNT pathway drivers had reduced tumour expression (Extended Data Fig. 5). Notably, tumours with RTK/WNT pathway *BRAF* and *RNF43* co-mutations had higher expression of *BRAF* and lower expression of *RNF43* compared with wild-type tumours and tumours carrying mutations of only one gene (Extended Data Fig. 1a). Decreased expression coupled to mutation characterized several genes related to antigen presentation (*HLA-A*, *B2M* and *CDH1*), transcription regulation (*ASXL1* and *NONO*), apoptosis (*BAX*), histone modification (*KMT2B*) and ribosomal functions (*PPL22*) (Extended Data Fig. 5 and Supplementary Table 25).

A total of 621 fusion transcripts were expressed in 338 nHM (41%) and 78 HM (32%) tumours, 17 of which were recurrent (Supplementary Fig. 6a). The most frequently fused genes were *PTPRK* ($n$ = 27 tumours), *RSPO3* ($n$ = 25), *SEPTIN14* ($n$ = 24), *FBXO25* ($n$ = 24) and *FBRSL1* ($n$ = 19; Supplementary Fig. 6b). Of the fusions, 15 were known CRC drivers, including *NTRK*, *BRAF* and *ERBB2* fusions, the *PTPRK*–*RSPO3* fusion shown to promote differentiation and loss of stemness[39,40], and the uncharacterized *FBXO25*–*SEPTIN14* fusion (Supplementary Fig. 6c,d).

## Prognostic gene expression signature

Mutational and transcriptional data can be used to develop subtyping classifiers in which the contributions of underlying genomic events are defined. The Consensus Molecular Subtypes (CMS) classification system is the state-of-the-art gene expression-based classification of CRC[41] but, as most CRCs are composed of several CMS subtypes when deconvoluted[42], a refined classification at the single-cell resolution (iCMS) has been proposed[7]. As CMS is based on 18 datasets generated using different technologies, we examined whether unsupervised de novo classification of the cases here would recapitulate CMS or iCMS. In this new classification, termed CRC prognostic subtypes (CRPSs), 97% of CMS1 tumours were classified as CRPS1, CMS2 tumours were distributed between CRPS2 (39%) and CRPS3 (59%), 68% of CMS3

## Table 1 | Prognostic genomic features by hypermutation status

| Feature | nHM | | HM | |
|---|---|---|---|---|
| | **Longer survival** | **Shorter survival** | **Longer survival** | **Shorter survival** |
| Coding driver gene mutation | *APC*[OS,RFS] *SIN3A*[OS] | *ARHGAP5*[OS,RFS] *ARID2*[OS] *BRAF*[OS,RFS] *RNF43*[OS,RFS] | *ARHGAP5*[RFS] *RNF43*[RFS] *TGFBR2*[RFS] | *BCL9*[OS] *PCBP1*[RFS] *TRPS1*[OS] |
| Mitochondrial gene mutation | *MT-CYB*[OS,RFS] | – | – | – |
| Non-coding driver gene mutation | – | *ID2*[OS] *HS3ST1*[OS] *DAPK1*[RFS] | – | – |
| CNV | +8p11.1[OS] +20q11.1[OS,RFS] +20q11.21[OS] +20q13.12[OS] +20q13.33[OS] | −4q34.1[OS] −8p23.1[OS] −11p15.5[RFS] −12q24.33[RFS] −16p13.3[RFS] −17p12[RFS] −17p13.3[RFS] −17q12[OS] −17q21.2[OS] −17q21.31[OS] −17q25.3[OS,RFS] −21p12[OS] | – | −3p21.31[RFS] −4q22.1[RFS] −11p15.5[OS,RFS] −15q26.3[RFS] +19q11[OS] |
| SV | – | *SMAD4*[RFS] *TCF7L2*[OS,RFS] | – | – |
| Mutational signature | – | – | SBS44[OS] | – |

Prognostic features identified by multivariable Cox with adjustment for tumour site, pretreatment status, tumour stage, age group and tumour grade. OS, overall survival for stages I–IV. RFS, recurrence-free survival for stages I–III.

tumours were classified as CRPS5, and CMS4 tumours were distributed between CRPS2 (38%) and CRPS4 (45%; Fig. 4a and Supplementary Fig. 7). Importantly, CRPS assigned all but 3 tumours, while 192 (18%) remained unclassified by CMS, most of which were assigned to CRPS1 and CRPS2. The CRPS1 group contained 88% of the HM cases, whereas only 56% were classified as CMS1. Accordingly, CRPS1 tumours most often occurred in right colon (79%), in older (median age, 76 years) and female (61%) patients and had the highest prevalence of somatic SNVs and the lowest of CNVs (Supplementary Table 1 and Extended Data Fig. 6a–c). The CRPS2 and CRPS3 subtypes were distributed equally between anatomical locations and had low frequencies of *BRAF* mutations (Extended Data Fig. 6a). The CRPS4 tumours were often rectal (47%), and exhibited stromal, TGFβ and WNT pathway activation, consistent with CMS4 (Fig. 4b and Supplementary Table 26). Finally, CRPS5 tumours were often from the right colon (46%), displayed WNT signalling repression (Supplementary Table 26) and had the highest prevalence of *KRAS*, *PIK3CA* and *FBXW7* mutations, but fewer *TP53* mutations and CNVs compared with CRPS2–CRPS4 (Fig. 4b and Extended Data Fig. 6a,b). The distribution of reclustered CRPS cases was robust to removal of stage IV and pretreated cases (Supplementary Fig. 8a–d). The CRPS subtypes were prognostic for OS in stages I–IV ($P = 0.01$), RFS in stages I–III ($P = 0.025$; Fig. 4c) and for survival after recurrence ($P = 0.034$; Supplementary Fig. 8e), with CRPS2 and CRPS3 associated with the longest OS and RFS, CRPS4 with shortest OS and RFS, CRPS5 with shorter RFS, and CRPS1 with the worst survival after recurrence. The CMS4 cases assigned to CRPS2 had a longer OS compared with those assigned to CRPS4 (Fig. 4d), which may reflect that the density of fibroblasts, macrophages and dendritic cells in CRPS2 is between those of CRPS3 and CRPS4[7] (Fig. 4b). To advance the understanding of nHM tumours, we identified prognostic features by CRPS, CMS and iCMS subtypes. Unfavourable-prognosis nHM iCMS3 tumours[7] were primarily CRPS1 and CRPS4–CRPS5, displaying similarities in mutation profiles and clinical outcomes (Extended Data Fig. 7a). Despite their lower overall CNV load compared with iCMS2 tumours, several late CNV deletions correlated with shorter survival (Supplementary Table 27), while *BRAF* and *RNF43* mutations were early events (Supplementary

Table 9). Most of the nHM iCMS2 tumours were CRPS2–CRPS3. Amplification of 20q11 correlated with longer survival, consistent with being the major contributing feature for CRPS2[7,43] (Supplementary Fig. 9b and Supplementary Table 22). Within the nHM CMS4 tumours, those classified as CRPS4 divided between iCMS2 and iCMS3, while those classified as CRPS2 were primarily iCMS2. This could explain the separation of these tumours from the wider CMS4 category into the CRPS2 and CRPS4 subtypes, with shorter survival for the CMS4 cases in the latter (Fig. 4d). The relatively poor prognosis of CRPS4 may stem from the majority of tumours displaying iCMS3 characteristics[7]. For external validation, we developed a ResNet50 CRPS-based classifier (Supplementary Fig. 9a) and analysed 2,832 cases from 10 cohorts yielding accuracy, precision, recall and $F_1$ score of >85%. The prognostic ability of CRPS and the correspondence between CMS and CRPS was recapitulated (Extended Data Fig. 7b–d), with CRPS2 having the longest OS and CRPS4 having the shortest ($P = 0.013$ for all CRPS). Pathway features of CRPS subtypes were reproduced in the validation cohort (Extended Data Fig. 7e). Together, the CRPS outperforms CMS for prognosis, assigns a very high proportion of tumours to subtypes and provides deeper insights into CRC subtypes when combined with single-cell signatures.

## Tumour hypoxia

Among 27 tumour types, CRC ranked the third most hypoxic[44]. To delineate links between genomic and transcriptomic alterations and tumour oxygenation, we analysed the transcriptomes using the Buffa hypoxia signature[45]. Tumours consistently had elevated hypoxia scores compared with unaffected control tissue (median, 1 versus −20); right colon tumours had the highest, followed by left colon and rectum tumours (median, 7 versus 1 versus −5; Extended Data Fig. 8a). Furthermore, tumours in female individuals and those of high grade, HM and MSI had elevated levels of hypoxia. Considering all tumours, the strongest associations were SBS1, SBS5 and ID1, prevalent in tumours with low hypoxia[44] (Extended Data Fig. 8b and Supplementary Table 28). The MMR-related signatures SBS44, SBS26,

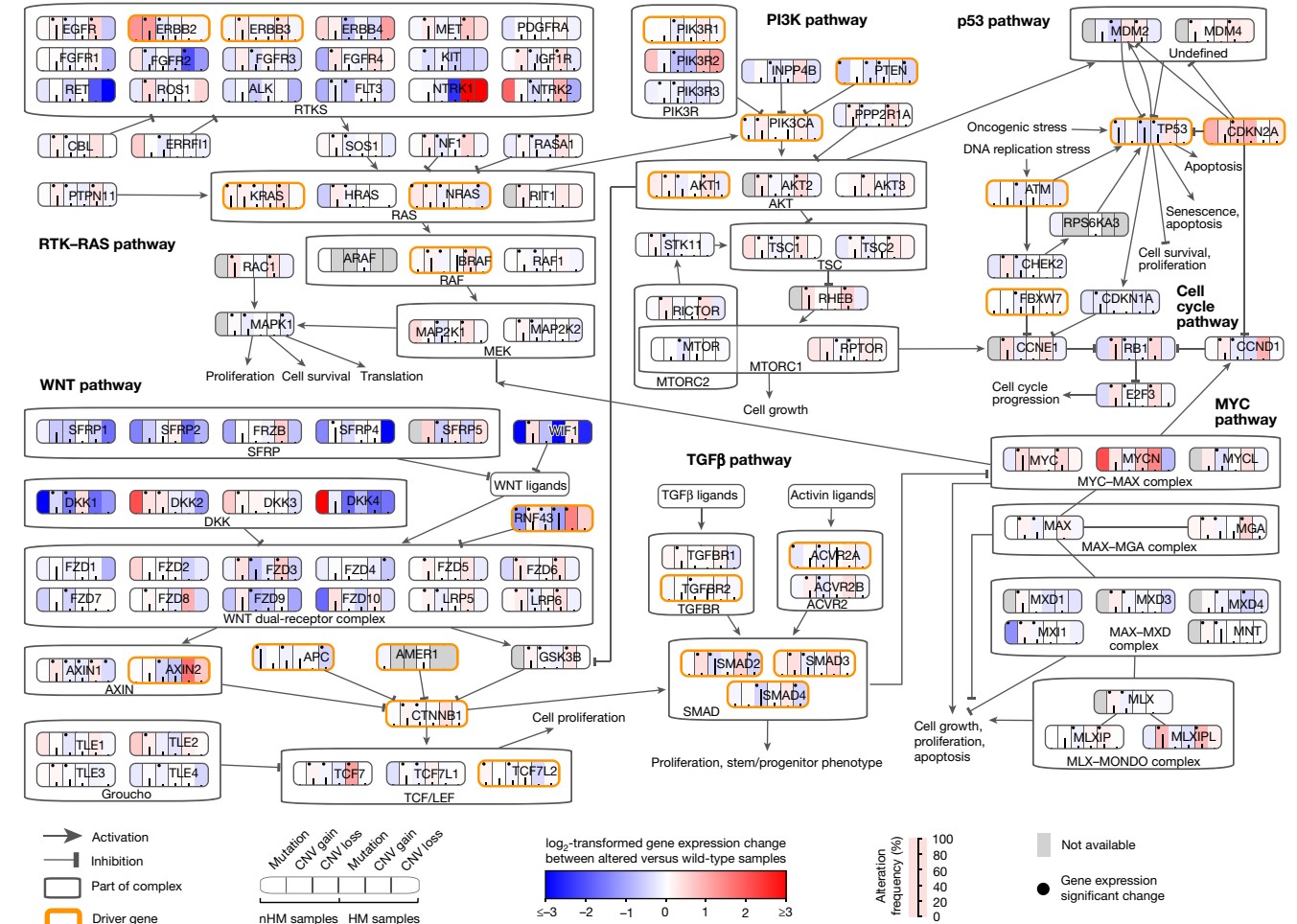

**Fig. 3 | Integrative analysis of somatic alterations and gene expression levels in CRC signalling pathways.** The frequencies of somatic alterations, including mutations and copy-number (CNV) loss and gain for each gene in nHM and HM tumours. Red (log$_2$[fold change (FC)] > 0) and blue (log$_2$[FC] < 0) colour intensities represent the log-transformed FC between mutated and wild-type tumours by type of somatic alteration (mutation, CNV gain and CNV loss for nHM and HM samples). Somatic alteration frequencies are indicated by the black line in each column. Black dots show gene expression changes with FDR-adjusted $P$ < 0.05 (two-sided Wilcoxon rank-sum test). Driver genes are marked by orange borders.

SBS14, SBS-CRC1 and DBS-CRC3 correlated with high hypoxia. By contrast, SBS18, related to damage by ROS, and ID18, related to colibactin exposure, were inversely correlated. The two ID mutational signatures associated with slippage during DNA replication were both correlated with hypoxia, with ID1 inversely correlated and ID2 correlated[44]. In nHM tumours, most driver genes, particularly *SMAD4*, *SMAD2* and *FBXW7*, correlated with high hypoxia. In HM tumours only *RGMB*, *SMAD2*, *AREG* and *RFX5* mutations correlated with low hypoxia (Extended Data Fig. 9a and Supplementary Table 28). Most SV types were associated with high hypoxia in nHM tumours[44], but only deletions were associated with high hypoxia in HM tumours (Extended Data Figs. 8c and 9b). Increased hypoxia was associated with a higher number of clonal, but not subclonal, mutations in nHM tumours[44] (Extended Data Fig. 9c). High TMB was associated with hypoxia when considering all tumours. Impaired mitochondrial activity and abnormal mtDNA-CN characterized hypoxic tumours[46], and mtDNA-CN was negatively correlated with hypoxia in both HM and nHM cases (Supplementary Table 28). The CRPS1 and CRPS4 tumours were the most hypoxic, whereas CRPS3 and CRPS5 were the least. No correlation between hypoxia and survival or tumour size determined by magnetic resonance imaging in rectal cancers was observed. In summary, these findings corroborate previous observations in nHM CRC and provide insights into hypoxia in HM CRC.

## Tumour microenvironment

The tumour microenvironment was characterized by transcriptome-based prediction of stromal and immune cell populations[47,48]. The CRPS groups displayed differential infiltration of immune cells (Supplementary Fig. 10a,b). CRPS1 was enriched for T cells, B cells, dendritic cells and macrophages, CRPS2 for haematopoietic stem cells, dendritic cells and macrophages, while CRPS3 tumours had low levels of immune cell infiltration but higher levels of megakaryocyte–erythroid progenitor cells (MEPs) and osteoblast-like cells. In CRPS4, fibroblasts, chondrocytes, endothelial cells, haematopoietic stem cells and macrophages were enriched, while epithelial, MEPs and T cells were low. The CRPS5 tumours were characterized by CD4 central memory and effector memory T cells. When stratified by HM and MSI status, nHM tumours had more fibroblasts and haematopoietic stem and granulocyte-monocyte progenitor cells, but less mesenchymal stem and immune cell infiltration compared with HM cases (Supplementary Table 29). In nHM/MSS cases, M2-like macrophages were associated with shorter OS and RFS, whereas T cells, dendritic and eosinophil cells were associated with longer OS and RFS (Supplementary Table 30).

Most MSI CRCs respond to immunotherapy, but 45% do not, motivating a finer-grained subtyping[49]. We divided MSI tumours into two

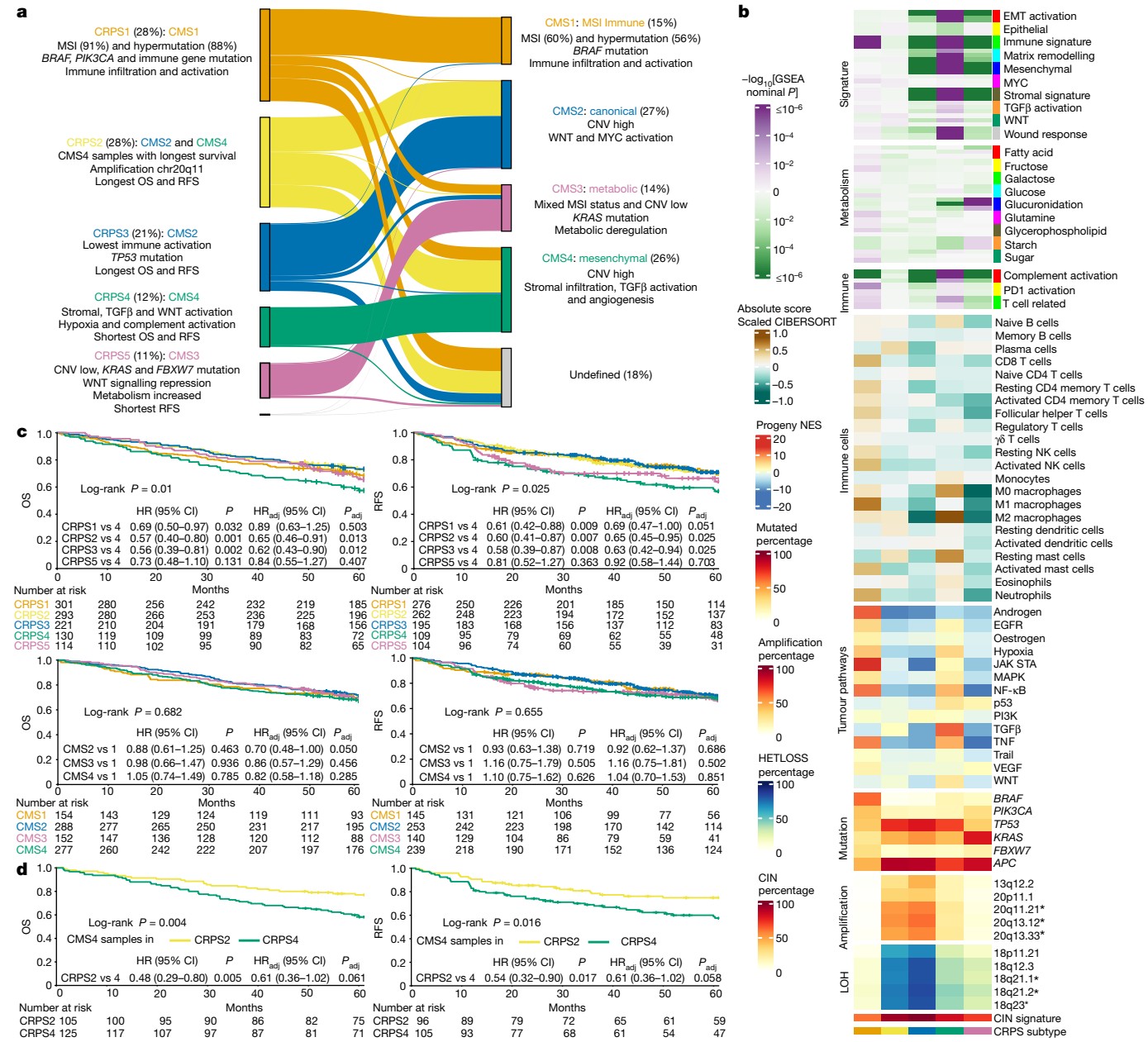

**Fig. 4 | Refined prognostic subtypes derived from 1,063 CRC transcriptomes.** The characteristics of the five distinct CRPSs obtained from unsupervised classification of tumour transcriptome data. **a**, Comparison of CRPS to the CMSs for the same dataset. The proportion of samples assigned to each subtype is shown as the percentage of the total number of tumours. The main molecular and clinical characteristics for each CRPS and CMS subgroup are indicated. **b**, Transcriptomic characteristics of 1,063 samples according to their CRPS classification. Prognostic focal CNV cytobands that are differentially altered in CRPS are indicated by asterisks (P < 0.05, multivariable Cox regression in nHM tumours). **c**, Kaplan–Meier survival curves (log-rank test) for overall (stages I–IV) and recurrence-free (stages I–III) survival in CRPS (top) and CMS (bottom) groups. **d**, Kaplan–Meier survival curves (log-rank test) for CMS4 samples allocated to CRPS2 and CMS4 samples allocated to the CRPS4 group. Adjusted HR (HR$_{adj}$) and P values (P$_{adj}$) or HR and P values were calculated using multivariable Cox regression with or without adjustment for tumour stage. CIN, chromosomal instability.

classes using unsupervised classification (Supplementary Fig. 11), where the first was characterized by lymphocytes and stromal cells, and the second by more abundant MEPs and T helper type 1 cells (Supplementary Fig. 12). The two MSI classes did not differ in OS or RFS, but M0 macrophages and B cells were linked to longer OS and shorter RFS in MSI class 1 whereas M2 macrophages, CD4$^+$ T cells and erythrocytes were linked to shorter OS and RFS in class 2 (multivariable HR > 1; Extended Data Fig. 10a and Supplementary Table 30). The MSI class 1 tumours more often had *ARID2* mutations, while class 2 had more *BRAF* and *SMAD4* CNVs, *FOXP2* amplifications and 7q11

gains (Extended Data Fig. 10b–d). The MSI class 1 tumours also had higher hypoxia (median score, 17 versus 7; Extended Data Fig. 8a). These differences in immune cell composition and hypoxia levels motivate future analyses of immunotherapy responses in the two MSI classes.

## Discussion

Here we carried out a large study integrating WGS and transcriptome data from CRCs, while providing sufficient clinical follow-up to enable

analyses of prognostic factors. The molecular genetic basis of CRC is comparably well characterized, but the majority of analysed tumours stems from clinical trials, large referral hospitals or from tumours actively treated at sampling. Tumours with genetic alterations associated with poor prognosis are under-represented in clinical trials in which inclusion is based on specific criteria, as well as in hospital-based cohorts in which only patients who may be eligible for treatment are analysed[4,8,11]. We performed integrative analyses of CRCs from the incident patient population undergoing surgical removal of the primary tumour. We extend the CRC driver gene compendium by 33 genes, of which two-thirds were previously undescribed as cancer drivers, although, in the majority of instances, belong to cancer-relevant pathways. Several new mutational signatures related to defective DNA MMR and *POLE* mutations were identified. Timing analyses revealed that the vast majority of chromosomal losses are early events, whereas amplifications occur late, and indicated that *TP53* mutation precedes *PIK3CA* mutation and loss of 10q (*PTEN*)[1]. Several previously unknown early events were identified, and the late timing of amplifications and mutations in 1q and *CEP170*, 8q and *TRPS1*, and 20q and *GNAS* motivates further analyses in the contexts of clonal fitness, invasion and metastasis.

Important findings from the integrated analyses were that (1) the favourable CRPS2–CRPS3 type tumours were enriched for chromosome 20 amplifications that have previously been linked to good prognosis[43]; (2) M1 macrophages were enriched in the good-prognosis CRPS1 tumours, and M2 macrophages in the poor-prognosis CRPS4 tumours; (3) key driver gene expression levels correlated with their mutation status; (4) prognostic mutations in regulatory elements were linked to altered expression of specific genes; (5) tumour hypoxia was linked to specific mutational signatures; and (6) MSI tumours divided into two classes with distinct molecular characteristics. Compared with current molecular classifiers, the prognostic CRPS signature provides refined CRC subtyping with the ability to classify the vast majority of tumours. The robustness of CRPS should be validated in larger cohorts with complete follow-up. Summarizing the identified prognostic genomic factors (Table 1), we confirm the previously reported prognostic relevance of mutant *APC*, *BRAF* and *RNF43*, and *SMAD4* loss in nHM CRCs[32–34,50], and report several previously unreported prognostic driver genes, including positive association of the prevalent *TGFBR2* mutations in HM CRCs with survival. Notably, the prognostic driver genes belonged to the WNT, EGFR–KRAS–BRAF or TGFβ pathways. Furthermore, the prognostic mutations in *MT-CYB*, and in the regulatory elements of *ID2*, *HS3ST1* and *DAPK1* warrant further studies. Together, these findings provide fertile grounds for functional studies of CRC genes and for the development of diagnostic and therapeutic modalities. Future characterization of epigenomes, proteomes and metabolomes of the same tumours and patients can provide additional insights into how different prognostic features relate to each other.

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

## Methods

### Patient cohort

Patients diagnosed with CRC between 2004 and 2019, at Uppsala University Hospital or Umeå University Hospital, were eligible for the study. Patients that had (1) a fresh-frozen biopsy or surgical specimen that was estimated by a pathologist to have a tumour cell content of ≥20%; and (2) a patient-matched source of control DNA from whole blood or fresh-frozen colorectal tissue stored in the biobank, were included. Clinical data were extracted from the national quality registry, the Swedish Colorectal Cancer Registry (SCRCR), and completed from medical records. The follow-up for alive patients was a minimum of 3.9 years and a median of 8 years (data lock 14 June 2023), with only one patient lost to follow-up and 994 (94%) with complete 5-year follow-up. Patients included with a diagnosis from 2010 (861 cases; 81%) were obtained from the Uppsala-Umeå Comprehensive Cancer Consortium (U-CAN) biobank collections (Uppsala Biobank and Biobanken Norr)[51]. Unfixed tissue materials from tumour and healthy colon and rectum were handled on ice and frozen on the day of sampling or surgery[52]. Tissue collected in Uppsala was embedded in optimal cutting temperature (OCT) compound (Sakura) and stored at −70 °C. Tissue collected at Umeå University Hospital was frozen in pieces and stored at −70 °C. Haematoxylin-and-eosin-stained sections from the frozen blocks were reviewed by a pathologist to confirm tumour histology and estimate tumour cell content. Matching healthy DNA samples were derived from peripheral blood (522 patients) or adjacent healthy tissue (541 patients). Control RNA was obtained from 120 patient-matched colon or rectum tissue samples. In total, tumours from 1,126 patients were sectioned and sequenced; however, 63 patients were excluded due to lack of high-quality DNA- or RNA-sequencing data from tumour or paired unaffected tissue.

### Tissue retrieval and nucleic acid extraction

For tissue samples from Uppsala, five and eight cryosections of 10 μm each were used for RNA and DNA extraction, respectively. The DNA was extracted using the NucleoSpin Tissue kit (740952, Macherey-Nagel), and RNA was extracted using the RNeasy Mini Kit (74106, Qiagen). For tissue samples from Umeå, DNA and RNA were extracted using the AllPrep DNA/RNA/miRNA Universal kit (80224, Qiagen). Control DNA from blood samples was extracted using the NucleoSpin 96 Blood Core kit (740456, Macherey-Nagel) on a Genomics STARlet robot (Hamilton). For control samples derived from tissue, DNA and RNA were extracted using the same procedures as described for the tumour samples. DNA concentration was measured using the Qubit broad-range dsDNA assay kit in the Qubit system (Invitrogen), and RNA concentration and quality were assessed using the Bioanalyzer RNA 6000 Nano kit (Agilent) for samples from Uppsala and the Tape Station 2200 (Agilent) for samples from Umeå. RNA samples with RIN ≥ 7, 28S:18S ratio ≥0.8 and concentration ≥60 ng μl$^{-1}$ were further analysed. We analysed bulk RNA from tumours and a smaller set of unaffected control CRC tissue to enable analyses across a large sample set. This approach, while common in such analyses, requires careful consideration of the impact of tissue heterogeneity on the results as systematic differences in cell type composition between CRC and healthy colorectal tissues could contribute to variations in gene expression profiles.

### Whole-genome sequencing and data processing

The WGS libraries were constructed from 1,063 primary CRC tumours and their paired control samples according to the manufacturer's instructions for the MGIEasy FS DNA Library Prep Set (1000006987, MGI). The libraries were sequenced on the DNBSEQ platform (MGI) and 100-bp paired-end sequencing was performed to yield data of ≥60× read coverage for all of the samples. During WGS data preprocessing, low-quality reads and adaptor sequences were removed by SOAPnuke (v.2.0.7)[53] with the parameters '-l 5 -q 0.5 -n 0.1 --f

AAGTCGGAGGCCAAGCGGTCTTAGGAAGACAA -r AAGTCGGATCG TAGCCATGTCGTTCTGTGAGCCAAGGAGTTG'. Sentieon Genomics software (v.sentieon-genomics-202010; https://www.sentieon.com/) was used to map and process high-quality reads for downstream analysis[54], which included the following optimised steps: (1) BWA-MEM (v.0.7.17-r1188) with the parameters '-M -K 100000000' in alt-aware mapping model was used to align each tumour and control sample to the human genome reference hg38 (containing all alternate contigs)[55]; (2) alignment reads were sorted by sort mode of Sentieon utility functions; (3) duplicate reads were marked by Picard (http://broadinstitute.github.io/picard/); (4) indel realignment and base quality score recalibration for aligned reads were carried out by GATK[56]; (5) and alignment quality control was done by Picard.

### Somatic short-variant calling

Putative somatic SNVs, MNVs and/or indels were identified in each tumour–control pair using multiple accelerated tools (TNhaplotyper, corresponding to MuTect2[57] of GATK3; TNhaplotyper2, corresponding to MuTect2[57] of GATK4; TNsnv, corresponding to MuTect[58]) and TNscope[59] of Sentieon Genomics software (v.sentieon-genomics-2020 10.01). Passed somatic SNVs, MNVs and indels detected by at least two tools were retrained as ensemble somatic short variants for each paired control–tumour sample. Allele depths of ensemble somatic short variants were recalculated by TNhaplotyper2 (v.sentieon-genomics-20 2010.01). High-confidence ensemble somatic short variants (depth of tumour ≥ 14, depth of control ≥ 8, variant allele reads count of tumour ≥ 2, variant allele reads count of control ≤ 2, variant allele fraction of tumour ≥ 0.005 and variant allele fraction of control ≤ 0.02) were selected for downstream annotation and analysis. These variants were annotated with VEP cache v.101 (corresponding to GENCODE v.35) by Personal Cancer Genome Reporter (PCGR) (v.v0.9.1)[60].

### Somatic SVs and CNV

Somatic SVs were detected in each paired control–tumour sample by BRASS (v.6.3.4; https://github.com/cancerit/BRASS) with the parameters '-j 4 --c 4 --s human --as GRCh38 --pr WGS', and ascatNgs[61] (v.4.5; https://github.com/cancerit/ascatNgs) with the parameters '-g L -q 20 -rs 'human' -ra GRCh38 -pr WGS -c 4 -force -nobigwig'. The genome cache file was generated by VAGrENT[62] (v.3.7.0; https://github.com/cancerit/VAGrENT) with CCDS2Sequence.20180614.txt (https://ftp.ncbi.nlm.nih.gov/pub/CCDS/current_human/CCDS2Sequence.20180614.txt) and ensembl release-104 (http://ftp.ensembl.org/pub/release-104, Homo_sapiens.GRCh38.104.gff3.gz, Homo_sapiens.GRCh38.cdna.all.fa.gz, Homo_sapiens.GRCh38.ncrna.fa.gz). Other files for the required parameters of BRASS and ascatNgs were extracted from CNV_SV_ref_GRCh38_hla_decoy_ebv_brass6+.tar.gz (ftp://ftp.sanger.ac.uk/pub/cancer/dockstore/human/GRCh38_hla_decoy_ebv/CNV_SV_ref_GRCh38_hla_decoy_ebv_brass6+.tar.gz). The SVs present in control samples were filtered from the following analyses. Somatic CNVs were detected in each paired control–tumour sample by facetsSuite (v.2.0.8; https://github.com/mskcc/facets-suite). An image of facetsSuite was pulled from docker://stevekm/facets-suite:2.0.8 and run with singularity (v.3.2.0)[63]. We used the aligned sequence BAM file as input data and executed FACETS in a two-pass mode with the default settings[64]. First, the purity model estimated the overall segmented copy-number profile, sample purity and ploidy. Subsequently, the dipLogR value inferred from diploid state in the purity model enabled the high-sensitivity model to detect more focal events. Allele-specific copy numbers for each high-confidence ensemble somatic short variant were annotated using the wrapper script 'annotate-maf-wrapper.R' with high-sensitivity output. The gene-level copy-number result was re-annotated with GENCODE v.35. Somatic copy-number states were grouped into eight classes based on total copy number (tcn) and minor copy number (also known as lower copy number; lcn) estimated by FACETS, including wild type class (one copy per allele; tcn=2, lcn=1), homozygous deletions (tcn=0,

lcn=0), LOH (tcn=1, lcn=0), copy-neutral LOH (tcn=2, lcn=0), gain-LOH (tcn=3 or 4, lcn=0), gain (tcn=3 or 4, lcn≥1), amp-LOH (tcn≥5, lcn=0) and amp (tcn≥5, lcn≥1).

## ecDNA detection

Amplicons were detected in each sample by PrepareAA (commit ba747ce; https://github.com/jluebeck/PrepareAA) with the parameters '--ref GRCh38 -t 4 --cngain 4.999999 --cnsize_min 50000 --downsample 10 --cnvkit_dir /home/programs/cnvkit.py --run_AA'[65,66]. An image of PrepareAA was obtained from docker://jluebeck/prepareaa:latest and run with singularity (v.3.2.0). The amplicons were classified by AmpliconClassifier (v.0.4.4; https://github.com/jluebeck/AmpliconClassifier) with the parameters '--ref hg38 --plotstyle noplot --report_complexity --verbose_classification --annotate_cycles_file'[67]. The samples were classified on the basis of which amplicons were present in the sample as previously described[24].

## CIN signature quantification

The activities of the 17 CIN signatures presented previously[68] were quantified using CINSignatureQuantification (v.1.0.0; https://github.com/markowetzlab/CINSignatureQuantification) with unrounded copy-number segments from facetsSuite. Tumours with normalized activities larger than zero, in any CIN signature, were identified as CIN samples.

## MSI detection

The MSI status of CRC tumours was determined by running the MSIsensor2 (v.0.1, commit e0798c7; https://github.com/niu-lab/msisensor2) tumour–control paired module (inherited from MSIsensor) with the parameters '-c 15 -b 4'. MSIsensor2 automatically detects somatic homopolymers and microsatellite changes and calculates the MSI score as the percentage of MSI-positive sites in all valid sites. MSIsensor2 software comprises of two modules: tumour-only and paired. The tumour-only module is an algorithm for tumour-only sequencing data, with a recommended cut-off score of 20. By contrast, the paired module is derived from the original MSIsensor1 and the recommended threshold score is 3.5 for MSI[69]. Correlation analyses between the two modules showed a strong correlation between their results, so we selected the paired module. Furthermore, some studies subdivide MSI samples into MSI-low (scores between 3.5 and 10) and MSI-high (scores above 10) based on the paired module. However, our analysis revealed that most of the samples with scores in the MSI-low range according to the paired module had scores above 20 in the tumour-only module, so we considered all samples with an MSI score of ≥3.5 as having MSI.

## Identification of significantly mutated genes

HM tumours associated with MSI or *POLE* mutation are frequently found in CRC. To avoid signals from samples with lower mutation burden from being masked during downstream WGS analyses, we first classified the tumours as HM or nHM based on the total count of somatic short variants according as previously described[70]:

$$N_{SNV} > N_{median\_SNV} + 1.5 \times \text{interquartile range}$$

After a first round of calculations, each HM sample was split into two separate artificial samples with an equal number of mutation counts. This process was repeated until no HM samples were detected by the formula. Outlier times indicate how many times a sample was called as HM in this process. The mutational heterogeneity caused by the increased mutation burden of HM tumours can reduce the power to detect driver genes and affect the identification of mutational signatures[4,27,71]. To identify CRC driver genes, we ran dNdScv[72] (v.0.1.0, commit dcbf8e5; https://github.com/im3sanger/dndscv) on the whole cohort and on HM and nHM samples separately. A list of known cancer genes to be excluded from the indel background model was compiled from the COSMIC Cancer Gene Census[73] (v.95) and intOGen Compendium Cancer Genes (release date 1 February 2020, https://www.intogen.org/)[72,74–80]. Covariates (a matrix of covariates (columns) for each gene (rows)) were updated to covariates_hg19_hg38_epigenome_pcawg.rda (commit 9a59b89; https://github.com/im3sanger/dndscv_data). The reference database was updated to RefCDS_human_GRCh38_GencodeV18_recommended.rda (commit 9a59b89; https://github.com/im3sanger/dndscv_data). The dNdScv R package includes two different dN/dS-based algorithms, dNdSloc and dNdScv. dNdSloc is similar to a traditional dN/dS implementation, while dNdScv also takes into account variable mutation rates across genes and adds a negative binomial regression model using epigenomic covariates to infer the background mutation rate. The list of significant genes was selected by Benjamini–Hochberg-adjusted $P$ values (qall_loc<0.1 or qglobal_cv<0.1) and merged from both dNdSloc and dNdScv. Long genes[81], olfactory receptor genes and genes with transcript per million (TPM) > 1 in less than ten tumours were excluded from the potential driver gene list. Mutually exclusive or co-occurring sets of driver genes were detected using the modified somaticInteractions function of Maftools[82] (v.2.12.0), which performs pair-wise Fisher's exact tests to detect significant (Benjamini–Hochberg false-discovery rate (FDR) < 0.1) pairs of genes.

## Identification of broad and focal somatic copy-number variation

To determine significantly recurrent broad and focal somatic CNVs, GISTIC2.0[83] (v.2.0.23) was run on resulting segmentation profiles from facetsSuite high-sensitivity models with the parameters '-ta 0.3 -td 0.3 -qvt 0.25 -rx 0 -brlen 0.7 -conf 0.99 -js 4 -maxseg 25000 -genegistic 1 -broad 1 -twoside 1 -armpeel 1 -savegene 1 -gcm extreme -smallmem 1 -v 30'. A higher-amplitude threshold according to GISTIC was used for focal copy-number-alteration classification, tumour and control log$_2$ ratio > 0.9 for amplifications and <−0.3 for deletions[83]. Recurrently amplified or deleted regions were identified by GISTIC peaks and genes within each peak were summarized for further analyses.

## Mutational signature analysis

Analyses of mutational signatures were performed by SigProfilerExtraction[84] (v.1.1.4) with the parameters '--reference_genome GRCh38 --opportunity_genome GRCh38 --minimum_signatures 1 --maximum_signatures 40 --nmf_replicates 500 --cpu 12 --gpu True --cosmic_version 3.2'. SigProfilerExtraction consists of two processes: de novo signature extraction and signature assignment[27,85,86]. Hierarchical de novo extraction of SBS, DBS and ID signatures from all samples was followed by estimation of the optimal solution (number of signatures) based on the stability and accuracy of all 40 solutions. After signatures were identified, the activities of each signature were estimated by assigning the number of mutations in each extracted mutational signature to each sample. SigProfilerExtraction also decomposed de novo signatures to the COSMIC[16] signature database[27] (v.3.2). The cosine similarity[87] between mutational signatures of this and the GEL cohorts[28], and this and the PCAWG cohorts[27] (COSMIC v.3.3), were calculated using R (v.4.2.0). A de novo signature was considered novel if the cosine similarity to both GEL and PCAWG signatures was <0.85. The mutational signature associations between decomposed signatures were calculated by Stats::cor (method = "spearman") and corrplot::cor_mtest (conf.level = 0.95, "spearman") in R (v.4.2.0), and those with an FDR-adjusted $P$ < 0.05 were considered to be statistically significant[88].

## Analyses of non-coding somatic drivers in regulatory elements

Regulatory elements were defined using SCREEN (Registry of cCREs V3; https://screen.encodeproject.org/), a registry of cCREs derived from ENCODE data[89]. Active cCREs annotated in 13 tissue samples (small intestine, transverse, sigmoid, left colon tissues) and 7 cell lines (CACO-2, HCT116, HT-29, LoVo, RKO, SW480 and HCEC 1CT) derived

from colon were collected and downloaded from SCREEN, where cCREs are classified into six active groups (promoter-like signatures (PLS), proximal enhancer-like signatures (pELS), distal enhancer-like signatures (dELS), DNase-H3K4me3, CTCF-only and DNase-only) based on integrated DNase, H3K4me3, H3K27ac and CTCF data. Furthermore, the list of genes possibly linked to a cCRE according to experimental evidence (for example, Hi-C) was extracted from the cCRE Details page of the website. Driver analyses were performed by ActiveDriverWGS[71,90] (v.1.1.2, commit 351ca77; https://github.com/reimandlab/ActiveDriverWGSR) with the parameters '-mc 4 -rg hg38 -fh 300' on non-HM samples for each cCREs groups. The missense mutations in the analyses of regulatory regions were removed to avoid confounding signals from known cancer drivers. Mutated elements with a Benjamini–Hochberg FDR < 0.05 were considered to be significant and were used in the following analyses[90]. To evaluate the functional effects of driver cCREs, we examined their prognostic value and compared the expression levels of their linked genes. Cox proportional hazard analyses were performed to identify prognosis-associated cCREs using the Survival R package (v.3.3-1). Furthermore, potential associations between each cCRE and the expression levels of their linked genes were analysed by comparing raw expression values between groups of mutated and wild-type samples using two-sided Wilcoxon rank-sum tests. An FDR adjustment was applied to the $P$ values from the Wilcoxon test and genes with FDR-adjusted $P < 0.05$ were considered to be differentially expressed with statistical significance. Finally, cCREs that had an impact on the expression of linked genes were analysed according to survival.

## Mitochondrial genome somatic mutation and copy-number estimation

We used multiple tools in the GATK4 (v.4.2.0.0) workflow to extract reads mapped to the mitochondrial genome from WGS, perform the mtDNA variant calling and filter the output VCF file based on specific parameters, according to GATK best practices (https://gatk.broadinstitute.org/hc/en-us/articles/4403870837275-Mitochondrial-short-variant-discovery-SNVs-Indels-). Furthermore, false-positive calls potentially caused by reads of mtDNA into the nuclear genome (NuMTs) were examined. These mutations normally have a low VAF but are highly recurrent in multiple tumours, as well as in matched control samples. To remove these false positives, we used stringent sample filtering, especially on variants with heteroplasmy <10%. We first performed two statistical tests as previously described[30]: (1) the VAF of a mutation in the matched control sequences needed to be <0.0034; and (2) the ratios of:

$$N_{MutCtrl}/RD_{Ctrl}/(N_{MutCtrl}/RD_{Ctrl} + N_{MutTum}/RD_{Tum})$$

needed to be <0.0629, where $N_{Mut}$ refers to mutation allele count, RD to average read depth, and Ctrl and Tum are control and matched tumour tissues, respectively. These cut-offs were adapted from a previous study[30] and set by the median results of all mutation candidates plus 2 times the interquartile range. As the mutation rate of tumour-specific NuMTs is around 2.3% (ref. 91), we retained mutations with a frequency of <0.023. To avoid false-negative calls, mutations with $VAF_{max} < 0.1$ and $VAF_{median} < 0.05$ were examined, and the tumours in which the mutation had VAF > 0.05 were retained[92]. The mean sequencing depth for the mitochondrial genome was 14,286-fold, allowing high-sensitivity detection of somatic mutations at a very low levels of heteroplasmy; thus, variants with $0.01 < VAF < 0.95$ were used for subsequent analyses. For mtDNA copy-number calculation, we used pysam (v.0.15.3) to filter and estimate the raw copy number of each sample. We then calculated the normalized copy number as described previously[5]. The survival best cut-point of mtDNA copy number was identified with surv_cutpoint (maxstat test: Maximally Selected Rank and Statistics) implemented in survminer (v.0.4.9). The associations between mutational signatures and mtDNA copy number were calculated by Stats::cor (method = "spearman") and corrplot::cor_mtest (conf.level = 0.95,

"spearman") in R (v.4.2.0), and those with FDR $P < 0.05$ were considered to be statistically significant[88].

## Relative timing of somatic variants and copy-number events

For each nHM tumour, allele-specific copy-number-annotated high-confidence ensemble somatic short variants and high-sensitivity copy-number events of autosomes (except the acrocentric chromosome arms 13p, 14p, 15p, 21p and 22p) were timed and related to one another with different probabilities using PhylogicNDT[25,93] (v.1.0, commit 84d3dd2; https://github.com/broadinstitute/PhylogicNDT). Single-patient timing and event timing in the cohort were inferred using PhylogicNDT LeagueModel as previously described[26]. The driver gene list identified in this cohort was specified to run PhylogicNDT.

## RNA sequencing and determination of gene expression levels

The rRNA was removed from total RNA using the MGIEasy rRNA Depletion Kit (1000005953, MGI) and sequencing libraries were prepared for the 1,063 primary CRC tumours and 120 adjacent control tissue samples using the MGIEasy RNA Library Prep Kit V3.0 (1000006384, MGI) according to the manufacturer's instructions. Sequencing of $2 \times 100$ bp paired-end reads was performed using the DNBSEQ platform (MGI) with a target depth of 30 million paired-end reads per sample. Pre-processing of RNA-seq data, including removal of low-quality reads and rRNA reads, was performed using Bowtie2 (v.2.3.4.1)[94] and SOAPnuke. Clean sequencing data were mapped to human reference GRCh38 using STAR (v.2.7.1a)[95]. Expression levels of genes and transcripts were quantified using RNA-SeQC (v.2.3.6)[96]. Transcripts with expression level 0 in all samples were excluded from further analyses and the mRNA expression matrix ($19,765 \times 1,183$) was converted to $\log_2(TPM + 1)$.

## Detection of oncogenic RNA fusions

Gene fusions were detected by STAR-Fusion[97] (v.1.10.0; https://github.com/STAR-Fusion/STAR-Fusion) using clean FASTQ files with the parameters '--FusionInspector validate --examine_coding_effect --denovo_reconstruct --CPU 8 --STAR_SortedByCoordinate' and Arriba[98] (v.2.1.0; https://github.com/suhrig/arriba) starting with BAM files aligned by STAR[95] (v.2.7.8a; https://github.com/alexdobin/STAR). An image of STAR-Fusion was pulled from docker://trinityctat/starfusion:1.10.0 and run with singularity (v.3.2.0). Genome lib used in STAR-Fusion was downloaded from CTAT genome lib (https://data.broadinstitute.org/Trinity/CTAT_RESOURCE_LIB/__genome_libs_StarFv1.10/GRCh38_gencode_v37_CTAT_lib_Mar012021.plug-n-play.tar.gz). Aligned BAM files for Arriba were generated as described in the user manual (https://arriba.readthedocs.io/en/latest/). Gene fusions from Arriba were then annotated by FusionAnnotator (v.0.2.0; https://github.com/FusionAnnotator/FusionAnnotator) and merged with results of STAR-Fusion. Merged results were then filtered and prioritized with putative oncogenic fusions by annoFuse[99] (v.0.91.0; https://github.com/d3b-center/annoFuse).

## Unsupervised expression classification for generation of CRPS

We used Seurat (v.4.1.0) to identify stable clusters of all CRC samples and among MSI tumours[100]. Potential batch effects or source differences between samples were corrected by Celligner[101] (v.1.0.1; https://github.com/broadinstitute/Celligner_ms), and the resulting matrix was imported into Seurat as scale data. Three different parameters were evaluated by repeating clustering with different k.param in FindNeighbors (10 to 30, step=5), number of principle components (10 to 100, step=5) and resolution in FindClusters (0.5 to 1.4, step=0.1). The stability of clusters was assessed by Jaccard similarity index and the preferred clustering result (resolution=0.9, PC = 20, K = 20) was determined by scclusteval[102] (v.0.0.0.9000).

## CMS and iCMS classification

For the CMS classification, three CMS classifier algorithms (CMSclassifier (v.1.0.0) with random-forest prediction[41], CMSclassifier-single

sample prediction[41] and CMScaller[103] (v.0.9.2)) were evaluated and the results from the CMSclassifier-random forest was used. Expression data were processed using these three R packages separately or combined, generating four sets of results. In the combined mode, the CMS subtype of each tumour was determined when two algorithms made the same prediction, otherwise it was assigned as NA. Among all four sets of results, CMSclassifier-random forest predicted the most control samples as NA and assigned more MSI samples to CMS1, indicating a lower false-positive rate and a higher accuracy. The Intrinsic CMS (iCMS) classification was performed based on 715 marker genes of intrinsic epithelial cancer signature as described previously[7]. The iCMS2 marker genes were obtained from the iCMS2_up and iCMS3_down lists, and the iCMS3_up and iCMS2_down lists were used as iCMS3 markers. Subsequently, the iCMS2 and iCMS3 scores for each tumour were calculated using the 'ntp' function of the CMScaller R package. Tumours were defined as indeterminate if permutation-based FDR was ≥ 0.05.

### Model building and validation of CRPS classification

To validate the CRPS de novo classification, we built a classification model based on a deep residual learning framework, involving the following steps: (1) gene expression data were first converted into pathway profiles by single-sample gene set enrichment analysis (ssGSEA[104]) implemented in Gene Set Variation Analysis (GSVA[105] (v.1.42.0), parameters 'min.sz=5, max.sz=300') using MSigDB[106–108] (v.7.4). We eventually obtained 30,049 pathways for 1,183 samples, including 1,063 tumours and 120 adjacent unaffected control samples. (2) RelieF implemented in scikit-rebate[109] (v.0.62) was used to refine the obtained pathway features. The RelieF algorithm used nearest-neighbour instances to calculate feature weights and assigned a score for the contribution of each feature to the CRPS classification. The features were then ranked by scores and the top 2,000 were selected for the model training. (3) We used TensorFlow[110] (v.2.3.1) to construct the supervised machine learning model with a 50-layer residual network architecture (ResNet50-1D), of which the 4 stacked blocks were composed of 48 convolutional layers, 1 max pool and 1 average pool layer. The filters and strides were set as previously described[111] and the kernel size was set to height. The activation function was set to SeLU, except for the last layer, which used Softmax for full connection. During model compilation, we used the Nadam algorithm as the optimizer in terms of speed of model training and chose Categorical Crossentropy as loss of function in the classification task. To train the model sufficiently, epochs were set to 500 and LearningRateScheduler in TensorFlow was used to control the learning rate precisely in the beginning of each epoch; finally, ModelCheckpoint in TensorFlow was used to save the model with the maximum F1 score. (4) All 1,183 samples were divided into a training set (80%), a test set (10%) and a validation set (10%). Before the model training, a 1D vector, which represents each gene sets row of samples ($gs_1$, $gs_2$, …, $gs_n$), was converted to a 2D matrix (1, $n_{features}$) with the np.reshape function, and used as the input data for Tensor (input shape structures were set to (none, −1, 2000)). ResNet50 learned the representations of the input data and was fitted to the training set. The number of output classes in TensorFlow was set to 6, corresponding to 5 clusters of CRPS and a normal sample cluster. To avoid bias caused by class imbalance during the learning process, the Random OverSampling Examples algorithm in Imbalanced-learn[112] (v.0.9.0) was applied to ensure that at least one sample from each CRPS class could be randomly selected for model training. Samples with class probabilities of less than 0.5 were categorized as NA. Moreover, Shapley Additive exPlanations (SHAP)[113] was applied to explain the model predictions on CRPS classifications, the molecular features of which could therefore be interpreted. To test the CRPS classification model, a total of ten external CRC datasets ($n$ = 2,832) from NCBI GEO[114] (GSE2109, GSE13067, GSE13294, GSE14333, GSE20916, GSE33113, GSE35896 and GSE39582), NCI Genomic Data Commons[115] (TCGA-COAD[4], TCGA-READ[4]) and AC-ICAM[31] were uniformly processed and transformed to pathway profiles with ssGSEA.

After class prediction of these CRC samples by our CRPS classification model, survival and pathway analyses were performed. Among these external datasets, only the GSE39582, TCGA and AC-ICAM cohorts have sufficient sample sizes and completeness of clinical data to allow survival analyses. Thus, the comparisons of prognostic prediction between CMS, iCMS and CRPS were performed using these three datasets individually and combined. Pathway analyses of CRPS from our dataset and from TCGA were performed using CMScaller[103]. The CRPS classification model is available at GitHub (https://github.com/SkymayBlue/U-CAN_CRPS_Model).

### Pathway analyses

GSEA[106] (v.4.2.3 desktop) and MSigDB[107,108] (v.7.4) were used in pathway analyses, with the following settings: filter 'geneset min=15 max=200'. We also used PROGENy[116] (v.1.16.0) to investigate 14 oncogenic pathways in CRPS, as previously described. The integrated presentation of pathways regulated by CRC somatic alterations were processed using PathwayMapper (v.2.3.0; http://pathwaymapper.org/)[117]. Pathway templates were merged, including cross-pathway interactions[118], using the Newt tool (v.3.0.5; https://newteditor.org/)[119], which allows experimental data to be visually overlaid on the pathway templates.

### Hypoxia scoring and associations with mutational features

Hypoxia scores were calculated for 1,063 CRC tumours and 120 unaffected control samples using the Buffa hypoxia signature[45] as previously described[44]. In brief, samples with an mRNA abundance above the median tumour value of each gene in the signature were given a Buffa hypoxia score of +1, otherwise they were given a Buffa hypoxia score of −1. The sum of the score for every gene in the signature is the hypoxia score of the sample. We used a linear model to analyse the associations between hypoxia scores and mutational features of interest in all tumours, nHM tumours and HM tumours using R stats package (v.4.1.0). For each mutational feature tested in the cohort, a full model and a null model were created and both were adjusted for tumour purity, age at diagnosis and sex[120]. The equations for the two models were adapted from a previous study[44]:

$$Full = hypoxia \sim feature + age + sex + purity$$

$$Null = hypoxia \sim age + sex + purity$$

Comparisons between the two models were made using ANOVA, and hypoxia was considered to be statistically significantly associated with a mutational feature when FDR- or Bonferroni-adjusted $P$ values were <0.1. Bonferroni adjustment was applied only to $P$ values when <20 tests were conducted. The scaled residuals for all full models were calculated using the simulateResiduals function in the DHARMa package[121] (v.0.4.5), and their uniform distributions were verified using the Kolmogorov–Smirnov test. Tested mutational features included mutational signatures, SNV, CNV and SV densities, driver mutations and subclonality. In the mutational signature analysis, the proportion of each signature in each tumour was used in the full model. To test the association between hypoxia and specific genetic alterations, we considered 22 metrics of mutational density, including 10 SNV mutation counts encompassing all regions, coding region, non-coding region, nonsynonymous, SNV, DNV, TNV, DEL, INS and INDEL; 8 metrics of CNV mutational density which were adapted from PCAWG[44], including the fraction of genome with total copy-number aberrations (PGA, total), PGA gain, PGA loss, PGA gain:loss, average CNV length, average CNV length gain, average CNV length loss and average CNV length gain:loss; and 4 SV types, including deletion, inversion, tandem-duplication and translocation. Mutational density by deciles of all 22 metrics were calculated using the R package dplyr[122]. Finally, in the subclonality analysis, clonal and subclonal mutations and numbers of subclones for each tumour were derived from PhylogicNDT as described above.

## Prediction of cell types in the tumour microenvironment

The CIBERSORT[48] (v.1.04) and xCell[47] (v.1.1.0) computational methods were applied with the default settings on TPM gene expression data for microenvironment estimation.

## Survival analyses

The OS was defined as time from diagnosis of primary tumour to death or censored if alive at last follow-up, RFS was defined as time from surgery to earliest local or distant recurrence date or death, or censored if no recurrence or death at last follow-up, while survival after recurrence was defined as the time from recurrence to death. The OS analyses included all patients with stage I–IV, whereas patients with stage IV at diagnosis were excluded in the RFS analyses. Separate OS analyses were also performed for stage I–III for some variables. Cox's proportional hazards models were built to determine the prognostic impact of clinical and genomic features using the R packages finalfit and survival (v.1.0.4/v3.3-1). Univariable Cox regression was performed on all identified coding or non-coding drivers and clinical variables, while multivariable Cox regression was applied to drivers that were statistically significant in the univariable analyses ($P < 0.05$) with co-variates including tumour site, pretreatment status, tumour stage, age groups, tumour grade and hypermutation status. The OS and RFS curves were constructed using the Kaplan–Meier method and the differences between groups were assessed using the log-rank test, using the R package survminer (v.0.4.9). In the Supplementary Tables 18, 19, 21, 23 and 30 showing associations with either OS or RFS, analyses showing $P < 0.05$ were marked in bold. No compensation for multiple testing was done in these analyses.

## Ethics declarations

Patient inclusion, sampling and analyses were performed under the ethical permits 2004-M281, 2010-198, 2007-116, 2012-224, 2015-419, 2018-490 (Uppsala EPN), 2016-219 (Umeå EPN) and the Swedish Ethical Review Authority 2019-566. All of the participants provided written informed consent at enrolment. All of the samples were stored in the respective central biobank service facilities in Uppsala (Uppsala Biobank) and Umeå (Biobanken Norr) and obtained for use in analyses here after approved applications. Sequencing and sequence data analyses of pseudonymized samples were performed at BGI Research, which had access to patient age range, sex and tumour-level data. Samples and data were transferred from UU to BGI Research under Biobank Sweden MTA and applicable GDPR standard terms for transfer to third countries. The analysis of patient-level data was performed at UU. The study conformed to the ethical principles for medical research involving human participants outlined in the Declaration of Helsinki.

## Reporting summary

Further information on research design is available in the Nature Portfolio Reporting Summary linked to this article.

## Data availability

Short somatic variant call, CNV and SV data are available at the European Variation Archive[123] under accession number PRJEB61514, expression profiles at the ArrayExpress[124] under accession number E-MTAB-12862 or all data at the CNGB Sequence Archive (CNSA)[125] of the China National GeneBank DataBase (CNGBdb)[126] under accession number CNP0004160. The raw transcriptome data generated in this Article are available under controlled access through EGA under accession number EGAD50000000169. WGS raw data and more detailed clinical information have been deposited at Uppsala University and inquiries to access them should be directed to the corresponding author and U-CAN, a cancer biobank at Uppsala University (https://www.uu.se/forskning/u-can/). Access to raw data and clinical information

is subject to Swedish legal regulations, GDPR, permission from the Swedish Ethical Review Authority and U-CAN terms. All patients in U-CAN have explicitly consented to genomic data deposition in public repositories. However, to protect their integrity and fulfil requirements in an evolving legal landscape, we have opted for restricted access to genome and transcriptome sequence datasets. Access requests can be addressed to the corresponding author and will be responded to within 2 weeks. The remaining data are available within the Article and Supplementary Information. The human genome reference hg38 (containing all alternate contigs) files were downloaded from GATK resource bundle (ftp.broadinstitute.org/gsapubftp-anonymous/bundle/hg38). The basic gene annotation file (gencode.v35.basic.annotation.gtf.gz) was downloaded from GENCODE (ftp.ebi.ac.uk/pub/databases/gencode/Gencode_human/release_35). A high-confidence list of genes with substantial published evidence in oncology (Cancer Gene Census v95) was downloaded from COSMIC (https://cancer.sanger.ac.uk/cosmic). A compendium of mutational cancer driver genes (release date 1 February 2020) was downloaded from intOGen (https://www.intogen.org/). COSMIC mutational signatures (v.3.3) were downloaded from COSMIC (https://cancer.sanger.ac.uk/signatures/downloads/). Genomics England (GEL) 100,000 Genomes Project (100kGP) mutational signatures (science.abl9283_tables_s1_to_s33.xlsx) were downloaded from the Science website (https://www.science.org/doi/10.1126/science.abl9283#supplementary-materials). The Registry of candidate cis-Regulatory Elements (cCREs V3) derived from ENCODE data was downloaded from SCREEN (https://screen.encodeproject.org/). Genome lib (GRCh38_gencode_v37_CTAT_lib_Mar012021.plug-n-play) used in STAR-Fusion was downloaded from CTAT genome lib (https://data.broadinstitute.org/Trinity/CTAT_RESOURCE_LIB/__genome_libs_StarFv1.10/). The Molecular Signatures Database (v.7.4) was downloaded from MSigDB (https://www.gsea-msigdb.org/gsea/downloads.jsp).

## Code availability

The source code for the CRPS classification model is available to use on GitHub under the GPL-2.0 License (https://github.com/SkymayBlue/U-CAN_CRPS_Model).

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

**Acknowledgements** This study was funded by the Swedish Cancer Society (CAN 2018/772 and 21 1719 Pj to T.S. and 22 2054 Pj 01H to B.G.), the Uppsala Cancer Foundation to B.G., the Erling-Persson Foundation (2020-0037 to T.S. and M.U.) and the Guangdong Provincial Key Laboratory of Human Disease Genomics (2020B1212070028 to K.W.). We thank the members of the China National GeneBank (CNGB) and BGI-Henan for assistance with sequencing and for computational resources. The U-CAN tumour biobank and this project were funded by a grant from the Swedish Government (CancerUU) to Uppsala University, Umeå University, KTH Royal Institute of Technology and Stockholm University (2010–ongoing).

**Author contributions** L.N., F.L., K.H., P.-H.E., A.M., V.L., H.B., M.E., F.P., M.U., B.G., C. Lin and T.S. conceived the study. L.N., F.L., M.W., K.H., I.L., P.-H.E., A.M., V.L., F.P., R.P., K.W., B.G., C. Lin and T.S. coordinated the study. L.N., F.L., M.W., T.L., K.H., E.T., I.L., A.M., A.L.-B., C.Z., S.E., L.M., E. Osterman, E. Osterlund, I.N., N.Y., U.G. and C. Lin performed data curation. L.N., F.L., M.W. and T.L. analysed data. L.N., F.L., T.L., K.H., P.-H.E., C. Larsson, F.P., R.P., X.X., K.W., B.G., C. Lin and T.S. administrated and supervised the study. L.N., F.L., M.W., T.L., B.G., C. Lin and T.S. wrote the paper with input from all of the other authors. All of the authors approved the manuscript before submission.

**Funding** Open access funding provided by Uppsala University.

**Competing interests** Authors declare no competing interests.

**Additional information**
**Correspondence and requests for materials** should be addressed to Kui Wu, Bengt Glimelius, Cong Lin or Tobias Sjöblom.

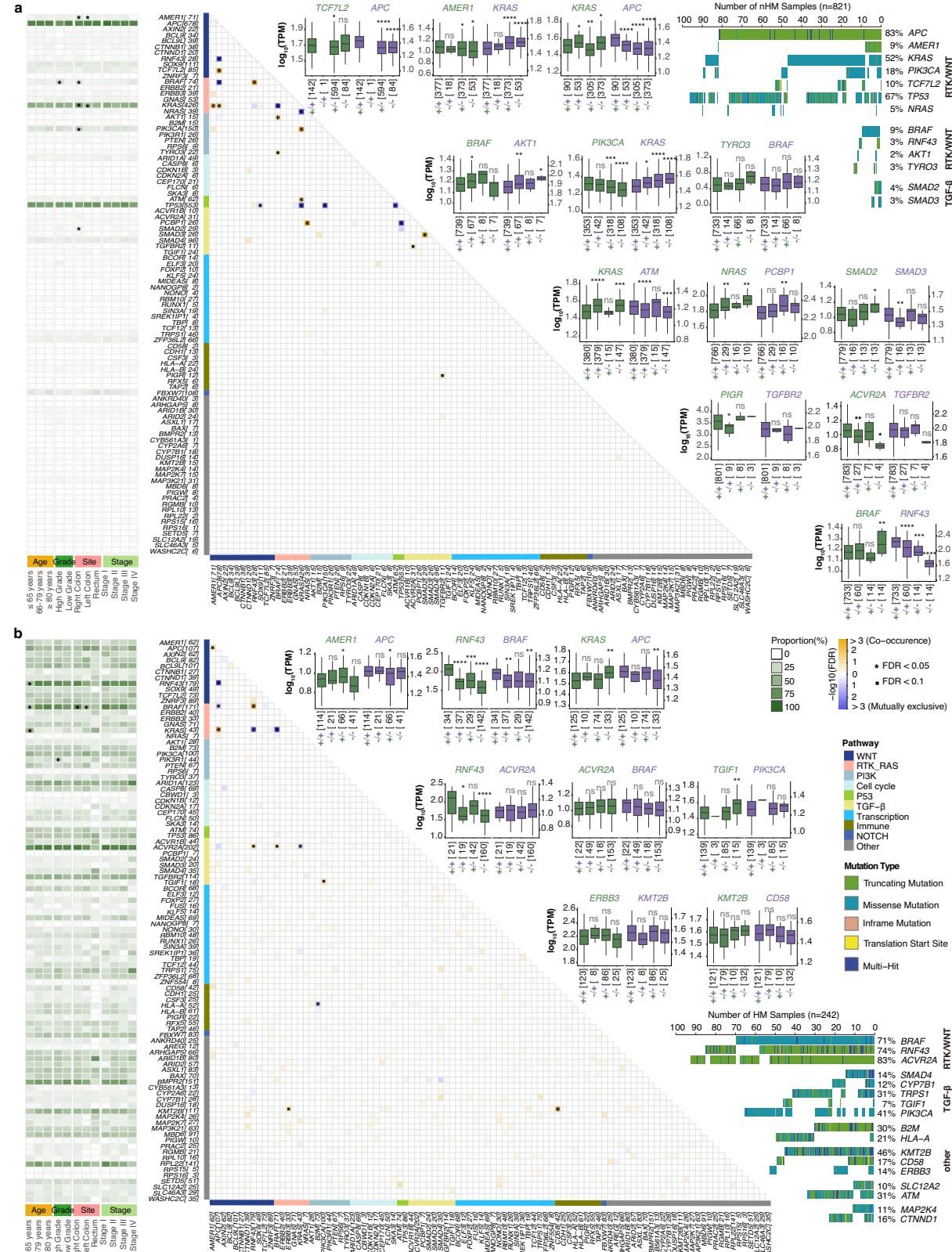

**Extended Data Fig. 1** | See next page for caption.

**Extended Data Fig. 1 | Mutually exclusive and co-occurring gene mutations in the 96 colorectal cancer driver genes displayed by hypermutation status.** Significant pairs of genes with mutually exclusive or co-occurring mutations were detected in **a**, non-hypermutated (n = 821) and **b**, hypermutated (n = 242) tumours with Fisher's Exact test adjusted by Benjamini-Hochberg False Discovery Rate (* FDR < 0.05 and ▪ FDR < 0.1). The number of patients with the mutation is shown inside brackets next to the gene name. Association of genes with clinical features with indication of the proportion of tumours affected is shown to the left (* FDR P < 0.05). Oncoplots display mutually exclusive and co-occurring driver gene mutations grouped by pathway with gene mutation prevalence shown to the right. The expression levels (log10(TPM)) of each pair of genes with co-occurring mutations were compared between wild-type samples (control group, +/+), samples carrying mutations of one gene (+/− or −/+) and samples carrying mutations of both genes (−/−) in the pair. Names of paired genes are indicated on the top of boxes and their colours correspond to colours of "+" or "−". The number of samples for each group is shown at the bottom of each box. The boxes represent the interquartile ranges (IQRs) between the first and third quartiles, the centre line represents the median, and the whiskers extend 1.5 times the IQR from the top and bottom of the box (* P < 0.05, ** P < 0.01, *** P < 0.001, **** P < 0.0001, Two-sided Wilcoxon Rank Sum Test).

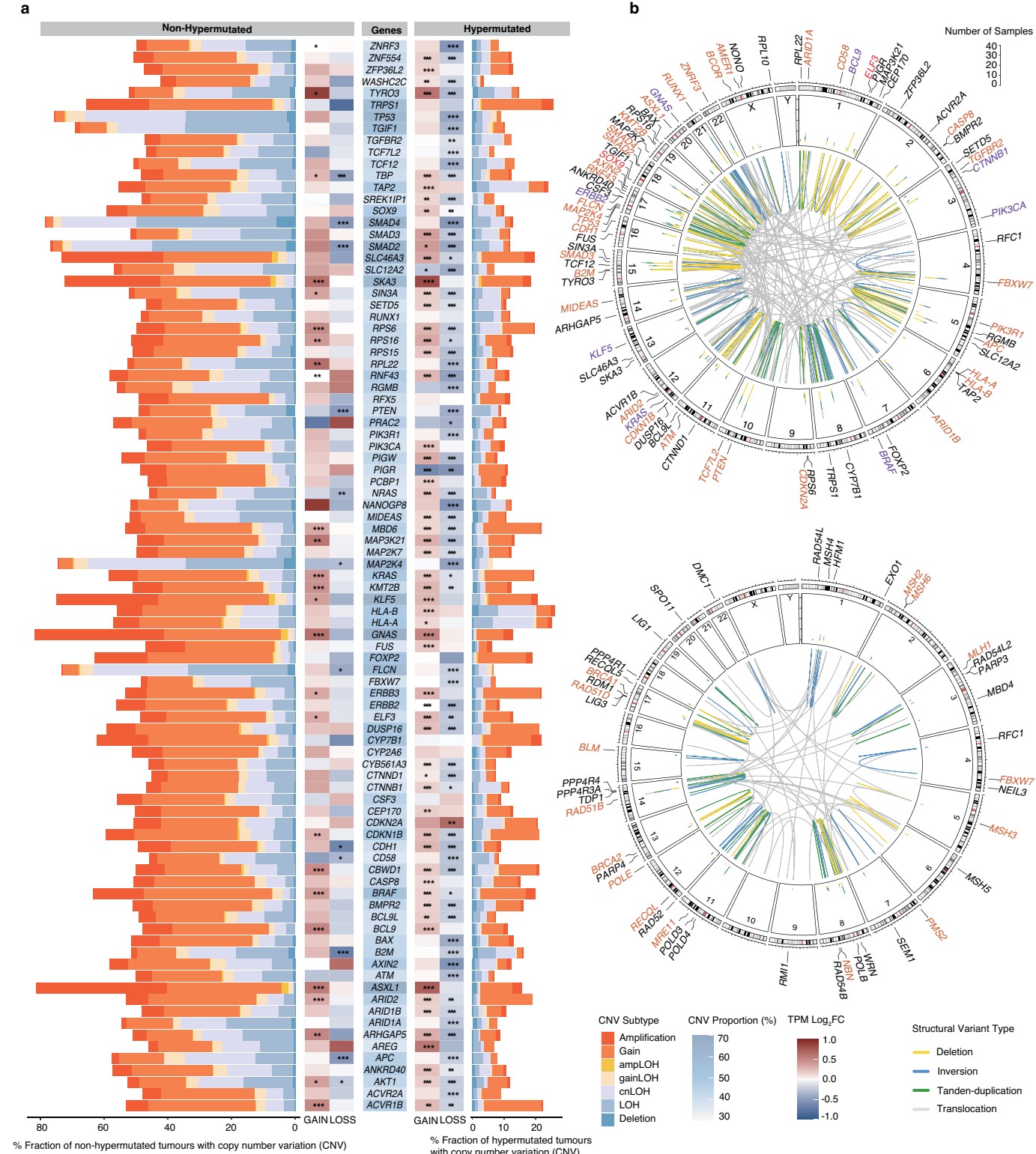

**Extended Data Fig. 2 | Copy number and structural variation landscape for the 96 driver genes. a**, Copy number variation subtypes were called by facetsSuite. RNA expression level (TPM Log2FC) in samples with gains or losses of the driver gene were compared with that of wild-type samples (*FDR < 0.05, **FDR < 0.01, ***FDR < 0.001, Two-sided Wilcoxon Rank Sum Test). **b**, Structural variants affecting driver genes (top) and DNA damage repair genes (bottom). Circos plots with counts (middle ring) for deletions (yellow), inversions (blue), tandem-duplications (green) and translocations (grey), displayed by gene and chromosomal location. CNV, copy number variation; LOH, loss of heterozygosity; cnLOH, copy number neutral LOH; ampLOH, amplification LOH.

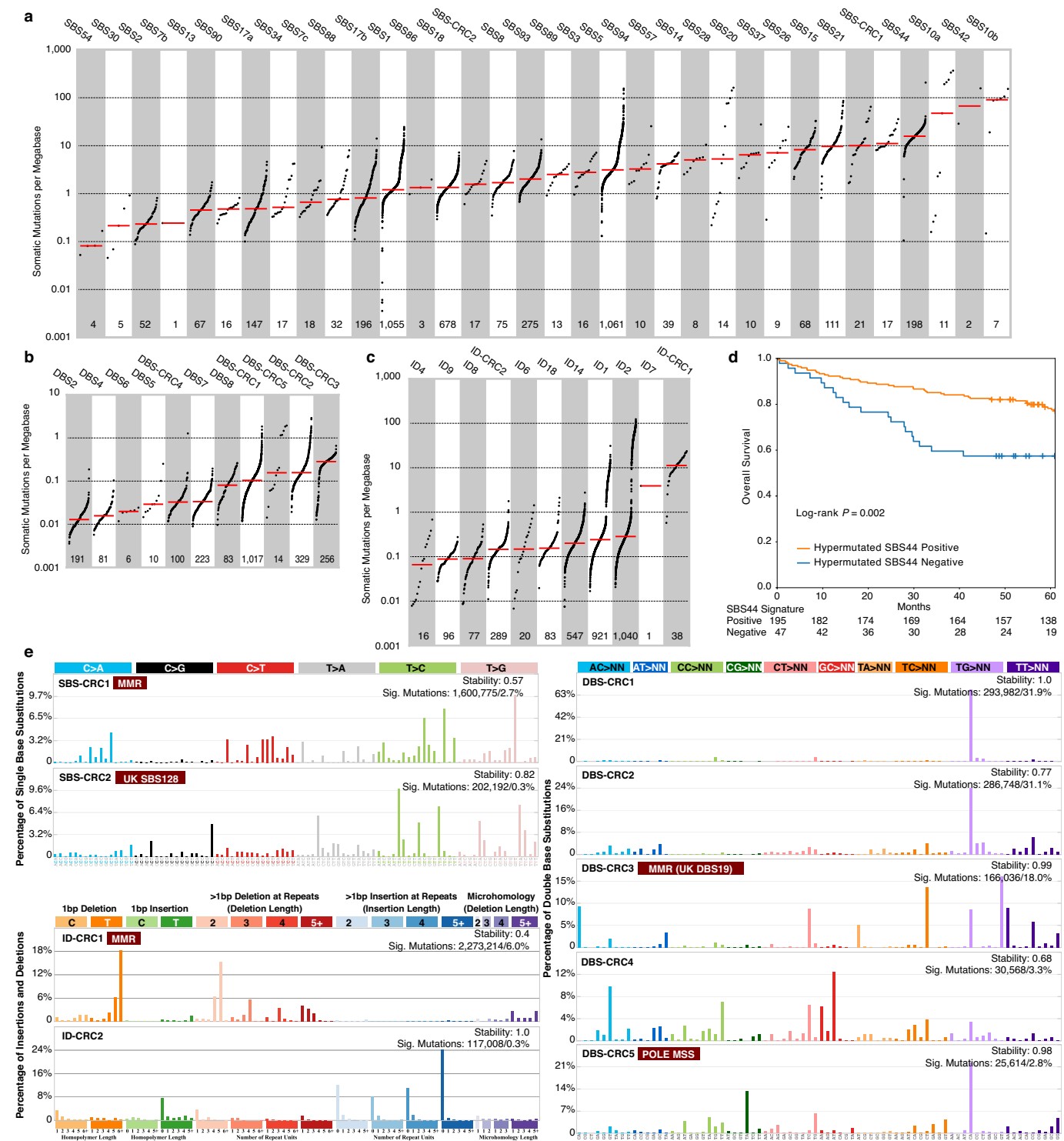

**Extended Data Fig. 3 | Identification of novel and prognostic somatic mutational signatures.** De novo signature extraction and Cosmic signature decomposition by SigProfilerExtraction. Signatures of (**a**) single-base substitution, (**b**) doublet-base substitution, (**c**) and small insertion and deletion sorted by median (red line) mutational burden per megabase with each dot representing one tumour and the number of tumours with each signature indicated below. **d**, Overall survival of patients with stage I-IV hypermutated tumours (n = 242) having the DNA mismatch repair SBS44 signature with Kaplan-Meier curves and log-rank test. **e**, SigProfilerExtraction profiles for the novel and SBS-CRC2, doublet-base substitution DBS-CRC1, DBS-CRC2, DBS-CRC3, DBS-CRC4 and DBS-CRC5 and small insertion and deletion ID-CRC1 and ID-CRC2 signatures. SBS, single-base substitution; DBS, doublet-base substitution; ID, small insertions and deletions; MMR, mismatch repair; MSS, microsatellite stable.

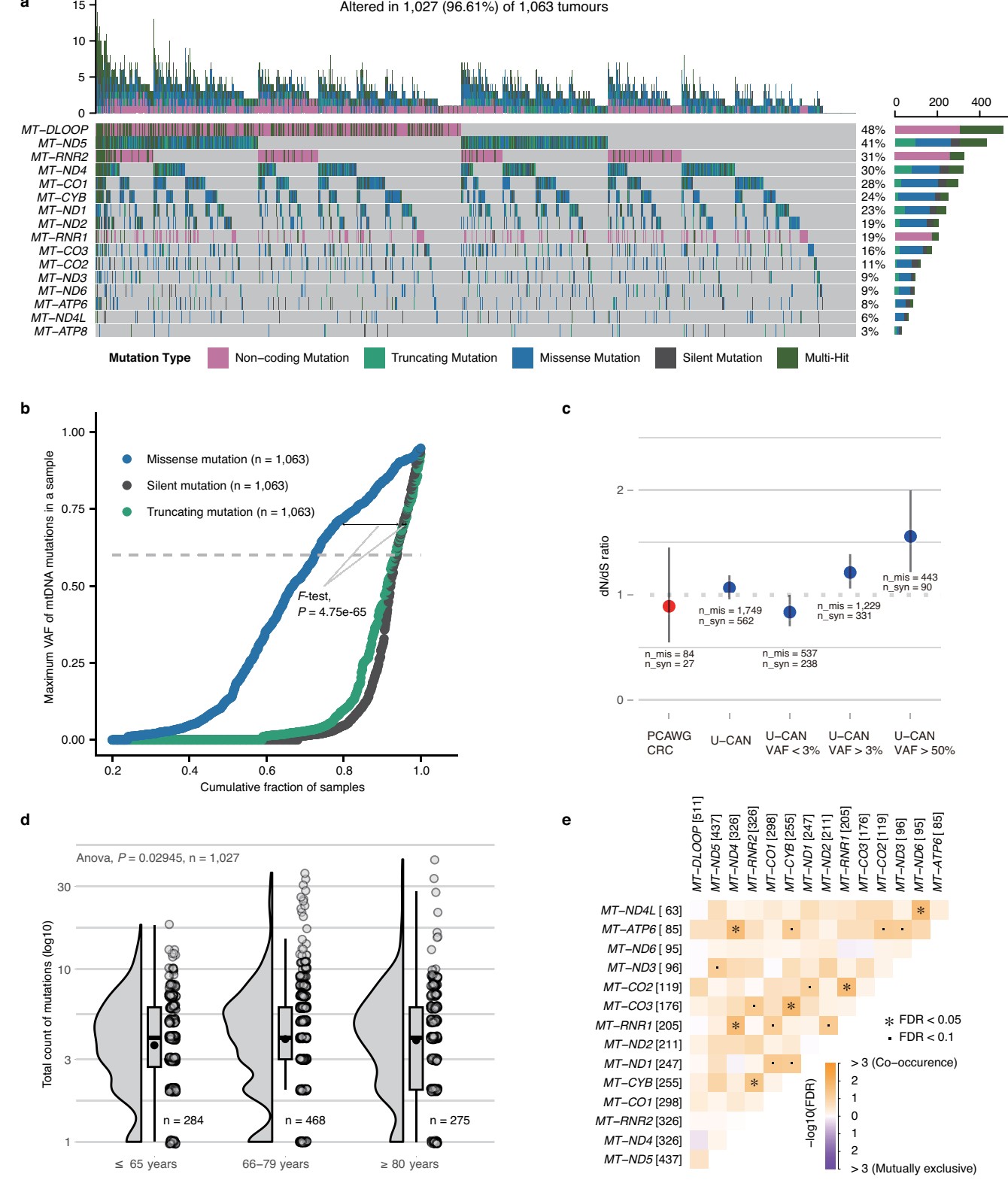

**Extended Data Fig. 4** | See next page for caption.

**Extended Data Fig. 4 | Somatic mutational landscape of mitochondrial genomes in colorectal cancer. a**, Oncoplot of somatic mitochondrial DNA gene (rows) mutations in 1,027 (97%) of the 1,063 sequenced tumours (columns). The TMB for each sample is presented at the top and the number of tumours with the mutation is shown on the right, coloured by mutation type. **b**, Variant allele frequency (VAF) accumulation curves for missense, silent and truncating mitochondria mutations (one-tailed F-test). **c**, dN/dS ratio for mtDNA somatic missense mutations by different VAF cut-offs. The numbers of missense and silent mutations for different VAF cut-offs were indicated. The error bars represent the 95% confidence intervals of the dN/dS ratio (likelihood).

**d**, Total amount of mitochondrial mutations displayed per age group with one-way ANOVA comparison. The boxes represent the interquartile ranges (IQRs) between the first and third quartiles, the centre line represents the median, and the whiskers extend 1.5 times the IQR from the top and bottom of the box. The numbers of tumours in each age group are shown at the bottom of the box plots and mean values are shown as black dots. **e**, Mutually exclusive or co-occurring mitochondrial gene mutations in all tumours with Fisher's Exact test adjusted by Benjamini-Hochberg False Discovery Rate (* FDR < 0.05 and ■ FDR < 0.1). The number of patients with the mutation is shown inside brackets next to the gene name.

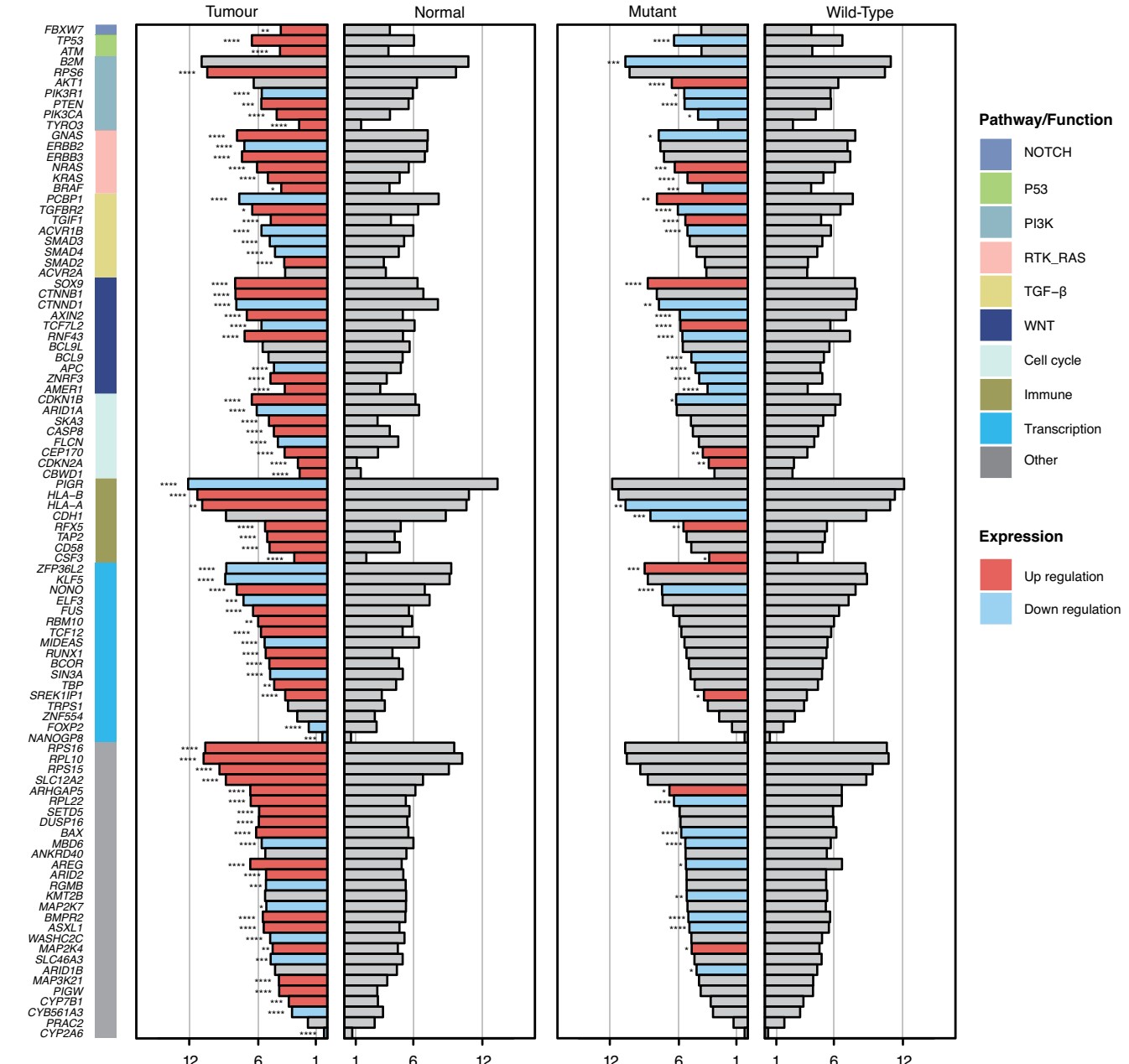

**Extended Data Fig. 5 | Gene expression profiles of the 96 driver genes.**
Mean expression of driver genes in normal colorectal tissues (n = 120) versus
tumours (n = 1,063) (left panel) and in wild-type (WT) versus mutant tumours
(right panel). Genes were sorted by pathways/functions. Significance for
differential gene expression was tested with Two-sided Wilcoxon Rank Sum
Test FDR (* FDR < 0.05, ** FDR < 0.01, *** FDR < 0.001, **** FDR < 0.0001).
Bars represented as log2(mean TPM + 1).

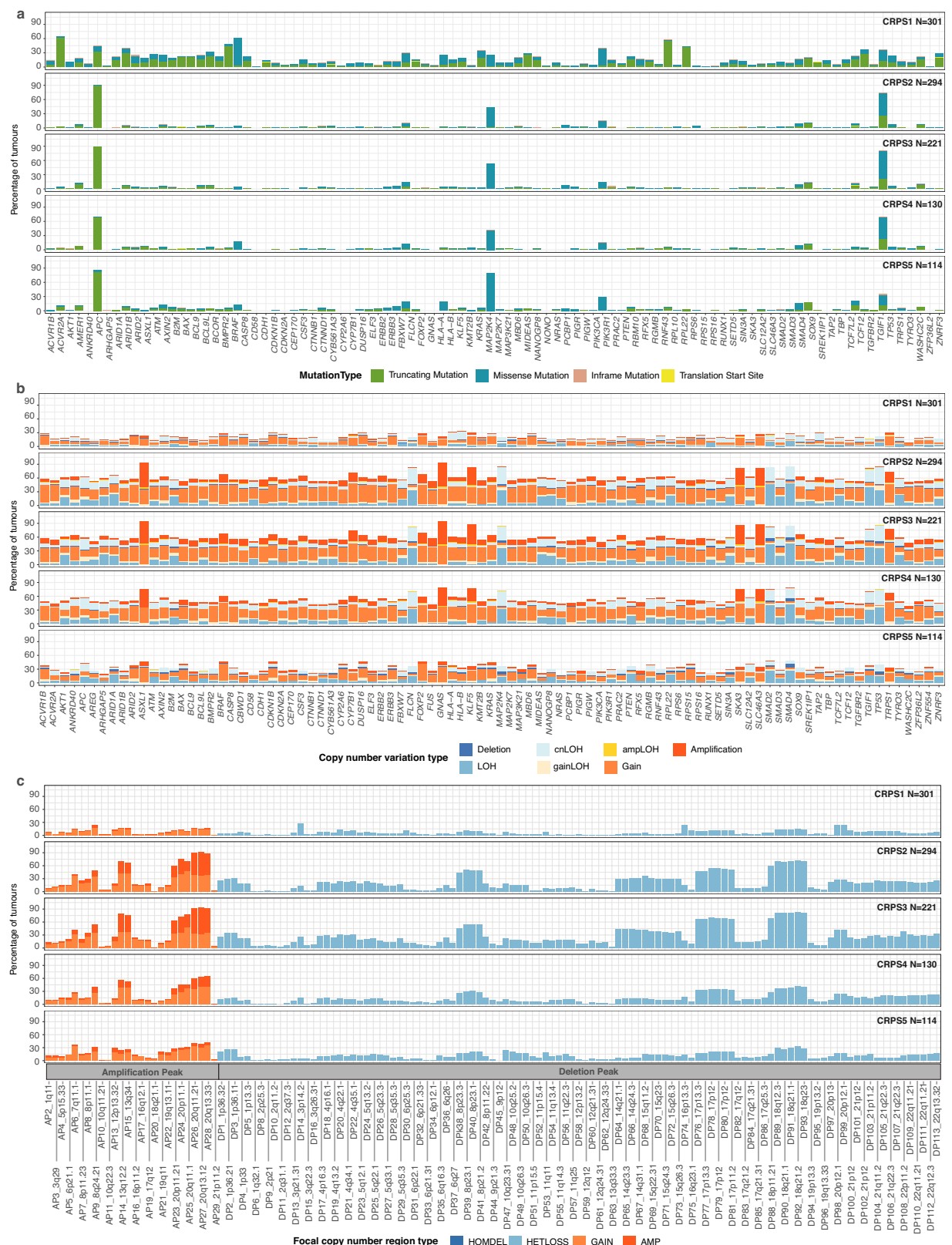

**Extended Data Fig. 6 | Somatic mutations and copy number variation in colorectal cancer prognostic subtypes (CRPS). a**, Somatic mutations in 96 driver genes for the 1,063 colorectal tumours displayed by CRPS subtype. **b**, Frequency and type of somatic copy number variation in 96 driver genes displayed by CRPS subtype. **c**, Focal copy number regions displayed by CRPS subtype determined by GISTIC if Q < 0.1. LOH, loss of heterozygosity; cn, copy number neutral; AMP, amplification; HOMDEL, homozygous deletion; HETLOSS, heterozygous deletion.

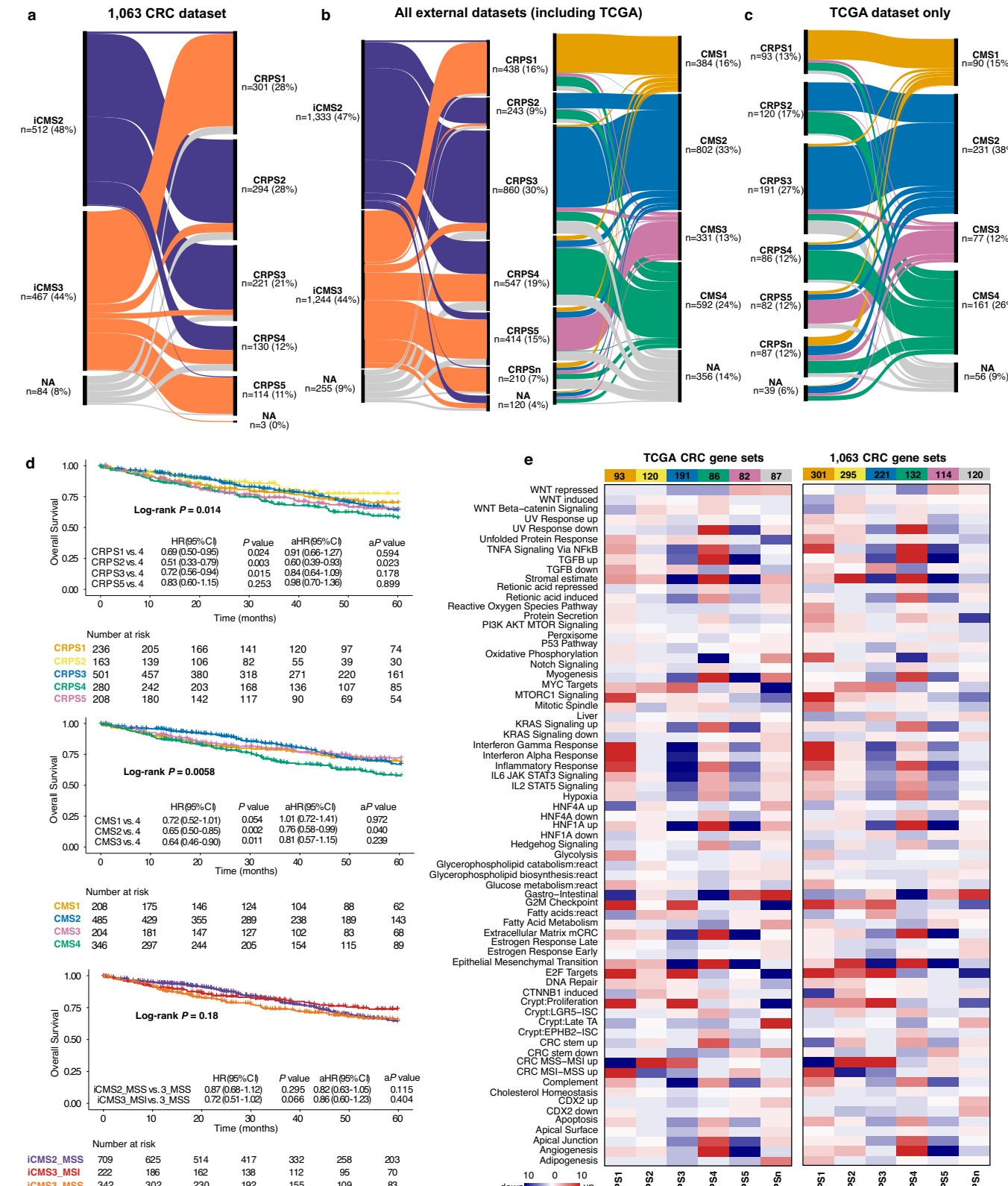

**Extended Data Fig. 7 | Validation of CRPS for colorectal tumour classification. a**, Comparison of CRPS to iCMS in this cohort. **b-c**, In total, eleven external CRC datasets (n = 2,832 samples) from NCBI GEO and NCI Genomic Data Commons were uniformly processed and transformed to pathway profiles with ssGSEA. Comparison of CRPS, CMS and iCMS classification for all external datasets (**b**) and the TCGA COAD/READ dataset only (**c**). The samples were coloured after their CMS subtype. **d**, Overall survival shown by CRPS, CMS and iCMS subgroups for external datasets, calculated with Kaplan-Meier curves and log-rank test. Adjusted HR (aHR) and P (aP) values or HR and P values were calculated by multivariable Cox with or without adjustment for tumour stage. **e**, Comparison of CMS Gene-Set activities using CMScaller (version: v0.9.2) for the TCGA dataset (left) and this cohort (right) displayed by CRPS subgroup (columns). Upregulation marked in red and downregulation in blue for each activity by row. NA, undefined subtype.

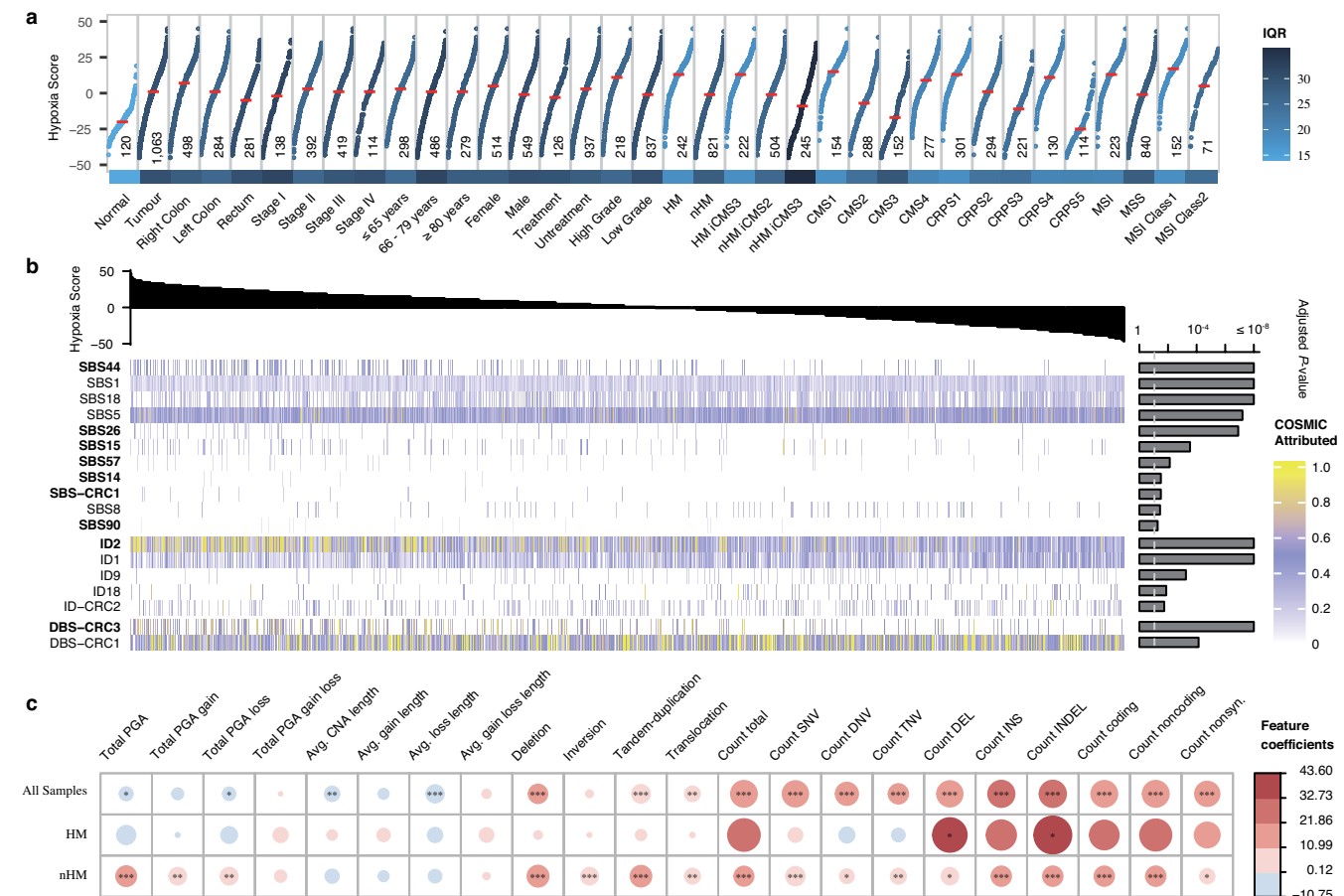

**Extended Data Fig. 8 | Hypoxia in colorectal cancer is associated with mismatch repair deficiency and genomic structural variation. a**, Hypoxia scores based on the Buffa mRNA abundance signature for 1,063 tumour and 120 normal CRC tissues, displayed by clinical, genomic and transcriptomic features. For each group, the median hypoxia score is marked (horizontal red line) and variability is coloured according to the interquartile range (IQR). **b**, Association of hypoxia score (top) with mutational signatures (bottom) coloured by normalized COSMIC signature activity attributed to each sample. Adjusted *FDR P*-values shown to the right and significance threshold indicated by dotted line (F-test full and null models' comparison, *FDR* < 0.05). Signatures

that showed positive correlation with the hypoxia score are shown in bold, the remainder showed negative correlation with the score. **c**, Association of hypoxia scores with somatic structural variants, displayed by hypermutation status. Size and colour of the dots represent regression coefficients of the full model * *FDR* < 0.05, ** *FDR* < 0.01, *** *FDR* < 0.001 (F-test full and null models' comparison). IQR, interquartile range; PGA, percentage of genome with copy number alterations; CNA, copy number alterations; SNV, single nucleotide variation; DNV, double nucleotide variation; TNV, triple nucleotide variation; DEL, deletion; INS, insertion; INDEL, insertion and deletion.

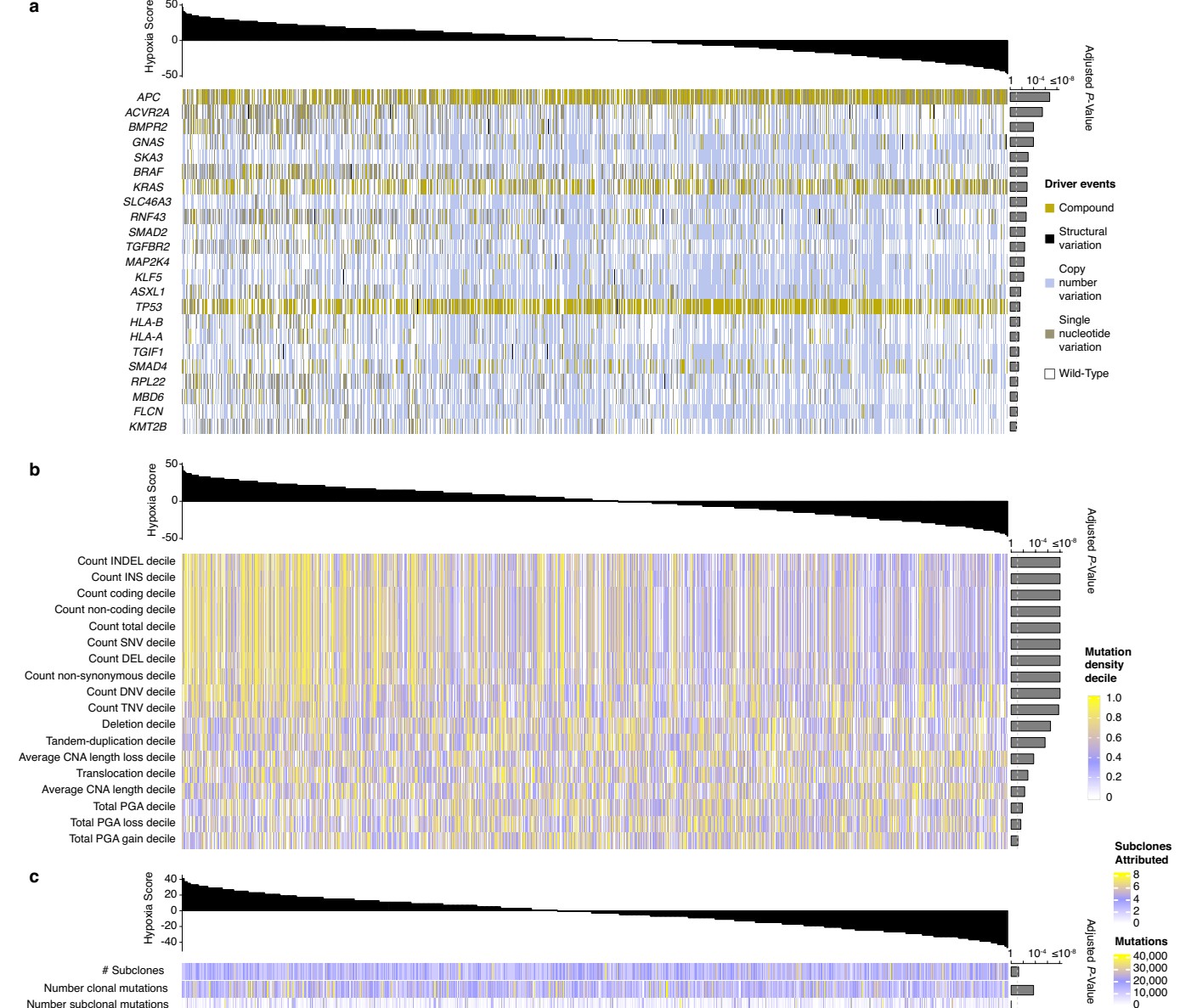

**Extended Data Fig. 9 | Hypoxia correlation with mutations, structural variants and mutational clonality.** Hypoxia scores based on the Buffa mRNA abundance signature were calculated for all tumours (top) and correlated with (**a**) mutations in the 96 driver genes, (**b**) mutation burden and somatic structural variants, and (**c**) numbers of mutations attributed as clonal and subclonal. Adjusted FDR P-values shown to the right and significance threshold indicated by dotted line (F-test full and null models' comparison, *FDR* < 0.05). PGA, percentage of genome with copy number alterations; CNA, copy number alterations; SNV, single nucleotide variation; DNV, double nucleotide variation; TNV, triple nucleotide variation; DEL, deletion; INS, insertion; INDEL, insertion and deletion.

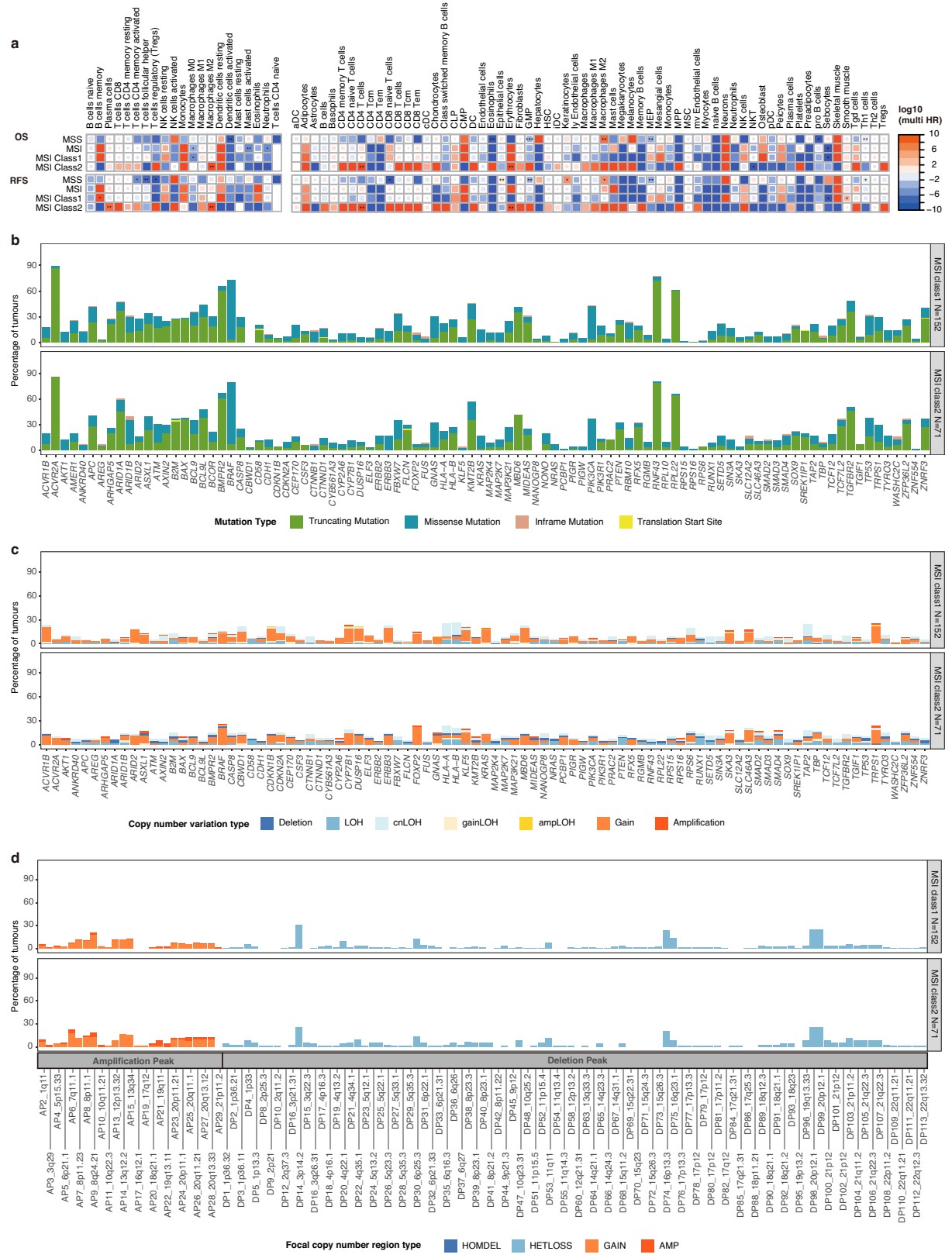

**Extended Data Fig. 10** | See next page for caption.

**Extended Data Fig. 10 | Survival, somatic mutations and copy number variation in two classes of MSI tumours. a**, Overall and recurrence free survival displayed by mismatch repair status and MSI class for cells predicted by CIBERSORT (left) and xCell (right) algorithms. Univariable Cox regression was performed on cell types that showed expression in at least 5 patients with survival data, and statistically significant differences (* P < 0.05, ** P < 0.01, and *** P < 0.001) were further tested by multivariable Cox regression with co-variates including tumour site, treatment status, tumour stage, age groups, and tumour grade. The hazard ratio values are indicated by colour intensity. **b**, Percentage of tumours with somatic mutations in 96 driver genes for MSI class 1 (top) and class 2 (bottom) cases. **c**, Percentage of tumours with somatic copy number variation of 96 driver genes for MSI class 1 (top) and class 2 (bottom) cases. **d**, Percentage of tumours with focal copy number regions (Q < 0.1) gained or lost, determined by GISTIC in the MSI class 1 (top) and class 2 (bottom) cases. LOH, loss of heterozygosity; cn, copy number neutral; AMP, amplification; HOMDEL, homozygous deletion; HETLOSS, heterozygous deletion.

# Reporting Summary

## Statistics

For all statistical analyses, confirm that the following items are present in the figure legend, table legend, main text, or Methods section.

| n/a | Confirmed | |
|---|---|---|
| ☐ | ☒ | The exact sample size (*n*) for each experimental group/condition, given as a discrete number and unit of measurement |
| ☐ | ☒ | A statement on whether measurements were taken from distinct samples or whether the same sample was measured repeatedly |
| ☐ | ☒ | The statistical test(s) used AND whether they are one- or two-sided<br>*Only common tests should be described solely by name; describe more complex techniques in the Methods section.* |
| ☐ | ☒ | A description of all covariates tested |
| ☐ | ☒ | A description of any assumptions or corrections, such as tests of normality and adjustment for multiple comparisons |
| ☐ | ☒ | A full description of the statistical parameters including central tendency (e.g. means) or other basic estimates (e.g. regression coefficient) AND variation (e.g. standard deviation) or associated estimates of uncertainty (e.g. confidence intervals) |
| ☐ | ☒ | For null hypothesis testing, the test statistic (e.g. *F*, *t*, *r*) with confidence intervals, effect sizes, degrees of freedom and *P* value noted<br>*Give P values as exact values whenever suitable.* |
| ☒ | ☐ | For Bayesian analysis, information on the choice of priors and Markov chain Monte Carlo settings |
| ☒ | ☐ | For hierarchical and complex designs, identification of the appropriate level for tests and full reporting of outcomes |
| ☐ | ☒ | Estimates of effect sizes (e.g. Cohen's *d*, Pearson's *r*), indicating how they were calculated |

*Our web collection on statistics for biologists contains articles on many of the points above.*

## Software and code

Policy information about availability of computer code

| | |
|---|---|
| Data collection | Data and metadata was extracted from the national quality registry, the Swedish Colorectal Cancer Registry (SCRCR), and completed from medical records, and no specific software was used. |
| Data analysis | Individual software components used in this study are as follows: SOAPnuke v2.0.7; Sentieon Genomics software v202010; Personal Cancer Genome Reporter (PCGR) v0.9.1; BRASS v6.3.4; ascatNgs v4.5; VAGrENT v3.7.0; facetsSuite v2.0.8; singularity v3.2.0; PrepareAA (git commit ID ba747ce); AmpliconClassifier v0.4.4; CINSignatureQuantification v1.0.0; MSIsensor2 v0.1 (git commit ID e0798c7); dNdScv v0.1.0 (git commit ID dcbf8e5); Maftools v2.12.0; GISTIC2.0 v2.0.23; SigProfilerExtraction v1.1.4; R v4.1.0 and v4.2.0; ActiveDriverWGS v1.1.2 (git commit ID 351ca77); GATK4 v4.2.0.0; pysam v0.15.3; PhylogicNDT v1.0 (git commit ID 84d3dd2); Bowtie2 v2.3.4.1; RNASeQC v2.3.6; STAR-Fusion v1.10.0; Arriba v2.1.0; STAR v2.7.1a and v2.7.8a; FusionAnnotator v0.2.0; annoFuse v0.91.0; Seurat v4.1.0; Celligner v1.0.1; scclusteval v0.0.09000; CMSclassifier v1.0.0; CMScaller v0.9.2; GSVA v1.42.0; scikit-rebate v0.62; Tensorflow v2.3.1; Imbalanced-learn v0.9.0; GSEA v4.2.3; PROGENy v1.16.0; PathwayMapper v2.3.0; Newt v3.0.5; DHARMa v0.4.5; CIBERSORT v1.04; xCell v1.1.0; R packages: stats v4.1.0, survival v3.3-1, survminer v0.4.9, finalfit v1.0.4. The CRPS clustering model (U-CAN_CRPS_Model v1.0.1) is available to use on https://github.com/SkymayBlue/U-CAN_CRPS_Model. |

For manuscripts utilizing custom algorithms or software that are central to the research but not yet described in published literature, software must be made available to editors and reviewers. We strongly encourage code deposition in a community repository (e.g. GitHub). See the Nature Portfolio guidelines for submitting code & software for further information.

# Data

Policy information about <u>availability of data</u>

All manuscripts must include a <u>data availability statement</u>. This statement should provide the following information, where applicable:

- Accession codes, unique identifiers, or web links for publicly available datasets
- A description of any restrictions on data availability
- For clinical datasets or third party data, please ensure that the statement adheres to our <u>policy</u>

Short somatic variant calls, copy number variation and structural variants data can be accessed at the European Variation Archive[123] with accession number PRJEB61514, expression profiles at the ArrayExpress[124] with accession number E-MTAB-12862, or all data at CNGB Sequence Archive (CNSA)[125] of China National GeneBank DataBase (CNGBdb)[126] with accession number CNP0004160. The raw transcriptome data generated in this paper is available under controlled access via EGA with accession number EGAD50000000169. WGS raw data and more detailed clinical information are deposited at Uppsala University and inquires to access them should be directed to the corresponding author and U-CAN, a cancer biobank at Uppsala University (https://www.uu.se/forskning/u-can/). Access to raw data and clinical information is subject to Swedish legal regulations, GDPR, permission from the Swedish Ethical Review Authority, and U-CAN terms. All patients in U-CAN have explicitly consented to genomic data deposition in public repositories. However, to protect their integrity and fulfil requirements in an evolving legal landscape, we have opted for restricted access to genome and transcriptome sequence datasets. Access requests can be addressed to the corresponding author and will be responded within 2 weeks. The remaining data are available within the Article and Supplementary Information.

The human genome reference hg38 (containing all alternate contigs) files were downloaded from GATK resource bundle (ftp.broadinstitute.org/gsapubftp-anonymous/bundle/hg38). The basic gene annotation file (gencode.v35.basic.annotation.gtf.gz) was downloaded from GENCODE (ftp.ebi.ac.uk/pub/databases/gencode/Gencode_human/release_35). A high-confidence list of genes with substantial published evidence in oncology (Cancer Gene Census v95) was downloaded from COSMIC (https://cancer.sanger.ac.uk/cosmic). A compendium of mutational cancer driver genes (release date 2020.02.01) was downloaded from intOGen (https://www.intogen.org/). COSMIC mutational signatures (version: v3.3) were downloaded from COSMIC (https://cancer.sanger.ac.uk/signatures/downloads/). Genomics England (GEL) 100,000 Genomes Project (100kGP) mutational signatures (science.abl9283_tables_s1_to_s33.xlsx) were downloaded from Science website (https://www.science.org/doi/10.1126/science.abl9283#supplementary-materials). The Registry of candidate cis-Regulatory Elements (cCREs V3) derived from ENCODE data were downloaded from SCREEN (https://screen.encodeproject.org/). Genome lib (GRCh38_gencode_v37_CTAT_lib_Mar012021.plug-n-play) used in STAR-Fusion was downloaded from CTAT genome lib (https://data.broadinstitute.org/Trinity/CTAT_RESOURCE_LIB/__genome_libs_StarFv1.10/). The Molecular Signatures Database (version: v7.4) was downloaded from MSigDB (https://www.gsea-msigdb.org/gsea/downloads.jsp).

# Human research participants

Policy information about <u>studies involving human research participants and Sex and Gender in Research.</u>

| Reporting on sex and gender | This study used the sex variable for the purpose of identifying and describing the study cohort and its results landscape. Additionally, the study findings were not specific to one sex, indicating that the results were applicable to both males and females. The sex variable was collected from the national quality registry based on patients' medical records. The cohort included 514 female sex (48%) and 549 male sex (52%) individuals. Patient consent and ethical permits were obtained for the use of this data. |
|---|---|
| Population characteristics | Detailed clinical data is provided in Supplementary Table 1, in line with other colorectal cancer population-based cohorts. Population ancestry is not registered in the clinical records and we didn't perform any ancestry-related analyses, however considering the geographical and demographic characteristics of this cohort, potentially most patients will be of European descent (more specifically Northern European). |
| Recruitment | Patients diagnosed with colorectal cancer between 2004 and 2019, at the Uppsala University Hospital or the Umeå University Hospital, were eligible for the study. Samples obtained had to meet criteria on sample availability and tumour cell content, meaning that the cohort represents patients that underwent surgery, leading to a small under-representation of stage I and IV cancers. There was no other major recruitment biases. |
| Ethics oversight | Patient inclusion, sampling and analyses were performed under the ethical permits 2004-M281, 2010-198, 2007-116, 2012-224, 2015-419, 2018-490 (Uppsala EPN), 2016-219 (Umeå EPN) and the Swedish Ethical Review Authority 2019-566. All participants provided written informed consent at enrolment. All samples were stored in the respective central biobank service facilities in Uppsala (Uppsala Biobank) and Umeå (Biobanken Norr) and obtained for use in analyses here after approved applications. Sequencing and sequence data analyses of pseudonymized samples were performed at BGI Research, which had access to patient age range, sex and tumour level data. Samples and data were transferred from UU to BGI Research under Biobank Sweden MTA and applicable GDPR standard terms for transfer to third countries. The analysis of patient-level data was performed at UU. The study conformed to the ethical principles for medical research involving human participants outlined in the Declaration of Helsinki. |

Note that full information on the approval of the study protocol must also be provided in the manuscript.

# Field-specific reporting

Please select the one below that is the best fit for your research. If you are not sure, read the appropriate sections before making your selection.

☒ Life sciences   ☐ Behavioural & social sciences   ☐ Ecological, evolutionary & environmental sciences

For a reference copy of the document with all sections, see nature.com/documents/nr-reporting-summary-flat.pdf

# Life sciences study design

All studies must disclose on these points even when the disclosure is negative.

| | |
|---|---|
| Sample size | Patients diagnosed with colorectal cancer between 2004 and 2019, at the Uppsala University Hospital or the Umeå University Hospital, were eligible for this study. Patients were included if they had i) a fresh frozen biopsy or surgical specimen that was estimated by a pathologist to have a tumour cell content of ≥20% and ii) a patient-matched source of normal DNA from whole blood or fresh frozen colorectal tissue stored in the biobank. Most samples were surgical specimens and treatment-naïve cases since these generally have enough tissue to be frozen besides the routine formalin-fixed paraffin-embedded storage. No statistical methods were used to predetermine sample size. |
| Data exclusions | Patients that had samples that fulfilled above criteria were excluded if at least one of the sample types (DNA tumour, DNA normal and RNA tumour) were not extracted with enough yield or quality, or if the respective sequencing were of inadequate coverage or evidence for cross-contamination between samples. |
| Replication | To validate the novel Colorectal Caner Progonstic Subtypes (CRPS) classification, we built a classification model based on the deep residual learning framework. To test our CRPS clustering model reproducibility, a total of 10 colorectal cancer data sets with 2,832 patients from both NCBI GEO and NCI Genomic Data Commons were uniformly processed. The accuracy, precision, recall and F1 score were >85% in the validation cases and the prognostic ability of CRPS was recapitulated in this external validation. Replication may not be applicable to the other landscape findings that are descriptive for this cohort. All attempts at replication were successful. |
| Randomization | No randomisation was performed - this was a descriptive study, not an experimental study. |
| Blinding | No blinding was undertaken - this was a descriptive study, not an experimental study. |

# Reporting for specific materials, systems and methods

We require information from authors about some types of materials, experimental systems and methods used in many studies. Here, indicate whether each material, system or method listed is relevant to your study. If you are not sure if a list item applies to your research, read the appropriate section before selecting a response.

## Materials & experimental systems

| n/a | Involved in the study |
|---|---|
| ☒ | Antibodies |
| ☒ | Eukaryotic cell lines |
| ☒ | Palaeontology and archaeology |
| ☒ | Animals and other organisms |
| ☐ | ☒ Clinical data |
| ☒ | Dual use research of concern |

## Methods

| n/a | Involved in the study |
|---|---|
| ☒ | ChIP-seq |
| ☒ | Flow cytometry |
| ☒ | MRI-based neuroimaging |

## Clinical data

Policy information about clinical studies

All manuscripts should comply with the ICMJE guidelines for publication of clinical research and a completed CONSORT checklist must be included with all submissions.

| | |
|---|---|
| Clinical trial registration | Not Applicable |
| Study protocol | Patients diagnosed with CRC between 2004 and 2019, at Uppsala University Hospital or Umeå University Hospital, were eligible for the study. Patients that had i) a fresh frozen biopsy or surgical specimen that was estimated by a pathologist to have a tumour cell content of ≥20% and ii) a patient-matched source of normal DNA from whole blood or fresh frozen colorectal tissue stored in the biobank, were included. Patients included with a diagnosis from 2010 (861 cases; 81%) were obtained from the Uppsala-Umeå Comprehensive Cancer Consortium (U-CAN) biobank collections (Uppsala Biobank and Biobanken Norr). A description of procedures for patient inclusion, sample biobanking, and access to samples can be found at https://www.uu.se/forskning/u-can/. |
| Data collection | Clinical data was extracted from the national quality registry, the Swedish Colorectal Cancer Registry (SCRCR), and completed from medical records. |
| Outcomes | Follow-up for alive patients was minimum 3.9 years and median 8 years (data lock 14th June 2023), with only one patient lost to follow-up and 994 (94%) with complete 5-year follow up. |

