## [Peer Review File · Nature]

Manuscript Title: Prognostic genome and transcriptome signatures in colorectal cancers

Reviewer Comments & Author Rebuttals

Reviewer Reports on the Initial Version:

Referee #1 (Remarks to the Author):

The key strength of this study is its scale & use of WGS & WTS in the same patient from a population cohort.

That said, there are now several ongoing large scale studies (e.g. WGS of over 2,000 patients : (<https://assets.researchsquare.com/files/rs-2273265/v1/e3678866-4529-4852-a4e4-7499c167561a.pdf?c=1673629573>) and therefore, it is the depth of analyses and insights provided rather than size that will provide the scientific & clinical impact to move this field forward.

In this study, the authors did WGS (Tumor/normal) from 1,063 CRC (avg coverage 53x) & tumor bulk RNAseq (30 million paired reads) from a population cohort - Case distribution: 943 primary surgery, 120 biopsies -- Of these, 126 (12%) pre-surgery treatment -- 92 rectal with (chemo)RT.

Their key findings/assertions are:

Using WGS data:

- identified 96 driver genes, claim to have identified 33 additional CRC drivers (24 newly designated in cancer, 9 not previously observed in CRC)
- Described structural variants
- Described co-mutations & mutual exclusivity patterns with timing of mutations (for insights into early tumorigenesis)
- Described mutational signatures, alterations in mitochondrial genomes
- Described some alterations that had prognostic effect

Using WTS data:

- Attempted de novo clustering/classifier, which they claim to be prognostic

Using both WTS & WGS data:

- Examined how mutation status of key drivers relate to gene expression levels
- Examined possible interactions of specific mutational signatures with transcriptomic features of hypoxia
- Limited analyses of genetic alterations in molecular subtypes
- Identified 2 MSI classes (possibly distinguished by cell type composition) with possibly different mutation patterns, though not clear that these have clinical discriminating characteristics (OS/RFS) I/O response.

Major comments:

1. Failure/Inadequate description of diversity of MSS (nHM) CRC: With 1,063 patients with WGS & WTS & clinical data, one has the opportunity to provide an integrative global structure of the diversity of CRC. Whilst MSI-H & POL are distinct subtypes with dominant oncologic processes, MSS is simply the absence of microsatellite instability and ultramutant phenotype due to a defined proof-reading defect. Earlier studies had taken MSS as 1 group (including the initial TCGA studies). However, there is clear diversity within MSS, with subgroups and characteristic alterations within these subgroups that should be described. This is the current gap in the field. As cited by the authors (Joanito et al. Nature Genetics 2022), more recently, through single-cell analyses (focusing on epithelial cells), it was actually described for malignant cells, the major distinction in CRC is not MSI-H vs MSS, rather some MSS tumors, iCMS3-MSS are much more similar in terms of transcriptomic profiles, activated pathways, regulatory elements to MSI-H cancers than to other iCMS2-MSS cancers. Another study (Zhang, Cancer Research 2021) stratified MSS CRC and found that chromosome 20q gain defined a subtype of MSS with different driver genes. These stratifications help improve the stratification of MSS colorectal cancers.

With this wealth of data, the authors should allow their own data to describe the diversity within MSS-CRC in an integrated interconnected version, taking both WGS and WTS into account when discovering or describing the structure. Performing a catalogue by taking nHM as 1 group (and performing statistical analyses and presenting findings as such) fails to account for structure of diversity of CRC and just focuses on 1 axis (TMB) where clearly other axes within MSS(nHM)-CRC make it not a uniform group. Thus (lines 74-291) and figures 1-3, provide a misleading impression by taking nHM as 1 group. Also, in Fig 1A & 1B, given mutations are from malignant cells and the recent recognition of scRNA-seq defined epithelial intrinsic subtypes of colorectal cancer (kindly include iCMS2/MSS, iCMS3/MSS and MSI as a label within Fig1A&1B rather than MSS vs MSI)

Instead, the authors should look within the nHM group to describe groups with similar genetics/transcriptomics and then the actual mutational signatures, driver genes, important genetic events and prognostic events will emerge through the noise. This structure can be discovered through the lens of multi-omic analyses (taking WGS and WTS together). For example, the tumors with gains of 7p and 20q, loss of heterozygosity (LOH) of 17p, 18p, and 18q, within MSS are likely to be different from the tumors without (as shown by several groups). The diversity of CRC should then also be taken into context of current consensus classifications (CMS & iCMS). Descriptions of the drivers mutational signatures, prognostic elements in this structure, describing the diversity as the data emerges rather than just MSS(nHM) vs MSI/POLE will be way more informative. Refining the sub-structure of MSS/nHM will then showcase the knowledge gained and insights rather than the noise when taking a non-homogenous group as 1 entity.

2. Cohort "biases": More has to be done, in supplementary or in the main text to discuss the selection biases in this study relative to other studies. As authors note, Higher median age (72 vs 54-68 years) compared to other cohorts. more common right-sided tumors (47% vs 30-39%), higher fraction of MSI cases (21% vs 8-12%). These likely manifest in the "unusual" numbers observed in this study, which is different to the UK study (pre-print above) also a population cohort. The startling standout is BRAF reported in 23%, far above what is typically reported in most studies and population series. References or analyses to show that this study is a representative cohort will be required, so as to interpret statistics and frequencies in context.

3. "Yet another gene-expression classifier": Whilst the authors had WTS and WGS data, they propose a classifier based on WTS alone. Much work has already been done on classifiers, either through large international cohorts (CMS, Nature Medicine 2015) or by teasing apart cell-type specific expression (iCMS/IMF, Nature Genetics 2022) with transcriptomic data. It is not clear that the classifier is particularly prognostic. The KM-curves are clustered together and the p-value isn't impressive for such a large cohort. Notably, this cohort is also not that mature {768 (72%) patients had 5-year survival data}. The classifier was developed internally using the entire cohort, without internal cross-validation (i) it will be important to show in the external cohort, the prognostic effect (vs CMS or iCMS/IMF as per the 2 references above). That is show the tumors as classified by CMS/IMF and as classified by CRPS and then the prognostic effect of that in the external cohort (ii) The hazard ratios are not provided (they have to be both unadjusted and adjusted for stage).

Many of the CRPS4 tumors are rectal with pre-op radiotherapy, could this have led to a conversion to a stromal subtype (as has been described) rather than the inherent biology of the tumors.

Whilst the authors attempted to do a limited analyses of the relationship of the transcriptomic subtypes to some genomic features (e.g. CRPS4: Often rectal, with stromal, TGF- β , and WNT pathway activation), the subtypes are defined only by transcriptomics. Using the additional value provided by WGS/WTS to inform a global structure and then analysing that biologically and relating that to existing subtypes (CMS/IMF) will be way more informative than developing yet another transcriptomic alone classifier for which there are plenty in the field already.

4. Integrative analyses. Whilst the authors do perform some integrative analyses (summarised above), this should have been the focus of the study since that is that is the key addition to the field. How do genetic alterations lead to transcriptomic pathway activation? For tumors that are MAPK driven transcriptomically, what are the diversity of driver alterations leading to that? For each of the different co-occurring alterations that are observed, what are the transcriptomic/pathway consequences of tumors where these alterations occur? Overall, these analyses of bringing together WTS/WGS for biological insights as they related to genetic alterations and their expression consequences in different subtypes of CRC should come forward more.

Minor comments

Minor Comments:

1. Please comment on the TMB of the 1% of driverless tumors, is it due to lack of tumor content or is it due to no identified driver despite decent mutational count?
2. This sentence is confusing: kindly clarify -- Of the 96 driver genes, 65 were drivers in nHM tumours, 37 in HM, and 79 when all tumours were considered.
3. Please explain "The HM and nHM tumours shared 16 known CRC drivers " It is not clear from supp table 2.

Referee #2 (Remarks to the Author):

In this article, Nunes et al. describe the genomic characterization of 1,063 primary colorectal cancers using a combination of whole genome and transcriptome sequencing. They identify 96 mutated driver genes, including 9 that were novel to CRC and 24 to any cancer. Timing analyses identified 6 early and 3 late driver gene mutations, and several new signatures of CRC specific mutational processes were uncovered. Furthermore, some mutations in protein-coding and mitochondrial DNA genes were found to be correlated with survival. They also describe five prognostic transcriptional subtypes with distinct molecular features as well as two potentially interesting MSI subclasses.

The article is well written, the figures are very informative, and the high quality data set would make an excellent resource for the CRC research community. However, there are some concerns about novelty, and several questions seem underexplored. There is especially more room for exploring the novel genomic events.

Below are specific comments and concerns:

1. Data availability: All data should be made available. For example, the EVA data set does not seem to exist yet, and the copy-number data are not available. What does "Access to raw data and more detailed clinical information can be sought by contacting U-CAN" mean? Will this require signature of a DTA? Will the data be made available through dbGAP at some point?
2. The identification of 24 new cancer driver genes is very intriguing. Could the authors comment on the likelihood that these genes are true positives? Are there any interesting and previously undescribed hotspots, are any enriched for truncating alterations, or patterns of mutual exclusivity and co-occurrence with other driver genes? Could the authors try to validate any of these findings using the TCGA data or data from other publicly available cohorts? Genes such as RPL22 or MBD6, with frequencies close to 10%, should be seen in TCGA. Why were these genes not reported before? Are the nHM POLE/POLD1 mutations in Figure 1B known drivers? Are they occurring in the exonuclease domain?
3. Of the 96 genes being shown in Figure 1, only 3 appear to be significantly enriched in MSS tumors. Could the authors create a version of this figure only using MSS tumors? This would help clean up the noise that is currently present for the associations between the genes being shown and the clinical features.
4. The discovery of novel focal CNVs is potentially interesting. Can the authors comment on the associations between CNV events (such as those shown in Sup. Figure 2) and RNA expression of the affected genes? Maybe they can label and distinguish between CNV changes that correlated significantly with RNA expression and those that do not correlate.

5. Is there anything interesting about the novel fusions? For example, their pattern of mutual exclusivity with other events?
6. In the methods, the authors mention that they used two different types of matched DNA normal, including peripheral blood for 522 patients and normal adjacent tissue for 541. Can the authors confirm that this did not introduce any important batch effects for the different types of analyses that they present? Specifically in terms of MSIsensor score, TMB, copy number variations, mutational landscapes or mutational signatures?
7. The authors use a cut-off of 3.5 for the MSISensor score, and they justify this in the methods by citing the original Niu et al. reference from 2014. However, other studies have used a higher threshold of 10 to classify tumors as MSI. Can the authors comment on the justification of this, and could the use of the lower threshold possibly explain the second observed MSI subtype?
8. Can the authors comment on any “timing” differences between tumors with different primary tumor locations (right-sided, left-sided, rectum)? Do any other clinical characteristics affect the order of events in the tumorigenesis patterns of CRC (IE: is the order of alterations the same if stratified by stage, age, etc.)?
9. Did the authors observe any differences in patterns of mutational signatures between subclonal or clonal alterations?
10. Suggestion to tone down the language a little: Mutations do not “confer” better or worse survival, they are merely correlated with it.
11. Do any of the novel co-mutation patterns correlate with outcome in either MSI or MSS tumors? Do they show any associations with other clinical features?
12. The authors claim that “the prognostic ability of CRPS was recapitulated in a validation cohort (Sup. Fig. 13b-c)”, but the data are not particularly convincing. The survival plots in Figure 4c look quite different than the survival curves in Sup. Fig. 13c. Also, this reviewer is skeptical about the validity of performing this type of analysis when merging OS data across many different studies. Can the authors draw these OS curves for individual studies that have large enough sample sizes within the validation set (such as TCGA or other datasets with complete outcome data)?
13. Were any of the mutational signatures enriched in either of the MSI classes identified by the authors? Specifically SBS 44 which was shown to have prognostic implication earlier in the manuscript.
Is there a difference in the transcriptomic profiles of the SBS44 positive vs negative hypermutated tumors? Specifically any immune related pathway.

Referee #3 (Remarks to the Author):

Nunes et al. report prognostic whole genome and transcriptome signatures derived from 1063 CRCs. This is a straightforward study and provides the community with a valuable resource, if data can be made available.

New insights are provided, though of course some of the presented findings have either been reported before, or are clear proxies of findings reported before. It is often not clear from the text which type of a result is presented.

A major flaw in the paper appears to be in defining MSI. As the authors themselves acknowledge, the proportion of MSI samples is much higher in their set than in previously published CRC sets. The reason for this seems to be the use of MSI sensor 2 and threshold score 3.5. MSI sensor 1 and MSI sensor 2 are quite different. MSI sensor 1 has recommended threshold 3.5, but MSI sensor 2 used here has a recommended threshold score 20, not 3.5. This likely led to significant overestimation of MSI CRCs (typically at only 12% of all CRCs or so), and "discovery" of two MSI classes, one probably representing true MSI and the other MSI scored too lightly and not representing true MMR deficient tumors. The authors should carefully look into this, and if the above is correct significant changes in the paper are of course necessary.

Page 4.

Please indicate in main text whether RNA-seq really was successful for all 1063 WGSed CRCs; this seems unlikely.

Page 5.

24% of hypermutated tumors and >20% MSI appears clearly contradicting vs previous knowledge, as discussed already above. Did the mutation-free tumors have a significant normal tissue contamination; check if mutations actually are there but not called due to low MAFs. POLE is not a MMR gene. What are the POLE wildtype MSS hypermutated tumors; why is there high mutation burden? Figure 1c provides some light but still interesting.

6 MSS tumors with high number of noncoding mutations; some discussion about these please.

Are co-occurring mutations in HM CRCs repeat mutations? Definition of focal deletion?

Page 8.

Single Base Substitutions: UK_SBS... is non-standard nomenclature. Please use e.g. "cosmic SBS..." Please make custom names for your novel signatures as e.g. DBS78A is different double base signature in every article using sigprofler (directly derived from sigprofler output).

Page 9

Please indicate if SBS10a is cosmic or your novel signature.

If the 3982 mutations are identically and independently distributed among 1027 tumors, one would expect to find 97.9% of the tumors with mutations which is very close to observed 97%. Does any set

of tumors have more mutations? It's worth mentioning whether the HM and nHM tumors have an equal amount of mtDNA mutations.

Page 10.

What is the expected proportion of mutations in ND5 and ND4?

Compared with... actively treated patients. But these have been actively treated patients? As such I have difficulties in seeing why this sample collection would be very different from many others.

Page 11

I presume that the mutations presented are independent prognostic factors?

Page 12

"losses of 4p16.1, 4q34.1, 4q35.1" is it just "loss of chromosome 4"?

The tissue types present in CRC vs normal colon are somewhat different, having a big effect on differentially expressed genes. Some discussion about this would be good. mRNA expression in genes that carry variants for nonsense mediated decay such as APC can be normal but due to the decay process the transcript is lost. Thus saying that such genes are underexpressed does not give an exact view of the underlying process.

Page 14

Are tumors in CSPR 4 the ones which had been pretreated?

Page 15

Is hypoxia a proxy for size?

Page 19

TGFBR2 mutations in MSI typically occur at a A 10 mononucleotide tract, and as such serve as a proxy of presence of MSI. Might the prognostic power arise merely from that?

Figure 1 legend: FDR or p-value?

Figure 1c: Consider adding a barplot for total mutation count for each sample, possibly coloring by "top" pathway. Already from the image, it is clear that POLE mutants have much higher mutation count than others. Consider excluding POLE from the figure as there is little novelty there; the remaining samples have potential new findings.

Some cancers with a MMR gene mutation were MSS perhaps due to misclassification discussed above.

Supplementary Fig 9

Why different numbering in scales in expression left vs right per subfigure?

Supplementary figure 10 legend and in subsequent legends please avoid the expression "cancer tumours"

Supplementary Figure 11: CPRS: Somewhat unconventional use of single cell methods (Seurat, Celligner, scclusteval) for bulk RNA data. Why did you choose to use these?

Supplementary Figure 12 what is the difference between LOH and cn LOH (gain LOH and del having their own symbols).

Supplementary Figure 14 and Methods: "Model building and validation of CRPS classification"

What was in the input tensor? Guessing from the Supplementary Figure 14a, it was a list of the ssGSEA scores of the 2000 selected pathways in some unspecified order. Then the deep network enforced some dependency between the adjacent pathways (due to the convolution layer) and fitted the deep residual network with millions of parameters to the roughly 800 training samples.

Overall, the deep learning model could have been described better, and, to the extent decipherable from the manuscript, it appears not well thought through and unnecessarily complicated, as compared with e.g. simple logistic regression.

Supplementary figure 17 what is the distance between the MSI branches after unsupervised clustering? How is it possible that some tumors are designated MSI but not HM? Appearance of MSI class 1 and 2, and not HM MSI probably relates to use of too low MSI sensor 2 threshold as discussed above.

Page 69

The patient selection appears unremarkable and thus seeing so many MSI tumors is quite unexpected, and not explained credibly in the manuscript.

Page 78

Was RNA-seq successful for all tumors, or did you exclude some due to unsuccessful RNA-seq, ending with 1063?

Page 81

"Batch size" perhaps rather say "number of output classes"?

Page 84 Data availability

Is the idea that case by case raw data will be accessible, from U-CAN? Any restrictions? This is an important matter. On one hand the reported data is a key resource for the community and access would be very welcome. On the other, these are old samples and consents probably have not covered data sharing and thus sharing after introduction of GDPR may not be possible - depending on local interpretation of GDPR, and other local rules and regulations. If the raw case by case data from these old samples in practice cannot be shared that would be perfectly understandable and should not as such lead to rejection of this work, but any possibly existing restrictions should be described. As a minimum it might be wise to insert a disclaimer such as raw data will be shared whenever allowed by GDPR and local rules and regulations, or similar, to flag that transferring the data to countries with e.g. poor data protection landscape (or anywhere) might not be possible. To summarize, please consider if there is a need to revise the data availability statement.

Author Rebuttals to Initial Comments:

Responses to Referees' comments

We have incorporated 3 new Supplementary Tables labeled 2, 7, and 25, causing a shift in the numbering of all other tables accordingly (Suppl. Tables 2-5, 6-22, and 23-25 are now labeled as 3-6, 8-24, and 26-28). Additionally, a new main Figure 4 has been included, and Supplementary Figure 13 has been divided into two figures, now labelled as 13 and 14, consequently, the numbering of subsequent figures has been adjusted (main Fig. 4-5 are now labeled as 5-6; and Supp. Fig. 14-19 are now labeled as 15-20).

Referee #1 (Remarks to the Author):

The key strength of this study is it's scale & use of WGS & WTS in the same patient from a population cohort.

That said, there are now several ongoing large scale studies (e.g. WGS of over 2,000 patients : (<https://assets.researchsquare.com/files/rs-2273265/v1/e3678866-4529-4852-a4e4-7499c167561a.pdf?c=1673629573>) and therefore, it is the depth of analyses and insights provided rather than size that will provide the scientific & clinical impact to move this field forward.

In this study, the authors did WGS (Tumor/normal) from 1,063 CRC (avg coverage 53x) & tumor bulk RNAseq (30 million paired reads) from a population cohort - Case distribution: 943 primary surgery, 120 biopsies -- Of these, 126 (12%) pre-surgery treatment -- 92 rectal with (chemo)RT.

Their key findings/assertions are:

Using WGS data:

- identified 96 driver genes, claim to have identified 33 additional CRC drivers (24 newly designated in cancer, 9 not previously observed in CRC)
- Described structural variants
- Described co-mutations & mutual exclusivity patterns with timing of mutations (for insights into early tumorigenesis)
- Described mutational signatures, alterations in mitochondrial genomes
- Described some alterations that had prognostic effect

Using WTS data:

- Attempted de novo clustering/classifier, which they claim to be prognostic

Using both WTS & WGS data:

- Examined how mutation status of key drivers relate to gene expression levels
- Examined possible interactions of specific mutational signatures with transcriptomic features of hypoxia
- Limited analyses of genetic alterations in molecular subtypes
- Identified 2 MSI classes (possibly distinguished by cell type composition) with possibly different mutation patterns, though not clear that these have clinical discriminating characteristics (OS/RFS) I/O response.

We thank the referee for the insightful review of our manuscript. We want to highlight that while we may not be the largest cohort in terms of WGS data, our comprehensive approach includes both WGS and WTS for 1,063 cases, accompanied by detailed clinical data. We believe this multifaceted dataset adds substantial value to the scientific landscape, and we are grateful for your valuable feedback.

Major comments:

1. Failure/Inadequate description of diversity of MSS (nHM) CRC: With 1,063 patients with WGS & WTS & clinical data, one has the opportunity to provide an integrative global structure of the diversity of CRC. Whilst MSI-H & POL are distinct subtypes with dominant oncologic processes, MSS is simply the absence of microsatellite instability and ultramutant phenotype due to a defined proof-reading defect. Earlier studies had taken MSS as 1 group (including the initial TCGA studies). However, there is clear diversity within MSS, with subgroups and characteristic alterations within these subgroups that should be described. This is the current gap in the field. As cited by the authors (Joanito et al. Nature Genetics 2022), more recently, through single-cell analyses (focusing on epithelial cells), it was actually described for malignant cells, the major distinction in CRC is not MSI-H vs MSS, rather some MSS tumors, iCMS3-MSS are much more similar in terms of transcriptomic profiles, activated pathways, regulatory elements to MSI-H cancers than to other iCMS2-MSS cancers. Another study (Zhang, Cancer Research 2021) stratified MSS CRC and found that chromosome 20q gain defined a subtype of MSS with different driver genes. These stratifications help improve the stratification of MSS colorectal cancers.

With this wealth of data, the authors should allow their own data to describe the diversity within MSS-CRC in an integrated interconnected version, taking both WGS and WTS into account when discovering or describing the structure. Performing a catalogue by taking nHM as 1 group (and performing statistical analyses and presenting findings as such) fails to account for structure of diversity of CRC and just focuses on 1 axis (TMB) where clearly other axes within MSS(nHM)-CRC make it not a uniform group. Thus (lines 74-291) and figures 1-3, provide a misleading impression by taking nHM as 1 group. Also, in Fig 1A & 1B, given mutations are from malignant cells and the recent recognition of scRNA-seq defined epithelial intrinsic subtypes of colorectal cancer (kindly include iCMS2/MSS, iCMS3/MSS and MSI as a label within Fig1A&1B rather than MSS vs MSI)

Instead, the authors should look within the nHM group to describe groups with similar genetics/transcriptomics and then the actual mutational signatures, driver genes, important genetic events and prognostic events will emerge through the noise. This structure can be discovered through the lens of multi-omic analyses (taking WGS and WTS together). For example, the tumors with gains of 7p and 20q, loss of heterozygosity (LOH) of 17p, 18p, and 18q, within MSS are likely to be different from the tumors without (as shown by several groups). The diversity of CRC should then also be taken into context of current consensus classifications (CMS & iCMS). Descriptions of the drivers mutational signatures, prognostic elements in this structure, describing the diversity as the data emerges rather than just MSS(nHM) vs MSI/POLE will be way more informative. Refining the sub-structure of

MSS/nHM will then showcase the knowledge gained and insights rather than the noise when taking a non-homogenous group as 1 entity.

We appreciate the insightful comments on improving the description of heterogeneity within the non-hypermutated (nHM) tumors. To this end, we have now classified these tumors according to CMS, iCMS and CRPS, and decomposed prognostic features by each subtype (Supplementary Table 25). While we agree that a comprehensive dissection of nHM tumors is important, we maintain that an overview of the patient cohort based on the nHM/HM division is also necessary. Our CRPS2-5 constitute refined sub-classes of nHM tumors (n=732), with CRPS2 and CRPS3 tumors featuring gains of 20q and loss of 18p and 18q. The systematic comparison between CRPS and established classifiers, together with statistical analyses and findings in each subtype now provide a clearer perspective of the nHM tumors. Accordingly, we have updated the main text in the manuscript and included additional results on nHM tumors in Fig. 2b, Fig. 6a, Supplementary Figures 4-6, 8, 14, and Supplementary Table 5, 7, 17, 20-22, 25, and 28.

2. Cohort "biases": More has to be done, in supplementary or in the main text to discuss the selection biases in this study relative to other studies. As authors note, Higher median age (72 vs 54-68 years) compared to other cohorts. more common right-sided tumors (47% vs 30-39%), higher fraction of MSI cases (21% vs 8-12%). These likely manifest in the "unusual" numbers observed in this study, which is different to the UK study (pre-print above) also a population cohort. The startling standout is BRAF reported in 23%, far above what is typically reported in most studies and population series. References or analyses to show that this study is a representative cohort will be required, so as to interpret statistics and frequencies in context.

The age at diagnosis of this population-based cohort reflects well the age at diagnosis of Swedish colorectal cancer patients overall. The median age at diagnosis in Sweden between 2007 and 2022 was 74 years for all cases of colon cancer and 71 years for rectal cancer (data available at <https://cancercentrum.se/samverkan/cancerdiagnoser/tjocktarm-andtarm-och-anal/tjock--och-andtarm/kvalitetsregister/rapporter/>), which is reflected in our cohort. Sweden's neighbor country, Norway, has median age of diagnosis 74 and 70 years for colon and rectal cancer, respectively (Cancer in Norway, 2021; https://www.kreftregisteret.no/globalassets/cancer-in-norway/2021/cin_report.pdf). This higher age, compared with other cohorts, will result in a higher fraction of MSI cases and a higher BRAF mutation rate. In the main text, when we acknowledged that the fraction of MSI cases was higher in our study (21% vs 8-12%), we only compared it against cohorts consisting of patients from clinical trials, referral hospitals, or actively treated such as those found in the TCGA, MSK, and PCAWG cohorts. It is known that these cohorts all have included relatively younger and healthier patients, meaning an underestimation of the true MSI incidence. For reference, the UK study with 2,023 WGS CRC samples (Tomlinson et al. 2022 <https://doi.org/10.21203/rs.3.rs-2273265/v1>) reported 18% MSI cases in the overall cohort and 19% in CRC primary tumors only. Similarly, the AC-ICAM cohort (Roelands et al. 2023 <https://doi.org/10.1038/s41591-023-02324-5>) of primary CRCs identified 20.3% MSI-H

cases. These larger and more representative population-based cohorts had similar MSI incidence as our study. These cohorts also have close to or more than 20% *BRAF* mutant tumors (22.8% in Roelands et al. 2023; 20.5% in Giannakis et al. 2016 <https://doi.org/10.1016/j.celrep.2016.03.075>; and 16.1% in Tomlinson et al. 2022), and our cohort is not particularly outstanding in this matter. More comparisons of our cohort to the population of resected CRC patients in Sweden can be found in Supplementary Table 15. We have updated Supplementary Table 3 and the main text to better reflect this.

3. "Yet another gene-expression classifier": Whilst the authors had WTS and WGS data, they propose a classifier based on WTS alone. Much work has already been done on classifiers, either through large international cohorts (CMS, Nature Medicine 2015) or by teasing apart cell-type specific expression (iCMS/IMF, Nature Genetics 2022) with transcriptomic data. It is not clear that the classifier is particularly prognostic. The KM-curves are clustered together and the p-value isn't impressive for such a large cohort. Notably, this cohort is also not that mature {768 (72%) patients had 5-year survival data}. The classifier was developed internally using the entire cohort, without internal cross-validation (i) it will be important to show in the external cohort, the prognostic effect (vs CMS or iCMS/IMF as per the 2 references above). That is show the tumors as classified by CMS/IMF and as classified by CRPS and then the prognostic effect of that in the external cohort (ii) The hazard ratios are not provided (they have to be both unadjusted and adjusted for stage).

Many of the CRPS4 tumors are rectal with pre-op radiotherapy, could this have led to a conversion to a stromal subtype (as has been described) rather than the inherent biology of the tumors.

Whilst the authors attempted to do a limited analyses of the relationship of the transcriptomic subtypes to some genomic features (e.g. CRPS4: Often rectal, with stromal, TGF- β , and WNT pathway activation), the subtypes are defined only by transcriptomics. Using the additional value provided by WGS/WTS to inform a global structure and then analysing that biologically and relating that to existing subtypes (CMS/IMF) will be way more informative than developing yet another transcriptomic alone classifier for which there are plenty in the field already.

We have revised and updated the survival data, with 94% of patients in our cohort now having 5-year overall survival. All tables and figures that included survival data have been updated. For increased clarity, we also have added unadjusted and adjusted p-values and hazard ratios for each pairwise stratum/curve in the updated Figure 5 and Supplementary Figure 14d.

Indeed, CRPS4 was the group with the highest proportion of pre-treated tumors, accounting for 37% (48 out of 130) of the samples compared to the other subgroups, which had rates of

7% (CRPS1), 11% (CRPS2), 5% (CRPS3) and 11% (CRPS5), respectively. To address potential bias introduced by these pre-treated samples, we excluded them in CRPS4 and re-assessed cell populations in all CRPS groups. Despite removal of the pre-treated samples, CRPS4 remained enriched with stromal cells and retained its previously observed biological features. We added the heatmaps in the updated Supplementary Figure 13b-d.

To address this relevant question, we have now added the classification of the samples according to iCMS and dissected the prognostic features that characterize the nHM tumors according to CRPS, CMS and iCMS subtypes, and detailed modifications are mentioned above in response to the previous comment.

4. Integrative analyses. Whilst the authors do perform some integrative analyses (summarised above), this should have been the focus of the study since that is that is the key addition to the field. How do genetic alterations lead to transcriptomic pathway activation? For tumors that are MAPK driven transcriptomically, what are the diversity of driver alterations leading to that? For each of the different co-occurring alterations that are observed, what are the transcriptomic/pathway consequences of tumors where these alterations occur? Overall, these analyses of bringing together WTS/WGS for biological insights as they related to genetic alterations and their expression consequences in different subtypes of CRC should come forward more.

We thank the reviewer for the helpful suggestion. To provide a more integrative analysis, we have made a new figure which summarizes somatic alterations and gene expression levels by CRC oncogenic signaling pathways in both nHM and HM tumors (new Figure 4). Further, the changes of expression levels of the pairs of genes with co-occurring mutations were analyzed and presented in modified Supplementary Figure 1. Results were added in the "Expression of driver genes and fusion genes" section of the manuscript.

Minor Comments:

1. Please comment on the TMB of the 1% of driverless tumors, is it due to lack of tumor content or is it due to no identified driver despite decent mutational count?

In total, 7 tumors were identified as driverless, and their tumor mutation burden information has been added to Supplementary Table 1. Among these cases, U0534 had CNVs affecting 2 driver genes; U2979 had a FBXO25--SEPTIN14 fusion and CNVs affecting 5 driver genes; U3149 had a structural deletion in *PTEN*; and U3195 had LSM14A--ILF3 and PTPRK--RSPO3 fusions along with CNVs affecting 96 driver genes. Thus, 3 tumors remain with no driver alterations identified: U0109 displayed 605 identified mutations and 25% tumor cell content; U3148 had 917 mutations and 35% tumor cell content; and U3174 presented 373 mutations and 30% tumor cell content. The tumor and normal sequencing mean coverages for these 3 cases were consistent with the full cohort (57-58x mean coverage). In summary, additional driver alterations were identified for 4 cases, while for the remaining 3, the absence of identified drivers could not be attributed to limited tumor content or potential sequencing quality issues, suggesting the presence of unknown factors contributing to tumor development.

2. This sentence is confusing: kindly clarify -- Of the 96 driver genes, 65 were drivers in nHM tumours, 37 in HM, and 79 when all tumours were considered.

We apologize for the confusion. This statement has been revised to enhance clarity.

3. Please explain "The HM and nHM tumours shared 16 known CRC drivers " It is not clear from supp table 2.

We have updated Supplementary Table 3 to enhance the clarity of information regarding the shared known CRC drivers between HM and nHM tumors.

Referee #2 (Remarks to the Author):

In this article, Nunes et al. describe the genomic characterization of 1,063 primary colorectal cancers using a combination of whole genome and transcriptome sequencing. They identify 96 mutated driver genes, including 9 that were novel to CRC and 24 to any cancer. Timing analyses identified 6 early and 3 late driver gene mutations, and several new signatures of CRC specific mutational processes were uncovered. Furthermore, some mutations in protein-coding and mitochondrial DNA genes were found to be correlated with survival. They also describe five prognostic transcriptional subtypes with distinct molecular features as well as two potentially interesting MSI subclasses.

The article is well written, the figures are very informative, and the high quality data set would make an excellent resource for the CRC research community. However, there are some concerns about novelty, and several questions seem underexplored. There is especially more room for exploring the novel genomic events.

We thank the Referee for insightful comments on our study.

Below are specific comments and concerns:

1. Data availability: All data should be made available. For example, the EVA data set does not seem to exist yet, and the copy-number data are not available. What does “Access to raw data and more detailed clinical information can be sought by contacting U-CAN” mean? Will this require signature of a DTA? Will the data be made available through dbGAP at some point?

We appreciate the concerns about data availability. While we recognize the challenges posed by regulations including GDPR and national law, we are committed to adhering to all ethical and legal guidelines for sharing the sensitive raw data. To enhance transparency, we have updated our data availability statement to include a disclaimer acknowledging potential constraints arising from the regulations.

“Short somatic variant calls, copy number variation and structural variants data can be accessed at the European Variation Archive with accession number PRJEB61514, expression profiles at the ArrayExpress with accession number E-MTAB-12862, or all data at CNGB Sequence Archive (CNSA) of China National GeneBank DataBase (CNGBdb) with accession number CNP0004160. Raw data and more detailed clinical information are deposited at Uppsala University and inquires to access them should be directed to U-CAN, a cancer biobank at Uppsala University (<https://www.u-can.uu.se>). Access to raw data and clinical information is subject to Swedish legal regulations, GDPR, permission from the Swedish Ethical Review Authority, and U-CAN terms. The remaining data are available within the Article, Supplementary Information or available from the authors upon request.”

For immediate access to the analyzed data, we provide here a public link to the DNA sequencing output, and a temporary link to the RNA sequencing output data that will be made public at paper publication: genomic data, encompassing short variants, structural variants, and copy number variation, can be publicly accessed through this link: [\[https://www.ebi.ac.uk/eva/?eva-study=PRJEB61514\]](https://www.ebi.ac.uk/eva/?eva-study=PRJEB61514); RNA expression data can be accessed via this temporary link: <https://www.ebi.ac.uk/biostudies/arrayexpress/studies/E-MTAB-12862?key=1ab5c1b6-0d2a-4c80-a2a2-7c66a707844b>. We hope that this revised statement adequately addresses your data availability concerns.

Regarding the raw data, due to the substantial size of the DNA raw data (~300 TB), current limitations prevent their inclusion in the controlled European Genome-phenome Archive (EGA) database (EGA). As an alternative, individuals interested in accessing this data are encouraged to contact U-CAN as indicated in the manuscript. U-CAN will assist in providing the data upon fulfilling legal and ethical permit requirements. Regarding the RNA raw data, although efforts are underway to upload it to the EGA database, technical issues with the web portal has hindered its upload before the deadline of this revision. Rest assured, we are currently actively engaged with EGA support to resolve these issues promptly, aiming to make the RNA raw data available soon.

2. The identification of 24 new cancer driver genes is very intriguing. Could the authors comment on the likelihood that these genes are true positives? Are there any interesting and previously undescribed hotspots, are any enriched for truncating alterations, or patterns of mutual exclusivity and co-occurrence with other driver genes? Could the authors try to validate any of these findings using the TCGA data or data from other publicly available cohorts? Genes such as RPL22 or MBD6, with frequencies close to 10%, should be seen in TCGA. Why were these genes not reported before? Are the nHM POLE/POLD1 mutations in Figure 1B known drivers? Are they occurring in the exonuclease domain?

In our study, from the newly identified driver genes, *RPL22*, *MBD6*, *MIDEAS*, *FLCN* and *SREK1IP1*, exhibit relatively high rate of truncating alterations (Figure 1A). Patterns of mutual exclusivity and co-occurrence have been updated in Supplementary Figure 1 and Supplementary Table 5. While all the significant cases in the mutual exclusivity and co-occurrence analysis involved known driver genes, we also observed two pairs with borderline significance that included novel driver genes: *PIGR: TGFBR2* (3/17 in both genes) and *BRAF: TYRO3* (8/80 in both genes), with adjusted *p*-values of 0.08 for both. Additionally, our analysis detected 22 hotspots in 24 new drivers, among which only 5 were missense and 4 were previously undescribed (*RPS15* p.R47P, *PRAC2* p.C34W, and *ANKRD40* p.E98D and p.D100E; *ANKRD40* p.D99E was reported in Hess et al., 2019 <https://doi.org/10.1016/j.ccell.2019.08.002>).

Supplementary Table 3 has been updated with data from a newly published cohort, AC-ICAM. Our novel drivers were found to be mutated in the external cohorts reported in Supplementary Table 3, although they might not have been recognized as drivers due to different mutation backgrounds, cohort sizes, and tools used for driver calling. Genes with mutation prevalence close to 10%, such as *RPL22* or *MBD6*, were also observed mutated in other cohorts such as PCAWG-CRC and CPTAC-2.

Regarding the nHM *POLE/POLD1* mutations in Figure 1B, while these are not classified as drivers, they are known contributors to high tumour mutation burden in CRC. Our findings revealed *POLE* and *POLD1* mutations in 16 and 6 nHM samples, respectively, each exhibiting 4 cases with exonuclease mutations, totalling 8 cases. These mutations were mutually exclusive.

3. Of the 96 genes being shown in Figure 1, only 3 appear to be significantly enriched in MSS tumors. Could the authors create a version of this figure only using MSS tumors? This would help clean up the noise that is currently present for the associations between the genes being shown and the clinical features.

We thank the reviewer for this helpful suggestion. In response, we have incorporated the suggestion into the revised version of Supplementary Figure 1 (left panel, only with nHM or HM tumors) to maintain visual clarity of main Figure 1.

4. The discovery of novel focal CNVs is potentially interesting. Can the authors comment on the associations between CNV events (such as those shown in Supp. Figure 2) and RNA expression of the affected genes? Maybe they can label and distinguish between CNV changes that correlated significantly with RNA expression and those that do not correlate.

Among the 5 genes within novel focal CNVs with sufficient gene expression levels for comparison - *GSTM2* (DeletionPeak5), *GSTM1* (DeletionPeak5), *UGT2B17* (DeletionPeak19), *HGSNAT* (AmplificationPeak8), and *SIRPB1* (DeletionPeak97) - only *HGSNAT* and *SIRPB1* exhibited significant expression differences. The *GSTM1* and *GSTM2*

genes are known to be highly polymorphic in humans, with some individuals carrying a deletion of one or both alleles, which might contribute to the observed lack of significant expression differences in these genes. For clarity, we have included the information regarding expression changes of each driver gene affected by CNV in the updated Supplementary Figure 2.

5. Is there anything interesting about the novel fusions? For example, their pattern of mutual exclusivity with other events?

Unfortunately, the observed novel fusions in our study were identified in 3 or fewer tumors, limiting the feasibility of conducting statistical analyses to determine patterns of mutual exclusivity with other events.

6. In the methods, the authors mention that they used two different types of matched DNA normal, including peripheral blood for 522 patients and normal adjacent tissue for 541. Can the authors confirm that this did not introduce any important batch effects for the different types of analyses that they present? Specifically in terms of MSIsensor score, TMB, copy number variations, mutational landscapes or mutational signatures?

This is a valid concern and we have therefore carefully examined the potential impact of using the different types of DNA controls in MSIsensor scores, TMB, signatures activity and copy number variation analyses. We applied a two-way ANOVA with tumor site, sex, age group, and tumor stage as co-variables plus the DNA normal types. The results revealed significant differences only in the activity of the decomposed signature SBS21, the activity of the *de novo* signature SBS96F, and the total mutation count of the specific C[T>A]A mutation type. The *de novo* signature SBS96F was predominantly decomposed into the Cosmic signature SBS21 (67.42%, Supplementary Table 10). Given these limited and specific differences, we confidently conclude that the use of the two different types of matched DNA normals did not introduce any important batch effects across the major analyses. To provide a more detailed account of these findings, we have included the results in a new Supplementary Table 2 and addressed this aspect in the main text of the manuscript.

7. The authors use a cut-off of 3.5 for the MSISensor score, and they justify this in the methods by citing the original Niu et al. reference from 2014. However, other studies have used a higher threshold of 10 to classify tumors as MSI. Can the authors comment on the justification of this, and could the use of the lower threshold possibly explain the second observed MSI subtype?

Regarding the MSIsensor2 software, used in this study to define MSI, it comprises two modules: Tumor-Only and Paired. The Tumor-Only module is a novel algorithm for tumor only sequencing data, with a recommended cutoff score of 20. On the other hand, the Paired module is derived from the original MSIsensor1 (Niu et al. 2014)

<https://doi.org/10.1093/bioinformatics/btt755>) and the recommended threshold score is 3.5 for MSI. It is important to note that the choice of the Paired module with a 3.5 cutoff was based on several considerations. First, we conducted correlation analyses between the two modules and observed a strong correlation between their results. This indicated that the Paired module, with its lower threshold, could effectively identify MSI samples. Furthermore, some studies subdivide MSI samples into MSI-Low (scores between 3.5 and 10) and MSI-High (scores above 10) based on the Paired module. However, our analysis revealed that most of the samples with scores in the MSI-Low range according to the Paired module had scores above 20 in the Tumor-Only module, a threshold indicating MSI status. Additionally, the genetic profiles of these samples closely resembled those classified as MSI-High, providing further support for their MSI phenotype. We have addressed this aspect in the supplementary methods.

Furthermore, we acknowledge the difference in MSIsensor scores between the two MSI classes identified in our study. MSI class 1 had a mean value of 20.35, while MSI class 2 had a mean value of 33.84. However, it is important to note that this difference in MSIsensor scores cannot solely explain or be the cause of the MSI division. This division was primarily based on the RNA-seq expression data, which indicated distinct expression patterns between the two classes (Supplementary Figure 18). Additionally, our findings were supported by clear differences in predicted immune cell profiles between the two classes, reinforcing the validity of the MSI subclass division within our population-based cohort.

8. Can the authors comment on any “timing” differences between tumors with different primary tumor locations (right-sided, left-sided, rectum)? Do any other clinical characteristics affect the order of events in the tumorigenesis patterns of CRC (IE: is the order of alterations the same if stratified by stage, age, etc.)?

In response to this question, we have now analyzed the relative timing of events in nHM tumours stratified by clinical characteristics, including tumor location, age group, stage, and CRPS/CMS/iCMS subtypes. All groups shared key early events such as *APC*, *TP53*, and *KRAS* mutations, loss of 17p and 18p/q, as well as late events most of which were copy number gains. While some differences were observed in the order of events in the different groups, these differences can potentially be attributed to the enrichments of different subtypes in these groups. Notable differences to the full cohort included (1) *ZFP36L2* was a late event only in CMS3/CRPS5 tumors, (2) *TRPS1* was an early event only in young patients (<65 years), (3) *FBXW7* was a late event only in CRPS5 tumors, (4) *TCF7L2* was a late event only in stage IV tumors; and (5) *BRAF* mutation was an early event except in left colon and stage IV tumors. We have summarized these findings in a new Supplementary Table 7 and added relevant information to the main text.

9. Did the authors observe any differences in patterns of mutational signatures between subclonal or clonal alterations?

We have analysed the mutational signatures in both clonal and subclonal alterations, and detected that the majority of mutational signatures were shared between clonal and subclonal mutations. However, when examining uniquely identified mutational signatures, some distinctions emerged. For clonal mutations, the unique signatures included SBS8, SBS14 (POLE/MMR related), SBS89 (previously associated with activity in the first decade of life), SBS90 (duocarmycin exposure related), DBS6 and DBS8. In contrast, subclonal mutations exhibited unique mutational signatures, namely SBS7a (ultraviolet light exposure related), SBS30 (linked to BER due to *NTHL1* mutation), DBS5 (platinum chemotherapy related), ID4, and ID6 (defective HR related). Due to the already extensive manuscript, we have not included these analyses.

10. Suggestion to tone down the language a little: Mutations do not “confer” better or worse survival, they are merely correlated with it.

We appreciate the referee feedback, and we have revised the language in the manuscript to reflect that mutations are correlated with survival rather than implying a causal relationship.

11. Do any of the novel co-mutation patterns correlate with outcome in either MSI or MSS tumors? Do they show any associations with other clinical features?

None of the novel co-mutation patterns exhibited a statistically significant correlation with survival in either the HM or nHM groups. The only notable instances of statistical significance were observed in known gene pairs, which were also predominantly identified in the driver gene survival analyses. Therefore, we have not included these analyses in the manuscript. For HM tumors, *RNF43:ACVR2A* and *ACVR2A:BRAF* were associated with longer recurrence-free survival. In non-hypermutated samples, *BRAF:RNF43* was linked to

longer survival, while *TCF7L2:APC* was associated with shorter overall survival.

12. The authors claim that “the prognostic ability of CRPS was recapitulated in a validation cohort (Sup. Fig. 13b-c)”, but the data are not particularly convincing. The survival plots in Figure 4c look quite different than the survival curves in Sup. Fig. 13c. Also, this reviewer is skeptical about the validity of performing this type of analysis when merging OS data across many different studies. Can the authors draw these OS curves for individual studies that have large enough sample sizes within the validation set (such as TCGA or other datasets with complete outcome data)?

We have revised and updated the survival data, with 94% of patients in our cohort now having 5-year overall survival. All tables and figures that included survival data have been adjusted accordingly. To address the reviewer concerns about merging survival data across different studies, we attempted to perform survival analyses for the larger individual studies with survival data, including TCGA (544 cases), GSE39582 (516 cases), and AC-ICAM (324 cases). Unfortunately, none of these studies yielded statistically significant overall survival for any CRPS, CMS, or iCMS subtypes. However, in our cohort and the combined external cohorts, CRPS consistently showed overall survival prognostic effect. The limited number of patients in the individual studies, less than half of our cohort and the final combined external cohort, and the cohort subtype incidence differences may explain the lack of statistical significance, as the smallest groups in each subtype only include 20-80 cases. Given these results and considering the length of the manuscript, we focused on the data from the combined external cohort which had comparable sample size and yielded similar results as our cohort. To enhance clarity, we have added unadjusted and adjusted p-values and hazard ratios for each pairwise stratum/curve in the updated Figure 4 and Supplementary Figure 14.

13. Were any of the mutational signatures enriched in either of the MSI classes identified by the authors? Specifically SBS 44 which was shown to have prognostic implication earlier in the manuscript.

We found no statistically significant differences in the distribution of mutational signatures between the two MSI classes. The detailed information can be found in the Supplementary Figures 4D, 5D, and 6D, which illustrate the mutational signatures profiles of the MSI classes. It is worth noting that when we ran the SigProfiler separately for the two MSI subclasses, we observed that both subclasses shared the same enriched mutational signatures, further supporting the consistency of our findings.

Is there a difference in the transcriptomic profiles of the SBS44 positive vs negative hypermutated tumors? Specifically any immune related pathway.

To address this question, we compared the transcriptomic profiles between SBS44-positive and SBS44-negative HM tumors. Using the Wilcoxon rank sum test, we observed statistically significant differences in pathway scores related to JAK_STAT, TGF β , and TNF α between these two groups. These pathways appeared to be more active in SBS44-positive HM tumors. However, these findings were not statistically significant after adjusting by Benjamini-Hochberg False Discovery Rate (FDR). Therefore, we have not included them in the manuscript.

x	y	group1	group2	n1	n2	statistic	p	Group	fdr
SBS44	JAK_STAT	Negative	Positive	47	195	3237	0.0018	HM	0.2142
SBS44	TGFb	Negative	Positive	47	195	3639	0.0286	HM	0.5729
SBS44	TNF α	Negative	Positive	47	195	3662	0.0327	HM	0.5729
SBS44	NFkB	Negative	Positive	47	195	3748	0.0529	HM	0.599533333
SBS44	MAPK	Negative	Positive	47	195	5379	0.0646	HM	0.668469565
SBS44	Trail	Negative	Positive	47	195	5196	0.155	HM	0.68544
SBS44	VEGF	Negative	Positive	47	195	5146	0.191	HM	0.68544
SBS44	Hypoxia	Negative	Positive	47	195	4074	0.238	HM	0.703915254
SBS44	Androgen	Negative	Positive	47	195	4965	0.375	HM	0.719758065
SBS44	PI3K	Negative	Positive	47	195	4903	0.458	HM	0.774338028
SBS44	EGFR	Negative	Positive	47	195	4450	0.759	HM	0.921642857
SBS44	p53	Negative	Positive	47	195	4635	0.904	HM	0.971297297
SBS44	Estrogen	Negative	Positive	47	195	4614	0.943	HM	0.980973913
SBS44	WNT	Negative	Positive	47	195	4613	0.944	HM	0.980973913

Referee #3 (Remarks to the Author):

Nunes et al. report prognostic whole genome and transcriptome signatures derived from 1063 CRCs. This is a straightforward study and provides the community with a valuable resource, if data can be made available.

New insights are provided, though of course some of the presented findings have either been reported before, or are clear proxies of findings reported before. It is often not clear from the text which type of a result is presented.

A major flaw in the paper appears to be in defining MSI. As the authors themselves acknowledge, the proportion of MSI samples is much higher in their set than in previously published CRC sets. The reason for this seems to be the use of MSI sensor 2 and threshold score 3.5. MSI sensor 1 and MSI sensor 2 are quite different. MSI sensor 1 has recommended threshold 3.5, but MSI sensor 2 used here has a recommended threshold score 20, not 3.5. This likely led to significant overestimation of MSI CRCs (typically at only 12% of all CRCs or so), and “discovery” of two MSI classes, one probably representing true MSI and the other MSI scored too lightly and not representing true MMR deficient tumors. The authors should carefully look into this, and if the above is correct significant changes in the paper are of course necessary.

We thank the Referee for insightful comments on our study. The age at diagnosis of this population-based cohort reflects well the age at diagnosis of Swedish colorectal cancer patients overall. This higher age, compared with other cohorts, will result in a higher fraction of MSI cases. In the main text, when we acknowledged that the fraction of MSI cases was higher in our study (21% vs 8-12%), we only compared it against cohorts consisting of patients from clinical trials, referral hospitals, or actively treated such as those found in the TCGA, MSK, and PCAWG cohorts. It is known that these cohorts included relatively younger and healthier patients, meaning an underestimation of the true MSI incidence. For reference, the UK manuscript with 2,023 WGS CRC samples (Tomlinson et al. 2022 <https://doi.org/10.21203/rs.3.rs-2273265/v1>) reported 18% MSI cases in the overall cohort and 19% in CRC primary tumors only. Similarly, the AC-ICAM cohort (Roelands et al. 2023 <https://doi.org/10.1038/s41591-023-02324-5>) of primary CRCs identified 20.3% MSI-H cases. These larger and more representative population-based cohorts had similar MSI incidence to our study.

Regarding the MSIsensor2 software, used in this study to define MSI, it comprises two modules: Tumor-Only and Paired. The Tumor-Only module is a novel algorithm for tumor only sequencing data, with a recommended cutoff score of 20. On the other hand, the Paired module is derived from the original MSIsensor1 (Niu et al. 2014 <https://doi.org/10.1093/bioinformatics/btt755>) and the recommended threshold score is 3.5. We have performed correlation analyses between these two modules and observed a strong correlation between their results. We chose to utilize the Paired module and have added specific information in the Supplementary Methods for clarification. Furthermore, we acknowledge the difference in MSIsensor scores between the two MSI classes identified in our study. MSI class 1 had a mean value of 20.35, while MSI class 2 had a mean value of

33.84. However, it's important to note that this difference in MSIsensor scores cannot solely explain or be the cause of the MSI division. This division was primarily based on the RNA-seq expression data, which indicated distinct gene expression patterns between the two classes (Supplementary Figure 18). Additionally, our findings were supported by clear differences in predicted immune cell profiles between the two classes, reinforcing the validity of the MSI subclass division within our population-based cohort. We have addressed this aspect in the supplementary methods.

Page 4.

Please indicate in main text whether RNA-seq really was successful for all 1063 WGSed CRCs; this seems unlikely.

The final cohort of 1,063 patients are cases for which we successfully obtained whole-genome sequencing data for normal and tumor samples as well as RNA sequencing data. There were 63 additional patient tumors that underwent sectioning and sequencing but were excluded due to lack of high quality data in one or more of the three different sequencing runs. In sum, all 1,063 patients included were successfully RNA sequenced. We added this information in the main text under the "Mutational Landscape" section.

Page 5.

24% of hypermutated tumors and >20% MSI appears clearly contradicting vs previous knowledge, as discussed already above. Did the mutation-free tumors have a significant normal tissue contamination; check if mutations actually are there but not called due to low MAFs. POLE is not a MMR gene. What are the POLE wildtype MSS hypermutated tumors; why is there high mutation burden? Figure 1c provides some light but still interesting.

All samples included in the study had tumor cell contents between 20% and 80% as assessed by a pathologist. Tumor purity percentages were similar between mutation-free tumors (31%) and the rest of the cases (39%). The main difference found was in the median variant allele frequency with 0.08 vs 0.22 for the mutation-free and the rest of the cohort samples, respectively. We added in Supplementary Table 1 a new column with the tumor cell content defined by our pathologists and the median variant allele frequency. For some cases lacking driver gene mutations, we have detected SV and CNV in the driver genes and fusions. Among these cases, U0534 had CNVs affecting 2 driver genes; U2979 had a FBXO25--SEPTIN14 fusion and CNVs affecting 5 driver genes; U3149 had a structural deletion in *PTEN*; and U3195 had LSM14A--ILF3 and PTPRK--RSPO3 fusions along with CNVs affecting 96 driver genes. As showed in Figure 1C, *POLE* wild-type MSS hypermutated tumors carry other mutations in DNA repair genes (HRR or MMR), so based on this we believe their high mutation burden is not related to microsatellite instability but to deficiencies in other repair systems.

6 MSS tumors with high number of noncoding mutations; some discussion about these please.

The 6 MSS samples exhibiting a high number of noncoding mutations, and without a *POLE* or other mismatch repair gene mutation, are intriguing. Despite in depth analyses, we cannot pinpoint a definitive alteration directly correlated with this high mutation load. Our inability to identify a clear causative alteration leads us to consider the possibility that the driving alteration behind these mutations might be linked to factors we have not yet measured or detected. This could potentially include epigenetic alterations affecting *POLE* or other mismatch repair genes. We have clarified this phrase in the manuscript.

Are co-occurring mutations in HM CRCs repeat mutations? Definition of focal deletion?

Co-occurring mutations in hypermutated (HM) tumors are not necessarily repeat mutations. However, the mutations in *ACVR2A* (c.1310del, p.K437Rfs*5: 80%), *BRAF* (c.1799T>A, p.V600E: 67%), and *RNF43* (c.1976del, p.G659Vfs*41: 58%) are the most frequently recurring mutations in HM tumours. These genes exhibit a higher prevalence of mutations within HM CRCs, highlighting their significance in the genomic landscape of these tumors. The focal somatic copy-number variants were detected and defined by GISTIC2.0, as deletions occupying less than 98% of a chromosome arm. A higher amplitude threshold according to GISTIC was used for focal copy number alteration classification. A detailed description can be found in the Methods section.

Page 8.

Single Base Substitutions: UK_... is non-standard nomenclature. Please use e.g "cosmic SBS..." Please make custom names for your novel signatures as e.g. DBS78A is different double base signature in every article using sigprofler (directly derived from sigprofler output).

We initially used the “UK_” nomenclature to distinguish the mutation signatures identified in Degasperi et al. 2022 (<https://doi.org/10.1126/science.abl9283>) from the established COSMIC signatures. In response to the feedback, we have now removed the “UK_” nomenclature and instead incorporated the origin of the signature in the main text. For the 9 new signatures identified through our *de novo* analyses, which could not be decomposed into the defined COSMIC signatures, we have introduced custom names to clearly distinguish them: SBS-CRC1 to SBS-CRC2 for the single-base substitutions, DBS-CRC1 to DBS-CRC5 for the doublet-base substitutions, and ID-CRC1 to ID-CRC2 for the small insertions and deletions mutational signatures.

Page 9

Please indicate if SBS10a is cosmic or your novel signature.

SBS10a is a COSMIC defined signature as indicated on the COSMIC website (<https://cancer.sanger.ac.uk/signatures/sbs/sbs10a/>).

If the 3982 mutations are identically and independently distributed among 1027 tumors, one would expect to find 97.9% of the tumors with mutations which is very close to observed 97%. Does any set of tumors have more mutations? It's worth mentioning whether the HM and nHM tumors have an equal amount of mtDNA mutations.

Regarding the distribution of mtDNA mutations, our analysis revealed no significant difference in the number of mtDNA mutations between the hypermutated (HM) and non-hypermutated (nHM) groups. We have added this information in the main text. However, higher mtDNA mutation load was proportional to more advanced age at diagnosis (Supplementary Figure 7d).

Page 10.

What is the expected proportion of mutations in ND5 and ND4?

ND5 and ND4 are both frequently mutated in colorectal cancer, but the reported mutation rates in CRC have varied between different cohorts. Specifically, the mutation rates were 15% and 11% in the ChangKang Chinese CRC cohort (Zhao et al. 2022 <https://doi.org/10.1038/s41467-022-30062-8>), 39% and 27% in PCAWG CRC (Yuan et al. 2020 <https://doi.org/10.1038/s41588-019-0557-x>), and 41% and 30% in our cohort. The differences in reported mutation rates may be attributed to variations in the indel calling capacity from different software, which may explain the discrepancies observed for these genes. It is noteworthy that the majority of truncating insertions and deletions identified in the mitochondrial genes here were found in ND5 and ND4 (Supplementary Table 14).

Compared with... actively treated patients. But these have been actively treated patients? As

such I have difficulties in seeing why this sample collection would be very different from many others.

We sought to primarily include patients with treatment-naïve tumors. As we mentioned in the main text, 126 (12%) tumors were surgical specimens from patients that received pre-treatment before surgery. Most of these are rectal cancers where pre-treatment is common practice to reduce tumor size before surgery, and where biopsy samples frequently are too small or preserved as FFPE rather than as fresh frozen materials.

Page 11

I presume that the mutations presented are independent prognostic factors?

Yes, the mutations presented in our study were evaluated as independent prognostic factors. To assess their prognostic impact, we performed multivariable Cox regression analysis which considered tumor site, pre-treatment status, tumor stage, tumor grade, age groups, and hypermutation status as covariates. We have updated the clinical data, and now have 5-year survival for 94% of the patients.

Page 12

"losses of 4p16.1, 4q34.1, 4q35.1" is it just "loss of chromosome 4"?

Of all 273 tumors with losses at these cytobands, 40% had lost all 3 cytobands while 60% had lost only 1 or 2 of 4p16.1, 4q34.1 or 4q35.1. Therefore, losses of 4p16.1, 4q34.1 and 4q35.1 do not necessarily mean loss of chromosome 4.

The tissue types present in CRC vs normal colon are somewhat different, having a big effect

on differentially expressed genes. Some discussion about this would be good. mRNA expression in genes that carry variants for nonsense mediated decay such as APC can be normal but due to the decay process the transcript is lost. Thus saying that such genes are underexpressed does not give an exact view of the underlying process.

Here, we used samples from healthy colorectal tissue from a subset of tumors as reference, which is common practice in such analyses. We acknowledge that nonsense mediated decay (NMD) is an important post-transcriptional regulatory mechanism that can influence the interpretation of mRNA expression data. In the context of CRC, mutations in genes like APC can lead to the generation of premature termination codons, which may trigger NMD and result in the degradation of the mutant mRNA. Consequently, it may appear that these genes are underexpressed, when in fact the underlying process is more complex due to NMD. While our analysis did not directly investigate the role of NMD, we addressed this shortcoming in the main text.

Page 14

Are tumors in CSPR 4 the ones which had been pretreated?

Indeed CRPS4 was the group with the highest proportion of pre-treated tumors, accounting for 37% (48 out of 130) of the samples compared to the other subgroups, which had rates of 7% (CRPS1), 11% (CRPS2), 5% (CRPS3) and 11% (CRPS5), respectively. To address potential bias introduced by these pre-treated samples, we excluded them in CRPS4 and reassessed cell populations in all CRPS groups. Despite the removal of the pre-treated samples, CRPS4 remained enriched with stromal cells and retained its previously observed biological features. We added the heatmaps in the updated Supplementary Figure 13b-d.

Page 15

Is hypoxia a proxy for size?

This is an insightful question, and we have also reflected over this possibility. Unfortunately, determining tumor size from biopsies or frozen surgical specimens is not possible and this information is not routinely incorporated in the clinical reports. A potential source for tumor size data is in the CT or MRI examinations. Although CT and MRI examinations may be performed in both colon and rectum, the latter examinations are predominantly performed in rectal and not colon cancers according to Swedish guidelines. The CT reports infrequently report maximum diameter whereas the tumor length is usually reported using MRI. Consequently, we could gather tumor size (length) data for 76% (215 out of 281) of all rectal cancer samples from the clinical records. In these cases, no statistically significant correlation between tumor size and hypoxia could be identified. Given these limitations in obtaining comprehensive and representative tumor size data for the entire CRC cohort, we opted not to include any analyses related to tumor size in our manuscript. However, we have

added the size information in Supplementary Table 1 as it can be a valuable resource for future investigations seeking this relationship.

Page 19

TGFBR2 mutations in MSI typically occur at a A 10 mononucleotide tract, and as such serve as a proxy of presence of MSI. Might the prognostic power arise merely from that?

To clarify, our analysis identified *TGFBR2* as prognostic in hypermutated (HM) tumors only, not in the combined cohort of non-hypermutated (nHM) and HM tumors. Following the clinical data update, *TGFBR2* was still prognostic for RFS in hypermutated (HM) tumors (HR=0.344, 95% CI: 0.190-0.624, $P=0.0004$; updated Supplementary Table 17) but not for OS. Thus, we cannot exactly confirm that the prognostic power of *TGFBR2* mutations in HM tumors is independent of MSI since a large part of this group constitutes MSI cases.

Figure 1 legend: FDR or p-value?

We have updated the figure legend and kept the correct annotation (FDR).

Figure 1c: Consider adding a barplot for total mutation count for each sample, possibly coloring by "top" pathway. Already from the image, it is clear that *POLE* mutants have much higher mutation count than others. Consider excluding *POLE* from the figure as there is little novelty there; the remaining samples have potential new findings.

We appreciate the suggestion and have included a barplot on top of Figure 1C, color-coded by the three DNA damage response pathways. However, we respectfully disagree with the proposal to exclude *POLE* mutants from the figure. While it is evident from the image that *POLE* mutants have one of the higher mutation counts (8/15 cases), when considering only the BER pathway genes, only 2 cases exclusively feature *POLE* mutations without any additional mutation in the pathway genes. Furthermore, *POLE* is a key gene within the BER pathway, making it a valuable inclusion in our figure.

Some cancers with a MMR gene mutation were MSS perhaps due to misclassification discussed above.

Carrying an MMR gene mutation may not necessarily result in a high rate of repeated homopolymers that can be detected by MSI sensor. As discussed above, we maintain confidence in the accuracy of our MSI classification process.

Supplementary Fig 9

Why different numbering in scales in expression left vs right per subfigure?

We have now modified Supplementary Figure 9 to ensure uniform numbering in both expression plots, enhancing the comprehensibility of the figure.

Supplementary figure 10 legend and in subsequent legends please avoid the expression "cancer tumours"

Thank you for noticing, we have changed the legends.

Supplementary Figure 11: CPRS: Somewhat unconventional use of single cell methods (Seurat, Celligner, scclusteval) for bulk RNA data. Why did you choose to use these?

Seurat, an R toolkit developed and maintained by the Satija Lab at NYCG, is indeed recognized for single-cell analysis. However, it is worth noting that Seurat can also be effectively applied to bulk RNA-seq analysis, as confirmed in discussions on its GitHub

repository (<https://github.com/satijalab/seurat/issues/826>). Furthermore, other tool like Celligner, a widely adopted computational tool for aligning bulk tumor data with cell line transcriptional profiles, utilized Seurat during its development (Warren et al. 2021 <https://doi.org/10.1038/s41467-020-20294-x>). Given Seurat's robust construction and its relevance in contemporary computational biology, we chose to leverage it in our study to define the CRPS subtypes.

Supplementary Figure 12 what is the difference between LOH and cn LOH (gain LOH and del having their own symbols).

As defined in the Supplementary Methods under the "Somatic structural variants and copy number variation" section, in our study, we grouped the CNVs into 8 distinct classes. These classes are defined by two parameters: total copy number (tcn) and minor copy number (lcn), as estimated by FACETS. The classification groups are as follows: wild type class (tcn=2, lcn=1), homozygous deletions (tcn=0, lcn=0), loss of heterozygosity (LOH; tcn=1, lcn=0), copy-neutral LOH (cn LOH; tcn=2, lcn=0), gain-LOH (tcn =3 or 4, lcn=0), gain (tcn =3 or 4, lcn ≥1), amp-LOH (tcn ≥5, lcn =0) and amp (tcn ≥5, lcn ≥1). In summary, LOH involves the loss of one copy of a chromosome, resulting in allelic imbalance, while cnLOH involves the duplication of one chromosome following by loss of the other allele.

Supplementary Figure 14 and Methods: "Model building and validation of CRPS classification"

What was in the input tensor? Guessing from the Supplementary Figure 14a, it was a list of the ssGSEA scores of the 2000 selected pathways in some unspecified order. Then the deep network enforced some dependency between the adjacent pathways (due to the convolution layer) and fitted the deep residual network with millions of parameters to the roughly 800 training samples.

The tensor input data included reshaped 1D vectors representing each gene set's row of samples (gs_1, gs_2, \dots, gs_n) into a 2D matrix ($1, n_{\text{features}}$). This transformed data served as the input tensor for the ResNet50 model, allowing it to capture complex patterns and dependencies in the genomic data. We divided the samples into 3 sets for training (80%), testing (10%) and validation (10%). The ResNet50 model was trained on the training set, using as input shape (None, -1, 2000), optimizing its deep residual network with millions of parameters. We have now included this information in more detail in the 'Model building and validation of CRPS classification' section of the Supplementary methods.

Overall, the deep learning model could have been described better, and, to the extent decipherable from the manuscript, it appears not well thought through and unnecessarily complicated, as compared with e.g. simple logistic regression.

Logistic regression models predict or output a probability of the observation falling into a certain category and is commonly used for solving binary classification problems. When it comes to a more complex classification (e.g. CRPS), we would like to interpret the non-linear relationships and emphasize feature selection, which can be fulfilled by other machine learning models. In addition, we have compared Resnet50 with other models, including linear models, support vector machines, decision trees, and ensemble models. Resnet 50 achieves ~85% accuracy, weighted precision, weighted recall, and weighted F1 score, which outperformed other models evaluated in terms of AUPRC and AUROC in the test set (10%

of cases, 119 samples).

Supplementary figure 17 what is the distance between the MSI branches after unsupervised clustering?

The Euclidean distances between the clusters of MSI subclasses in UMAP is 3.539, and the two classes are visually separated (Supplementary Figure 18). To statistically compare the distance between two subclasses, we compared distances between MSI class 1 samples and the centroid of MSI class 1 with distances between MSI class 2 samples and the centroid of MSI class 1, showing that MSI class 2 samples were further away from the MSI class 1 cluster, and vice versa.

How is it possible that some tumors are designated MSI but not HM? Appearance of MSI class 1 and 2, and not HM MSI probably relates to use of too low MSI sensor 2 threshold as discussed above.

As discussed previously, a cut-off value as 3.5 is correct for the MSI sensor2 paired module. Also, we defined HM status according to the total tumor mutation burden (TMB) of each sample and calculated based on the formula described in the Supplementary Methods under the “Identification of significantly mutated genes” section. It is possible that the TMBs of these specific MSI samples did not meet the criteria for HM status. Such variability in MSI and HM classification has been observed in other datasets as well, reinforcing the importance of considering both MSI and TMB measurements for a comprehensive characterization. In the AC-ICAM dataset they also identified 1 nHM sample in the 57 MSI-H samples (Roelands et al. 2023 <https://doi.org/10.1038/s41591-023-02324-5>; Supplementary Source Data 11).

Page 69

The patient selection appears unremarkable and thus seeing so many MSI tumors is quite unexpected, and not explained credibly in the manuscript.

As for MSI rate, we reported 21% MSI with MSIsensor cutoff ≥ 3.5 and 16.7% MSI if MSIsensor cutoff > 10 . The UK pre-print with 2,023 WGS CRC samples (Tomlinson et al. 2022 <https://doi.org/10.21203/rs.3.rs-2273265/v1>) included 18% of MSI cases in the full cohort and 19% in the CRC primary tumors only. The AC-ICAM cohort (Roelands et al. 2023 <https://doi.org/10.1038/s41591-023-02324-5>) of primary CRCs included 20.3% of MSI-H cases. As discussed above we believe MSIsensor cutoff of 3.5 represents the true MSI cases and the percentage is close to other large cohorts of primary CRC.

Page 78

Was RNA-seq successful for all tumors, or did you exclude some due to unsuccessful RNA-seq, ending with 1063?

The final cohort of 1,063 patients are cases for which we successfully obtained whole-genome sequencing data for normal and tumor samples as well as RNA sequencing data. There were 63 additional patient tumors that underwent sectioning and sequencing but were excluded due to lack of high-quality data in one or more of the three sequencing runs. In sum, all 1,063 patients included were successfully RNA sequenced. We added this information in the main text under the "Mutational Landscape" section.

Page 81

"Batch size" perhaps rather say "number of output classes"?

We agree with the suggestion of the reviewer and have changed the Supplementary Methods section accordingly.

Page 84 Data availability

Is the idea that case by case raw data will be accessible, from U-CAN? Any restrictions? This is an important matter. On one hand the reported data is a key resource for the community and access would be very welcome. On the other, these are old samples and consents probably have not covered data sharing and thus sharing after introduction of GDPR may not be possible - depending on local interpretation of GDPR, and other local rules and regulations. If the raw case by case data from these old samples in practice cannot be shared that would be perfectly understandable and should not as such lead to rejection of this work, but any possibly existing restrictions should be described. As a minimum it might be wise to insert a disclaimer such as raw data will be shared whenever allowed by GDPR and local rules and regulations, or similar, to flag that transferring the data to countries with e.g. poor data protection landscape (or anywhere) might not be possible. To summarize, please consider if there is a need to revise the data availability statement.

We appreciate the concerns about data availability. While we recognize the challenges posed by regulations including GDPR and national law, we are committed to adhering to all ethical and legal guidelines for sharing the sensitive raw data. To enhance transparency, we have updated our data availability statement to include a disclaimer acknowledging potential constraints arising from the regulations.

“Short somatic variant calls, copy number variation and structural variants data can be accessed at the European Variation Archive with accession number PRJEB61514, expression profiles at the ArrayExpress with accession number E-MTAB-12862, or all data at CNGB Sequence Archive (CNSA) of China National GeneBank DataBase (CNGBdb) with accession number CNP0004160. Raw data and more detailed clinical information are deposited at Uppsala University and inquires to access them should be directed to U-CAN, a cancer biobank at Uppsala University (<https://www.u-can.uu.se>). Access to raw data and clinical information is subject to Swedish legal regulations, GDPR, permission from the Swedish Ethical Review Authority, and U-CAN terms. The remaining data are available within the Article, Supplementary Information or available from the authors upon request.”

For immediate access to the analyzed data, we provide here a public link to the DNA sequencing output, and a temporary link to the RNA sequencing output data that will be made public at paper publication: genomic data, encompassing short variants, structural variants, and copy number variation, can be publicly accessed through this link: [\[https://www.ebi.ac.uk/eva/?eva-study=PRJEB61514\]](https://www.ebi.ac.uk/eva/?eva-study=PRJEB61514); RNA expression data can be accessed via this temporary link: <https://www.ebi.ac.uk/biostudies/arrayexpress/studies/E-MTAB-12862?key=1ab5c1b6-0d2a-4c80-a2a2-7c66a707844b>. We hope that this revised statement adequately addresses your data availability concerns.

Regarding the raw data, due to the substantial size of the DNA raw data (~300 TB), current limitations prevent their inclusion in the controlled European Genome-phenome ArchiveEGA database (EGA). As an alternative, individuals interested in accessing this data are encouraged to contact U-CAN as indicated in the manuscript. U-CAN will assist in providing the data upon fulfilling legal and ethical permit requirements. Regarding the RNA raw data, although efforts are underway to upload it to the EGA database, since it is substantial smaller, technical issues with the web portal hindered its upload before the deadline of this initial revision. Rest assured, we are currently actively engaged with EGA support to resolve these issues promptly, aiming to make the RNA raw data available soon.

Reviewer Reports on the First Revision:

Referee #1 (Remarks to the Author):

I appreciate the efforts made by the authors. I agree with the editor and the other reviewers that data availability is a key issue for this manuscript. Beyond that the analyses of the data require some further details in its presentation and information content. Specifically, the below should be addressed in order for this manuscript (and its supplementary content/ analyses) to be a resource to the field as well as to provide for a comprehensive reliable description and analyses of the data.

1. Failure/Inadequate description of diversity of MSS (nHM) CRC:

The authors responded " To this end, we have now classified these tumors according to CMS, iCMS and CRPS, and decomposed prognostic features by each subtype "

nHM is clearly not a homogenous subgroup, the CMS, iCMS and proposed CRPS subtypes are the relevant subgroups where each group either has similar bulk gene expression, malignant tumor expression or prognosis.

Thus, what is missing is not really the decomposed prognostic feature within each subtype but rather the driver alterations and mutational frequencies within each of these important subtypes (CMS1-4, iCMS2 nHM, iCMS3 nHM and iCMS HM and CRPS 1-5.

This information is informative to the field. Could the authors please add additional columns to Supp Table 3, To Columns E to G, add additional columns for each of the subtypes for driver status, and for the mutated cases after column S for each of the subtypes. Please also add additional rows to for additional drivers to cover the union of all drivers across CMS1-4, 3 iCMS classes and 5 CRPS , beyond the current set of 103 driver genes.

The effort the authors made in supp table 5 is well appreciated.

2. Cohort bias: Kindly provide information on the right and left colon status in this cohort and the Swedish resected patients

3. Yet another gene-expression classifier: For the prognostic effect, given what is known about the different effects on relapse free survival and survival after relapse for CMS & iCMS, please comment on whether this is similar for CRPS and provide a supplementary table on the relapse free survival and survival after relapse for CRPS. If these HRs are informative kindly provide a figure or supplementary figure as well.

4. I appreciate the attempt at a new figure 4. Unfortunately, it is really difficult to read and important pieces of information do not easily get conveyed and information content is lost by the binary classification just by direction (rather than magnitude) as either $\log_2FC > 0$ or < 0 for altered vs non-altered.

For each gene 6 columns are presented for mutations and copy number (CNV) loss and gain for each gene in non-hypermuted (nHM) and hypermutated (HM) tumours. Have each column have the color intensity by log fold change of altered vs nonaltered , have a black line within each column to represent the % altered. In this way the quantitative information of % altered and magnitude of log fold gene expression differences between altered and not altered can be displayed intuitively.

Referee #2 (Remarks to the Author):

Overall, the revisions to the manuscript by Nunes et al. have improved the study significantly. This reviewer is glad to see that the de-identified variant data have been made available, which will make this data set more valuable for the community. The manuscript also contains many new analyses, and existing analyses have been made clearer. However, some concerns remain, mostly about the focus of the manuscript, the presentation of the data, and the relative lack of findings that would significantly change our understanding of colorectal cancer.

Specific remaining concerns:

1. The manuscript claims to have identified many novel genes important in colorectal cancer, some of them never described as significant in any cancer. However, all of these novel genes were identified in hypermutated tumors, raising concerns about the validity of the findings. Furthermore, only three significantly recurrently mutated genes were identified in the non-hypermutated tumors. A larger number of significantly mutated genes should have been identified in these tumors, raising concerns about the method applied.

2. Overall the figures are a bit overwhelming with the amount of content, and they are difficult to interpret given the color choices and sometimes small font size.

The POLE and POLD1 mutations shown in Figure 1b seem to be all mutations observed in these two genes, including passenger variants / VUS. This figure would be much more informative if only the known driver mutations were shown, as they would likely be mutually exclusive with the MSI-H cases. Furthermore, this reviewer is curious whether there are any POLD1 somatic driver mutations at all?

3. The mutation diagrams for POLE and POLD1 shown in the rebuttal seem to only show the subset of mutations identified in the MSS tumors, and none of them are known drivers. This figure would be more informative if it showed the POLE mutations observed in the hypermutated cases.

4. If the outcome results cannot be replicated in three separate individual external studies (point 12 in the rebuttal), can the authors comment on how robust their findings are?

Referee #3 (Remarks to the Author):

Nunes et al. have revised their manuscript reporting genome and transcriptome wide analysis of 1063 CRCs.

Re data sharing it is stated (manuscript) that

“Raw data and more detailed clinical information are 1097 deposited at Uppsala University and inquires to access them should be directed to U-CAN, a 1098 cancer biobank at Uppsala University (<https://www.u-can.uu.se>). Access to raw data and 1099 clinical information is subject to Swedish legal regulations, GDPR, permission from the 1100 Swedish Ethical Review Authority, and U-CAN terms.”

In the (rebuttal)

“Rest assured, we are currently actively engaged with EGA support to resolve these issues promptly, aiming to make the RNA raw data available soon.”

And indeed this was successful, and the data has been submitted to EGA - though 24 items to follow greet anyone willing to have access. I guess that the data availability statement will be rephrased in this new setting. As mentioned in my previous comments it is on one hand very welcome that data can be shared, and on the other, sharing data (such as RNA seq raw reads) with ID information should not be a prerequisite for publication. So, this referee does not mind if in practice the procedures are (too) complicated. However, if in practice the data sharing procedure will be unreasonably laborious to those who request it, it might be best for everyone to just say it is not available. Perhaps the Editor can guide the authors here.

“Our findings revealed POLE and POLD1 mutations in 16 and 6 nHM samples, respectively, each exhibiting 4 cases with exonuclease mutations, totalling 8 cases.”

I do find this unexpected. POLE exonuclease mutations should lead to an extreme HM phenotype. Something is not right here.

“Here, we used samples from healthy colorectal tissue from a subset of tumors as reference, which is common practice in such analyses.”

I fully agree. But my comment was: “The tissue types present in CRC vs normal colon are somewhat different, having a big effect on differentially expressed genes. Some discussion about this would be good.” That others have done the same just makes the point even more valid. I really think that people just tend to close their eyes in this context because there is no easy fix. However, why not openly state that there is this challenge in creating the normal colon expression data.

Original comment: TGFBR2 mutations in MSI typically occur at a A 10 mononucleotide tract, and as such serve as a proxy of presence of MSI. Might the prognostic power arise merely from that? "To clarify, our analysis identified TGFBR2 as prognostic in hypermutated (HM) tumors only, not in the combined cohort of non-hypermutated (nHM) and HM tumors. Following the clinical data update, TGFBR2 was still prognostic for RFS in hypermutated (HM) tumors (HR=0.344, 95% CI: 0.190-0.624, P=0.0004; updated Supplementary Table 17) but not for OS. Thus, we cannot exactly confirm that the prognostic power of TGFBR2 mutations in HM tumors is independent of MSI since a large part of this group constitutes MSI cases."

It could also be that when an MSI cancer grows slowly (e.g. lot's of cell death) it has the time to create a lot of genetic diversity (many mutations), and thus high mutation count – as a proxy of slow growth – could be an indicator of better prognosis. Any correlation in your data?

On page 3 in the introduction "Approximately 80-85% of CRCs are classified as copy-number altered microsatellite stable (MSS), 10-16% as highly mutated tumours with microsatellite instability (MSI), and 1-2% as ultra-mutated tumours resulting from somatic POLE mutations. " Perhaps add references, in particular as the results of the current paper are somewhat different.

Figure 1c: "Consider adding a barplot for total mutation count for each sample,..." This referred to *total mutation count*, i.e. the total number of somatic mutations in the tumor; a number in millions for POLE mutant tumors, 100s of thousands for MSI, and 10s of thousands for MSS. The purpose is to show that POLE mutant tumors have mutations in just about every gene just by having so many mutations all over the genome.

The discrepancy of MSI proportion in earlier cohorts and this work, along with ambiguity of MSIsensor threshold is well explained in the rebuttal but should be improved in the manuscript main text. E.g. Page 4, around lines 80-81 would benefit from a few soothing words about consistency with the quality register and a reference to Supplementary Table 15

Software for CRPS classification is slightly buggy (tmp/ directory needs to be created manually for each input. Software requires fairly specific python version (runs with 3.8, does not with 3.10)) and is not too user friendly (User is left on her/his own for running ssGSEA and producing the appropriate input format for the classification tool. The classification tool returns false data with no warning to user when given bad input.)

Referee #3 (Remarks on code availability):

I was able to run the code with the model input provided. Formatting input for users own expression data is not included in the shared code.

Referee #4 (Remarks to Author):

Nature is committed to facilitate training in peer-review and to ensure that everyone involved in our peer review process is appropriately recognised. This reviewer co-reviewed one of the listed reports.

Author Rebuttals to First Revision:

Referee #1 (Remarks to the Author):

I appreciate the efforts made by the authors. I agree with the editor and the other reviewers that data availability is a key issue for this manuscript. Beyond that the analyses of the data require some further details in its presentation and information content. Specifically, the below should be addressed in order for this manuscript (and its supplementary content/ analyses) to be a resource to the field as well as to provide for a comprehensive reliable description and analyses of the data.

1. Failure/Inadequate description of diversity of MSS (nHM) CRC:

The authors responded " To this end, we have now classified these tumors according to CMS, iCMS and CRPS, and decomposed prognostic features by each subtype "

nHM is clearly not a homogenous subgroup, the CMS, iCMS and proposed CRPS subtypes are the relevant subgroups where each group either has similar bulk gene expression, malignant tumor expression or prognosis.

Thus, what is missing is not really the decomposed prognostic feature within each subtype but rather the driver alterations and mutational frequencies within each of these important subtypes (CMS1-4, iCMS2 nHM, iCMS3 nHM and iCMS HM and CRPS 1-5).

This information is informative to the field. Could the authors please add additional columns to Supp Table 3, To Columns E to G, add additional columns for each of the subtypes for driver status, and for the mutated cases after column S for each of the subtypes. Please also add additional rows to for additional drivers to cover the union of all drivers across CMS1-4, 3 iCMS classes and 5 CRPS , beyond the current set of 103 driver genes.

The effort the authors made in supp table 5 is well appreciated.

Regarding the number of driver genes, we want to clarify that the initial set of driver genes was 96, not 103. We recognize that the Excel table doesn't start until row 9, leading to a potential misunderstanding. Further, after conducting the analysis based on CMS, CRPS, and iCMS subtypes, we identified 13 new significantly mutated genes that were not previously identified when we used the full cohort and hypermutation status. These new genes exhibit subtype specificity, with mutation prevalence <10% in the full cohort. Overall, 8 out of these 13 genes are previously established cancer genes, and 3 of the 8 have been previously identified as significantly mutated genes in colorectal cancer in other studies (FZD3, JUN, and USP9X). Thank you for the positive feedback on the previous answer, and we have added the requested information in Supplementary Table 3 as suggested.

2. Cohort bias: Kindly provide information on the right and left colon status in this cohort and the Swedish resected patients

In order to provide comprehensive information on the right and left colon status for the Swedish resected patients, we have analyzed the Swedish Colorectal Cancer Registry (SCRCCR), as we did previously for the other clinical information (age, sex, histology, grade, treatment, metastases and survival). We have now included the requested numbers in Extended Data Table 2 (previously termed Supplementary Table 15).

3. Yet another gene-expression classifier: For the prognostic effect, given what is known about the different effects on relapse free survival and survival after relapse for CMS & iCMS, please comment on whether this is similar for CRPS and provide a supplementary table on the relapse free survival and survival after relapse for CRPS. If these HRs are informative kindly provide a figure or supplementary figure as well.

In our response to the comment, we referenced the previously plotted recurrence-free survival in main Figure 5c. As previously stated, CMS recurrence-free survival in our cohort was not statistically significant. However, a consistent trend was observed, between CMS and CRPS with reduced recurrence-free survival in CMS3 and CMS4, corresponding to CRPS4 and CRPS5 also having the shortest survival. We have now incorporated information on survival after recurrence for CRPS, CMS, and iCMS in Supplementary Figure 8, which is cited in the main text. Our results show that CRPS1 and the corresponding CMS1 are associated with the shortest survival after recurrence, consistent with findings from the original CMS subtyping paper (Nat Med. 2015 Nov; 21(11):1350-6. doi: 10.1038/nm.3967). We have also added in the methods the definition for recurrence after survival as the time from recurrence to death.

4. I appreciate the attempt at a new figure 4. Unfortunately, it is really difficult to read and important pieces of information do not easily get conveyed and information content is lost by the binary classification just by direction (rather than magnitude) as either $\log_2FC > 0$ or < 0 for altered vs non-altered.

For each gene 6 columns are presented for mutations and copy number (CNV) loss and gain for each gene in non-hypermuted (nHM) and hypermutated (HM) tumours. Have each column have the color intensity by log fold change of altered vs nonaltered, have a black line within each column to represent the % altered. In this way the quantitative information of % altered and magnitude of log fold gene expression differences between altered and not altered can be displayed intuitively.

We acknowledged your comments on the clarity and readability of Figure 4 and have made efforts to address them. We have now updated the red/blue colour scheme to represent $\log_2FC > 0$ and < 0 , respectively, for altered and wild-type samples. Additionally, we have added a black line within each column to visually indicate the percentage of altered somatic alterations, as suggested. We hope that these changes align with your comment and improve the overall clarity of the figure.

Referee #2 (Remarks to the Author):

Overall, the revisions to the manuscript by Nunes et al. have improved the study significantly. This reviewer is glad to see that the de-identified variant data have been made available, which will make this data set more valuable for the community. The manuscript also contains many new analyses, and existing analyses have been made clearer. However, some concerns remain, mostly about the focus of the manuscript, the presentation of the data, and the relative lack of findings that would significantly change our understanding of colorectal cancer.

Specific remaining concerns:

1. The manuscript claims to have identified many novel genes important in colorectal cancer, some of them never described as significant in any cancer. However, all of these novel genes were identified in hypermutated tumors, raising concerns about the validity of the findings. Furthermore, only three significantly recurrently mutated genes were identified in the non-hypermutated tumors. A larger number of significantly mutated genes should have been identified in these tumors, raising concerns about the method applied.

It appears there might be a misunderstanding regarding the identification of novel genes in our manuscript. The main Figure 1a is not presenting the results of driver identification; instead, it illustrates the association between the identified drivers and various clinical and genomic features, including survival. As you correctly noted, in Figure 1a, the top three genes are significantly more associated with non-hypermutated tumours, while most show significant associations with hypermutated tumours, and a few with both. However, we direct your attention to Supplementary Table 3, specifically the column labelled "Driver gene status", where we present the information regarding driver gene identification. According to the information in this table, among the total 96 identified drivers, 37 were from hypermutated tumours, while 65 were from non-hypermutated tumours. Thus, most driver genes were identified in non-hypermutated tumours. The observed higher mutation frequency in hypermutated tumours makes that most of these drivers to be more significantly associated with this group, but they were not necessarily identified as driver genes in that subset. We hope this clarification addresses your concerns and provides a more accurate understanding of our findings

2. Overall the figures are a bit overwhelming with the amount of content, and they are difficult to interpret given the color choices and sometimes small font size.

The POLE and POLD1 mutations shown in Figure 1b seem to be all mutations observed in these two genes, including passenger variants / VUS. This figure would be much more informative if only the known driver mutations were shown, as they

would likely be mutually exclusive with the MSI-H cases. Furthermore, this reviewer is curious whether there are any *POLD1* somatic driver mutations at all?

We have revised Figure 1b as per your suggestion, focusing only on known driver mutations in *POLE* and *POLD1*. The updated figure shows that only very few *POLE* mutations are drivers, and all *POLD1* mutations are passengers. It is clear now that driver *POLE* mutations were associated with part of the hypermutated MSS tumours, explaining the elevated mutation burden in those cases, and mutually exclusive with the MSI cases. We believe that these changes align with your suggestion and facilitate the interpretation of Figure 1b.

3. The mutation diagrams for *POLE* and *POLD1* shown in the rebuttal seem to only show the subset of mutations identified in the MSS tumors, and none of them are known drivers. This figure would be more informative if it showed the *POLE* mutations observed in the hypermutated cases.

The lollipop plots for *POLE* and *POLD1* have been revised based on known driver mutations, according to Cancer Hotspots and OncoKB. Among the known *POLE* driver mutations only 5 (P286R, S297F, V411L, M444K and A456P) were found in 6 hypermutated MSS samples, confirming their mutual exclusivity with MSI cases (as stated above). In our MSS samples, there were no somatic *POLD1* driver mutations identified. In MSI samples, two mutations from *POLD1* (X1013_SpliceSite and P116Hfs*53) were potential hotspots, each found in 15 MSI samples, with no overlap between the two. Moreover, the tumour mutation burden (TMB) in these MSI samples with *POLD1* mutations was higher than those without, and samples carrying

POLD1 “hotspot” X1013_splice exhibited even higher TMB.

4. If the outcome results cannot be replicated in three separate individual external studies (point 12 in the rebuttal), can the authors comment on how robust their findings are?

The robustness of our findings is evident in the consistent overall survival prognostic effect demonstrated by CRPS subtyping in our cohort as well as the combined external cohorts. Despite the challenges of replication in three separate individual external studies, it's crucial to highlight that the differences in cohort sizes, population representation, and limited follow-up data in these external cohorts may have contributed to the lack of statistical significance when considered separately. We acknowledge the importance of future large cohorts with complete follow-up for proper validation of our findings. Despite these challenges, we believe our preliminary validation with the currently available data online supports the robustness of our model. The consistently significant outcomes in our cohort and the combined external cohorts provide valuable insights, and we remain open to future validation as more comprehensive datasets become available. We have now addressed this in the discussion section of the manuscript.

Referee #3 (Remarks to the Author):

Nunes et al. have revised their manuscript reporting genome and transcriptome wide analysis of 1063 CRCs.

Re data sharing it is stated (manuscript) that

“Raw data and more detailed clinical information are deposited at Uppsala University and inquires to access them should be directed to U-CAN, a cancer biobank at Uppsala University (<https://www.u-can.uu.se>). Access to raw data and clinical information is subject to Swedish legal regulations, GDPR, permission from the Swedish Ethical Review Authority, and U-CAN terms.”

In the (rebuttal)

“Rest assured, we are currently actively engaged with EGA support to resolve these issues promptly, aiming to make the RNA raw data available soon.”

And indeed this was successful, and the data has been submitted to EGA - though 24 items to follow greet anyone willing to have access. I guess that the data availability statement will be rephrased in this new setting. As mentioned in my previous comments it is on one hand very welcome that data can be shared, and on the other, sharing data (such as RNA seq raw reads) with ID information should not be a prerequisite for publication. So, this referee does not mind if in practice the procedures are (too) complicated. However, if in practice the data sharing procedure will be unreasonably laborious to those who request it, it might be best for everyone to just say it is not available. Perhaps the Editor can guide the authors here.

We have revised the data availability statement to include information about the raw transcriptome data submission to EGA. Regarding the accessibility of raw data, particularly raw data deposited at EGA, we want to emphasize that our intent was to balance between making the data more easily transferable while maintaining a controlled access framework. The 24 items associated with EGA access were generated following the Harmonised Data Access Agreement (hDAA) for Controlled Access Data, a European standardization framework for data integration and data-driven *in silico* models for personalized medicine. This framework helps maintain a controlled and standardized approach to data access, ensuring that requesters follow proper ethical considerations and compliance with established regulations. For the whole-genome sequencing raw data, given its substantial size, interested parties can request access through the corresponding author, who, in collaboration with the U-CAN biobank, will facilitate transfer or provide access under controlled agreements. We believe that our approach aligns with standard practices for controlled access to sensitive data, ensuring responsible and ethical sharing. We have aimed to share a substantial amount of data, including open access to detailed clinical data and analysed genomic and transcriptomic data. We appreciate your

understanding and trust that our approach to data sharing aligns with both your expectations and the guidelines of the journal.

“Our findings revealed *POLE* and *POLD1* mutations in 16 and 6 nHM samples, respectively, each exhibiting 4 cases with exonuclease mutations, totalling 8 cases.”

I do find this unexpected. *POLE* exonuclease mutations should lead to an extreme HM phenotype. Something is not right here.

Following the suggestion of reviewer 2, we have now revised Figure 1b focusing only on known driver mutations in *POLE* and *POLD1*. The updated figure shows that only a very few *POLE* mutations are in fact drivers, and all *POLD1* mutations are passengers. It is clear now that driver *POLE* mutations were associated with part of the hypermutated MSS tumours, explaining the elevated mutation burden in those cases, and mutually exclusive with the hypermutated/MSI cases. This explains why *POLE* mutations in hypermutated samples do not lead to an extreme phenotype.

“Here, we used samples from healthy colorectal tissue from a subset of tumors as reference, which is common practice in such analyses.”

I fully agree. But my comment was: “The tissue types present in CRC vs normal colon are somewhat different, having a big effect on differentially expressed genes. Some discussion about this would be good.” That others have done the same just makes the point even more valid. I really think that people just tend to close their eyes in this context because there is no easy fix. However, why not openly state that there is this challenge in creating the normal colon expression data.

We apologize for any previous misunderstanding. In response, we have now added a statement in the Methods section to address the rationale behind utilizing bulk tumour and small pool of normal tissue RNA sequencing, and the potential problems arising in the analyses. We hope this clarification addresses your concern.

Original comment: TGFBR2 mutations in MSI typically occur at a A 10 mononucleotide tract, and as such serve as a proxy of presence of MSI. Might the prognostic power arise merely from that?

“To clarify, our analysis identified TGFBR2 as prognostic in hypermutated (HM) tumors only, not in the combined cohort of non-hypermutated (nHM) and HM tumors. Following the clinical data update, TGFBR2 was still prognostic for RFS in hypermutated (HM) tumors (HR=0.344, 95% CI: 0.190-0.624, P=0.0004; updated Supplementary Table 17) but not for OS. Thus, we cannot exactly confirm that the prognostic power of TGFBR2 mutations in HM tumors is independent of MSI since a large part of this group constitutes MSI cases.”

It could also be that when an MSI cancer grows slowly (e.g. lot's of cell death) it has the time to create a lot of genetic diversity (many mutations), and thus high mutation count – as a proxy of slow growth – could be an indicator of better prognosis. Any correlation in your data?

We have now investigated the association of *TGFBR2* with MSI and tumour mutation burden (TMB) deeper. A specific *TGFBR2* hotspot mutation, c.458del in our MSI samples, was correlated with higher tumour mutational burden (TMB) compared to other *TGFBR2* mutants or wild-type as illustrated in the box plot below. However, despite this correlation, the independent prognostic power of *TGFBR2* mutations in recurrence-free survival for our MSI samples remains unaffected, as indicated in the forest plot below. This shows that the observed survival differences associated with *TGFBR2* mutations are not solely dependent on TMB. Further exploration of the underlying mechanisms and functional consequences of *TGFBR2* mutations in the context of mismatch repair deficiency is of great interest.

SurvivalSurv(RFS_5Years_with_censored, RFS_Status_5Years_with_censored)

TGFBR2.Status.1	WT	-
	c.458del.Mut	0.42 (0.21–0.84, p=0.014)
	other.Mut	0.21 (0.07–0.61, p=0.004)
TMB.Status	HM	-
	nHM	2.74 (0.34–21.90, p=0.342)
Tumor_Site	Left_Colon	-
	Rectum	1.59 (0.10–24.98, p=0.742)
	Right_Colon	0.91 (0.38–2.20, p=0.834)
Tumor_Stage	I	-
	II	1.19 (0.44–3.23, p=0.728)
	III	1.69 (0.58–4.94, p=0.336)
Age_Type	...65 years	-
	...80 years	4.61 (1.30–16.39, p=0.018)
	66–79 years	1.46 (0.39–5.47, p=0.572)
Tumor_Grade	High-Grade	-
	Low-Grade	0.46 (0.24–0.86, p=0.015)
Before_Surgery	Treatment	-
	Untreatment	0.76 (0.12–4.93, p=0.771)

On page 3 in the introduction "Approximately 80-85% of CRCs are classified as copy-number altered microsatellite stable (MSS), 10-16% as highly mutated tumours with microsatellite instability (MSI), and 1-2% as ultra-mutated tumours resulting from somatic POLE mutations. " Perhaps add references, in particular as the results of the current paper are somewhat different.

References have now been added to the statement on page 3 of the introduction, providing support for the mentioned percentages.

Figure 1c: "Consider adding a barplot for total mutation count for each sample,..." This referred to *total mutation count*, i.e. the total number of somatic mutations in the tumor; a number in millions for POLE mutant tumors, 100s of thousands for MSI, and 10s of thousands for MSS. The purpose is to show that POLE mutant tumors have mutations in just about every gene just by having so many mutations all over the genome.

We have implemented the suggested modification to Figure 1c, incorporating a barplot for total mutation count for each sample. The updated figure shows that *POLE* mutant tumours exhibit mutations in several genes involved in various pathways, highlighting the broad impact of their elevated mutation burden on the genome.

The discrepancy of MSI proportion in earlier cohorts and this work, along with ambiguity of MSI sensor threshold is well explained in the rebuttal but should be improved in the manuscript main text. E.g. Page 4, around lines 80-81 would benefit from a few soothing words about consistency with the quality register and a reference to Supplementary Table 15

To address this concern, we have included a brief mention on page 4 emphasizing data consistency with the clinical registers and directing readers to Extended Data Table 1 (old Supplementary Table 15).

Software for CRPS classification is slightly buggy (tmp/ directory needs to be created manually for each input. Software requires fairly specific python version (runs with 3.8, does not with 3.10)) and is not too user friendly (User is left on her/his own for running ssGSEA and producing the appropriate input format for the classification tool. The classification tool returns false data with no warning to user when given bad input.)

We appreciate the feedback on the software and we have now tried to make the necessary adjustments to address the reported issues. The software now automatically creates the "tmp/" directory for each input, eliminating the need for manual intervention. We have provided more detailed instructions and assistance for running ssGSEA and preparing the input format for the classification tool. Users can now either use their own expression data (will be processed by the built-in ssGSEA module), or self-prepared pathway enrich scores as input. If Python versions are causing problems for using the tool, the user may create an "issue" on GitHub, and we can provide a Docker image of CRPS classification tool for downloading. Moreover, we have implemented validation checks to detect and notify users of potential issues with input data, minimizing the risk of false results without proper

warning. We have updated the changes of the tool and instructions on formatting input in README on GitHub. We hope these improvements enhance the overall user experience and we value your continued input.

Referee #3 (Remarks on code availability):

I was able to run the code with the model input provided. Formatting input for users own expression data is not included in the shared code.

We really appreciate the feedback on the software and we have tried to make the necessary adjustments to address the reported issue.

Referee #4 (Remarks to Author):

Nature is committed to facilitate training in peer-review and to ensure that everyone involved in our peer review process is appropriately recognised. This reviewer co-reviewed one of the listed reports.

Reviewer Reports on the Second Revision:

Referee #1 (Remarks to the Author):

I thank the authors for the revisions and my concerns are addressed.

However, I do have a specific request to improve the utility of the manuscript.

I appreciate the attempt to improve the visualisation of the data in figure 4. Still, whilst condensing the data is good for broad overview and observing trends, the underlying "raw"/analysed data is still important. I would request for a supplementary table to specifically accompany figure 4

For the reader's convenience, For each of the genes and for each of the 6 tracks: mutations and copy number (CNV) loss and gain for each gene in non-hypermuted (nHM) and hypermutated (HM) tumours:

provide the frequency of somatic alteration, the log FC between in mutated vs and wild-type tumours, FDR value and whether it is a driver gene

This would better facilitate understanding and information dissemination by and to readers. Given that these changes are straightforward to implement, I am comfortable proceeding with the manuscript's acceptance without the need for further review (i.e. editorially reviewed). However, I would still like to receive the amended table via email to examine the trends and understand the relationships between gene alterations and expression changes in altered/wildtype for my own understanding/comprehension of these changes.

Referee #2 (Remarks to the Author):

The authors have mostly addressed this reviewer's concerns.

Just one issue remains: The clarifications on the driver vs passenger distinction in POLE and POLD1 mutations are very helpful, but it is now clearer that Fig. 1C suffers from the same problem, making this figure very misleading. It shows mutations in many DNA repair related genes and pathways, with not much of a distinction between the type of mutations (mutation type is shown, but not known driver status). Given the high background mutation rate in these tumors, it is expected that many of these genes are affected by random (likely passenger) mutations. In how many of these non-MSI hypermutated tumors can the high mutation burden really be explained? Maybe just 7 - those with the POLE missense mutations (assuming these are the driver mutations)? The POLE truncating mutation is not expected to cause hypermutation. Also, the MMR mutations at the bottom of the panel, certainly MLH1 mutations, would be expected to cause an MSI phenotype.

This is all to suggest that this part of the manuscript could be improved to avoid misleading readers of the manuscript.

Referee #3 (Remarks to the Author):

Regarding the data availability a word of caution:

“As of now, our published data includes complete open access to comprehensive datasets, encompassing complete clinical data, coding and noncoding mutations, complete structural variation, copy number variation, and analysed RNA expression data.”

Even a stringent filtering often fails to completely remove rare germline variation – the kind that has the biggest power to identify individuals.

Also, treating raw RNA-seq data and WGS data differently in view of accession appears counter intuitive, as much of the germline exonic variation can be derived also from the former.

We are pleased that the authors have given effort on the usability of the CRPS classification tool.

Figure 1c has been improved nicely.

Author Rebuttals to Second Revision:

Responses to Referees' comments

Referee #1 (Remarks to the Author):

I thank the authors for the revisions and my concerns are addressed. However, I do have a specific request to improve the utility of the manuscript. I appreciate the attempt to improve the visualisation of the data in figure 4. Still, whilst condensing the data is good for broad overview and observing trends, the underlying "raw"/analysed data is still important. I would request for a supplementary table to specifically accompany figure 4

For the reader's convenience, For each of the genes and for each of the 6 tracks: mutations and copy number (CNV) loss and gain for each gene in non-hypermuted (nHM) and hypermutated (HM) tumours: provide the frequency of somatic alteration, the log FC between in mutated vs and wild-type tumours, FDR value and whether it is a driver gene

This would better facilitate understanding and information dissemination by and to readers. Given that these changes are straightforward to implement, I am comfortable proceeding with the manuscript's acceptance without the need for further review (i.e. editorially reviewed). However, I would still like to receive the amended table via email to examine the trends and understand the relationships between gene alterations and expression changes in altered/wildtype for my own understanding/comprehension of these changes.

We would like to thank the Referee for the constructive feedback and for acknowledging the revisions we have made. We have addressed the suggestion and have now added Supplementary Table 24, which includes the underlying data for Figure 4 (now Figure 3).

Referee #2 (Remarks to the Author):

The authors have mostly addressed this reviewer's concerns.

Just one issue remains: The clarifications on the driver vs passenger distinction in POLE and POLD1 mutations are very helpful, but it is now clearer that Fig. 1C suffers from the same problem, making this figure very misleading. It shows mutations in many DNA repair related genes and pathways, with not much of a distinction between the type of mutations (mutation type is shown, but not known driver status). Given the high background mutation rate in these tumors, it is expected that many of these genes are affected by random (likely passenger) mutations. In how many of these non-MSI hypermutated tumors can the high mutation burden really be explained? Maybe just 7 - those with the POLE missense mutations (assuming these are the driver mutations)? The POLE truncating mutation is not expected to cause hypermutation. Also, the MMR mutations at the bottom of the panel, certainly MLH1 mutations, would be expected to cause an MSI phenotype.

This is all to suggest that this part of the manuscript could be improved to avoid misleading readers of the manuscript.

We would like to thank the Referee for the constructive feedback and for pointing out the potential for misinterpretation in Figure 1c regarding driver versus passenger mutations in DNA repair genes and pathways. We understand the concern about the high background mutation rate in hypermutated tumors and the necessity to clearly distinguish between driver and passenger mutations. To address this issue and avoid misleading readers, we have included cautionary notes in both the main text and figure legend.

Referee #3 (Remarks to the Author):

Regarding the data availability a word of caution:

“As of now, our published data includes complete open access to comprehensive datasets, encompassing complete clinical data, coding and noncoding mutations, complete structural variation, copy number variation, and analysed RNA expression data.”

Even a stringent filtering often fails to completely remove rare germline variation – the kind that has the biggest power to identify individuals.

Also, treating raw RNA-seq data and WGS data differently in view of accession appears counter intuitive, as much of the germline exonic variation can be derived also from the former.

We are pleased that the authors have given effort on the usability of the CRPS classification tool.

Figure 1c has been improved nicely.

We would like to thank the Referee for the constructive feedback and for acknowledging the improvements made to Figure 1c and the classification tool. Regarding the concerns about data availability and access, we would like to provide some clarifications. Both RNA-seq and whole-genome sequencing (WGS) raw data were treated equally regarding access, as they are not available through open access to ensure compliance with privacy and data protection regulations. For RNA-seq raw data, we have uploaded these to the European Genome-phenome Archive (EGA). This allows for researchers to request access to the data more easily while still maintaining controlled access to protect individual privacy. Due to the large size of the WGS raw data, we have encountered challenges in uploading the dataset to EGA. However, researchers can still request access to the WGS data through the corresponding author. We are committed to facilitating access to these datasets and ensuring that all requests are processed in a timely manner while protecting their integrity and fulfill requirements in an evolving legal landscape. We have updated the data availability section and hope these clarifications address your concerns.